# Causal Component Analysis

**Liang Wendong** [1,2]    **Armin Kekić** [1]    **Julius von Kügelgen** [1,3]    **Simon Buchholz** [1]

**Michel Besserve** [1]    **Luigi Gresele**[*1]    **Bernhard Schölkopf**[*1]

[1] Max Planck Institute for Intelligent Systems, Tübingen, Germany
[2] ENS Paris-Saclay, Gif-sur-Yvette, France    [3] University of Cambridge, United Kingdom
`{wendong.liang,armin.kekic,jvk,simon.buchholz}@tue.mpg.de`
`{besserve,luigi.gresele,bs}@tue.mpg.de`

## Abstract

Independent Component Analysis (ICA) aims to recover independent latent variables from observed mixtures thereof. Causal Representation Learning (CRL) aims instead to infer causally related (thus often statistically *dependent*) latent variables, together with the unknown graph encoding their causal relationships. We introduce an intermediate problem termed *Causal Component Analysis (CauCA)*. CauCA can be viewed as a generalization of ICA, modelling the causal dependence among the latent components, and as a special case of CRL. In contrast to CRL, it presupposes knowledge of the causal graph, focusing solely on learning the unmixing function and the causal mechanisms. Any impossibility results regarding the recovery of the ground truth in CauCA also apply for CRL, while possibility results may serve as a stepping stone for extensions to CRL. We characterize CauCA identifiability from multiple datasets generated through different types of interventions on the latent causal variables. As a corollary, this interventional perspective also leads to new identifiability results for nonlinear ICA—a special case of CauCA with an empty graph—requiring strictly fewer datasets than previous results. We introduce a likelihood-based approach using normalizing flows to estimate both the unmixing function and the causal mechanisms, and demonstrate its effectiveness through extensive synthetic experiments in the CauCA and ICA setting.

## 1 Introduction

Independent Component Analysis (ICA) [7] is a principled approach to representation learning, which aims to recover independent latent variables, or sources, from observed mixtures thereof. Whether this is possible depends on the *identifiability* of the model [32, 57]: this characterizes assumptions under which a learned representation provably recovers (or *disentangles*) the latent variables, up to some well-specified ambiguities [22, 58]. A key result shows that, when nonlinear mixtures of the latent components are observed, the model is non-identifiable based on independent and identically distributed (i.i.d.) samples from the generative process [8, 19]. Consequently, a learned model may explain the data equally well as the ground truth, even if the corresponding representation is strongly entangled, rendering the recovery of the original latent variables fundamentally impossible.

Identifiability can be recovered under deviations from the i.i.d. assumption, e.g., in the form of temporal autocorrelation [14, 18] or spatial dependence [15] among the latent components; *auxiliary variables* which render the sources *conditionally* independent [17, 21, 25, 26]; or additional, noisy *views* [11]. An alternative path is to restrict the class of mixing functions [4, 13, 63].

---

[*]Shared last author.    Code available at https://github.com/akekic/causal-component-analysis.

37th Conference on Neural Information Processing Systems (NeurIPS 2023).

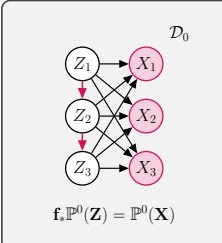 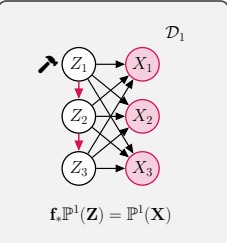 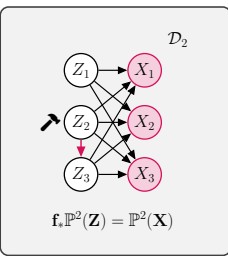 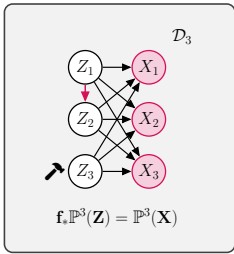

Figure 1: **Causal Component Analysis (CauCA).** We posit that observed variables $\mathbf{X}$ are generated through a nonlinear mapping $\mathbf{f}$, applied to unobserved latent variables $\mathbf{Z}$ which are causally related. The causal structure $G$ of the latent variables is assumed to be known, while the causal mechanisms $\mathbb{P}_i(Z_i \mid \mathbf{Z}_{\text{pa}(i)})$ and the nonlinear mixing function are unknown and to be estimated. (Known or observed quantities are highlighted in red.) CauCA assumes access to multiple datasets $\mathcal{D}_k$ that result from stochastic interventions on the latent variables.

Despite appealing identifiability guarantees for ICA, the independence assumption can be limiting, since interesting factors of variation in real-world data are often statistically, or causally, dependent [51]. This motivates Causal Representation Learning (CRL) [45], which aims instead to infer causally related latent variables, together with a causal graph encoding their causal relationships. This is challenging if both the graph and the unmixing are unknown. Identifiability results in CRL therefore require strong assumptions such as counterfactual data [1, 3, 9, 37, 54], temporal structure [30, 33], a parametric family of latent distributions [5, 6, 30, 47, 59, 60], graph sparsity [28–30], pairs of interventions and *genericity* [55] or restrictions on the mixing function class [2, 49, 52]. It has been argued that knowing either the graph or the unmixing might help better recover the other, giving rise to a *chicken-and-egg* problem in CRL [3].

We introduce an intermediate problem between ICA and CRL which we call *Causal Component Analysis (CauCA)*, see Fig. 1 for an overview. CauCA can be viewed as a generalization of ICA that models causal connections (and thus statistical dependence) among the latent components through a causal Bayesian network [41]. It can also be viewed as a special case of CRL that presupposes knowledge of the causal graph, and focuses on learning the unmixing function and causal mechanisms.

Since CauCA is solving the CRL problem with partial ground truth information, it is strictly easier than CRL. This implies that impossibility results for CauCA also apply for CRL. *Possibility* results for CauCA, on the other hand, while not automatically generalizing to CRL, can nevertheless serve as stepping stones, highlighting potential avenues for achieving corresponding results in CRL. Note also that there are only finitely many possible directed acyclic graphs for a fixed number of nodes, but the space of spurious solutions in representation learning (e.g., in nonlinear ICA) is typically infinite. By solving CauCA problems, we can therefore gain insights into the minimal assumptions required for addressing CRL problems. CauCA may be applicable to scenarios in which domain knowledge can be used to specify a causal graph for the latent components. For instance, in computer vision applications, the image generation process can often be modelled based on a fixed graph [44, 50].

**Structure and Contributions.** We start by recapitulating preliminaries on causal Bayesian networks and interventions in § 2. Next, we introduce Causal Component Analysis (CauCA) in § 3. Our primary focus lies in characterizing the identifiability of CauCA from multiple datasets generated through various types of interventions on the latent causal variables (§ 4). Importantly, all our results are applicable to the *nonlinear* and *nonparametric* case. The interventional perspective we take exploits the *modularity* of the causal relationships (i.e., the possibility to change one of them without affecting the others)—a concept that was not previously leveraged in works on nonlinear ICA. This leads to extensions of existing results that require strictly fewer datasets to achieve the same level of identifiability. We introduce and investigate an estimation procedure for CauCA in § 5, and conclude with a summary of related work (§ 6) and a discussion (§ 7). We highlight the following *main contributions*:

- We derive sufficient and necessary conditions for identifiability of CauCA from different types of interventions (Thm. 4.2, Prop. 4.3, Thm. 4.5).
- We prove additional results for the special case with an empty graph, which corresponds to a novel ICA model with interventions on the latent variables (Prop. 4.6, Prop. 4.7, Corollary 4.8, Prop. 4.9).
- We show in synthetic experiments in both the CauCA and ICA settings that our normalizing flow-based estimation procedure effectively recovers the latent causal components (§ 5).

## 2 Preliminaries

**Notation.** We use $\mathbb{P}$ to denote a probability distribution, with density function $p$. Uppercase letters $X, Y, Z$ denote unidimensional and bold uppercase $\mathbf{X}, \mathbf{Y}, \mathbf{Z}$ denote multidimensional random variables. Lowercase letters $x, y, z$ denote scalars in $\mathbb{R}$ and $\mathbf{x}, \mathbf{y}, \mathbf{z}$ denote vectors in $\mathbb{R}^d$. We use $[\![i, j]\!]$ to denote the integers from $i$ to $j$, and $[d]$ denotes the natural numbers from 1 to $d$. We use common graphical notation, see App. A for details. The *ancestors* of $i$ in a graph are the nodes $j$ in $G$ such that there is a directed path from $j$ to $i$, and they are denoted by $\mathrm{anc}(i)$. The *closure* of the parents (resp. ancestors) of $i$ is defined as $\overline{\mathrm{pa}}(i) := \mathrm{pa}(i) \cup \{i\}$ (resp. $\overline{\mathrm{anc}}(i) := \mathrm{anc}(i) \cup \{i\}$).

A key definition connecting directed acyclic graphs (DAGs) and probabilistic models is the following.

**Definition 2.1** (Distribution Markov relative to a DAG [41]). *A joint probability distribution $\mathbb{P}$ is Markov relative to a DAG $G$ if it admits the factorization* $\mathbb{P}(Z_1, \ldots, Z_d) = \prod_{i=1}^d \mathbb{P}_i(Z_i | \mathbf{Z}_{pa(i)})$.

Defn. 2.1 is a key assumption in directed graphical models, where a distribution being Markovian relative to a graph implies that the graph encodes specific independences within the distribution, which can be exploited for efficient computation or data storage [43, §6.5].

**Causal Bayesian networks and interventions.** Causal systems induce multiple distributions corresponding to different interventions. Causal Bayesian networks [CBNs; 41] can be used to represent how these interventional distributions are related. In a CBN with associated graph $G$, arrows signify causal links among variables, and the conditional probabilities $\mathbb{P}_i(Z_i | \mathbf{Z}_{\mathrm{pa}(i)})$ in the corresponding Markov factorization are called *causal mechanisms*.[2]

Interventions are modelled in CBNs by replacing a subset $\tau_k \subseteq V(G)$ of the causal mechanisms by new, intervened mechanisms $\{\widetilde{\mathbb{P}}_j(Z_j | \mathbf{Z}_{\mathrm{pa}^k(j)})\}_{j \in \tau_k}$, while all other causal mechanisms are left unchanged. Here, $\mathrm{pa}^k(j)$ denotes the parents of $Z_j$ in the post-intervention graph in the interventional regime $k$ and $\tau_k$ the intervention targets. We will omit the superscript $k$ when the parent set is unchanged and assume that interventions do not add new parents, $\mathrm{pa}^k(j) \subseteq \mathrm{pa}(j)$. Unless $\widetilde{\mathbb{P}}_j$ is a point mass, we call the intervention *stochastic* or soft. Further, we say that $\widetilde{\mathbb{P}}_j$ is a *perfect* intervention if the dependence of the $j$-th variable from its parents is removed ($\mathrm{pa}^k(j) = \varnothing$), corresponding to deleting all arrows pointing to $i$, sometimes also referred to as *graph surgery* [48].[3] An *imperfect* intervention is one for which $\mathrm{pa}^k(j) \neq \varnothing$. We summarise this in the following definition.

**Definition 2.2** (CBN). *A causal Bayesian network (CBN) consists of a graph $G$, a collection of causal mechanisms $\{\mathbb{P}_i(Z_i | \mathbf{Z}_{pa(i)})\}_{i \in [d]}$, and a collection of interventions $\{\{\widetilde{\mathbb{P}}_j^k(Z_j | \mathbf{Z}_{pa^k(j)})\}_{j \in \tau_k}\}_{k \in [K]}$ across $K$ interventional regimes. The joint probability for interventional regime $k$ is given by:*

$$\mathbb{P}^k(\mathbf{Z}) := \begin{cases} \prod_{i=1}^d \mathbb{P}_i(Z_i | \mathbf{Z}_{pa(i)}) & k = 0 \\ \prod_{j \in \tau_k} \widetilde{\mathbb{P}}_j^k(Z_j | \mathbf{Z}_{pa^k(j)}) \prod_{i \notin \tau_k} \mathbb{P}_i(Z_i | \mathbf{Z}_{pa(i)}) & \forall k \in [K] \end{cases} \tag{1}$$

*where $\mathbb{P}^0$ is the* unintervened*, or observational, distribution, and $\mathbb{P}^k$ are interventional distributions.*

*Remark* 2.3. The joint probabilities $\mathbb{P}^k$ in (1) are uniquely factorized into causal mechanisms according to $G$. We therefore use the equivalent notation $(G, (\mathbb{P}^k, \tau_k)_{k \in [\![0, K]\!]})$, where $\mathbb{P}^k$ is defined as in (1).

## 3 Problem Setting

The main object of our study is a latent variable model termed *latent causal Bayesian network (CBN)*.

**Definition 3.1** (Latent CBN). *A latent CBN is a tuple $(G, \mathbf{f}, (\mathbb{P}^k, \tau_k)_{k \in [\![0, K]\!]})$, where $\mathbf{f} : \mathbb{R}^d \to \mathbb{R}^d$ is a diffeomorphism (i.e. invertible with both $\mathbf{f}$ and $\mathbf{f}^{-1}$ differentiable).*

---

[2]The term can also be used in structural causal models to denote deterministic functions of endogenous and exogenous variables in *assignments*, see [43, Def. 3.1]. A central idea in causality [41, 43] is that causal mechanisms are *modular* or *independent*, i.e., it is possible to modify some without affecting the others: after an intervention, typically only a subset of the causal mechanisms change.

[3]A special case of perfect interventions are *hard* interventions, where $\widetilde{\mathbb{P}}_j$ corresponds to a Dirac distribution: $\mathbb{P}(\mathbf{Z} | \mathrm{do}(Z_j = z_j)) = \delta_{Z_j = z_j} \prod_{i \neq j} \mathbb{P}_i(Z_i | \mathbf{Z}_{\mathrm{pa}(i)})$.

**Data-generating process for Causal Component Analysis (CauCA).** In CauCA, we assume that we are given multiple datasets $\{\mathcal{D}_k\}_{k \in [\![0,K]\!]}$ generated by a latent CBN $(G, \mathbf{f}, (\mathbb{P}^k, \tau_k)_{k \in [\![0,K]\!]})$:

$$\mathcal{D}_k := \left( \tau_k, \left\{ \mathbf{x}^{(n,k)} \right\}_{n=1}^{N_k} \right), \quad \text{with} \quad \mathbf{x}^{(n,k)} = \mathbf{f}\left( \mathbf{z}^{(n,k)} \right) \quad \text{and} \quad \mathbf{z}^{(n,k)} \overset{\text{i.i.d.}}{\sim} \mathbb{P}^k, \quad (2)$$

where $N_k$ denotes the sample size for interventional regime $k$, see Fig. 1 for an illustration. The graph $G$ is assumed to be known. Further, we assume that the intervention targets $\tau_k$ are observed, see § 7 for further discussion. Both the mixing function $\mathbf{f}$ and the latent distributions $\mathbb{P}^k$ in (2) are unknown.

The problem we aim to address is the following: given only the graph $G$ and the datasets $\mathcal{D}_k$ in (2), can we learn to *invert* the mixing function $\mathbf{f}$ and thus recover the latent variables $\mathbf{z}$? Whether this is possible, and up to what ambiguities, depends on the identifiability of CauCA.

**Definition 3.2** (Identifiability of CauCA). *A model class for CauCA is a tuple $(G, \mathcal{F}, \mathcal{P}_G)$, where $\mathcal{F}$ is a class of functions and $\mathcal{P}_G$ is a class of joint distributions Markov relative to $G$. A latent CBN $(G, \mathbf{f}, (\mathbb{P}^k, \tau_k)_{k \in [\![0,K]\!]})$ is said to be in $(G, \mathcal{F}, \mathcal{P}_G)$ if $\mathbf{f} \in \mathcal{F}$ and $\mathbb{P}^k \in \mathcal{P}_G$ for all $k \in [\![0, K]\!]$. We say $(G, \mathcal{F}, \mathcal{P}_G)$ has known intervention targets if all its elements share the same $G$ and $(\tau_k)_{k \in [\![0,K]\!]}$.*

*We say that CauCA in $(G, \mathcal{F}, \mathcal{P}_G)$ is identifiable up to $\mathcal{S}$ (a set of functions called "indeterminacy set") if for any two latent CBNs $(G, \mathbf{f}, (\mathbb{P}^k, \tau_k)_{k \in [\![0,K]\!]})$ and $(G', \mathbf{f}', (\mathbb{Q}^k, \tau_k)_{k \in [\![0,K]\!]})$, the equality of pushforward $\mathbf{f}_* \mathbb{P}^k = \mathbf{f}'_* \mathbb{Q}^k \ \forall k \in [\![0, K]\!]$ implies that $\exists \mathbf{h} \in \mathcal{S}$ s.t. $\mathbf{h} = \mathbf{f}'^{-1} \circ \mathbf{f}$ on the support of $\mathbb{P}$.*

We justify the definition of known intervention targets and generalize them to a more flexible scenario in App. E. Defn. 3.2 is inspired by the identifiability definition of ICA in [4, Def. 1]. Intuitively, it states that, if two models in $(G, \mathcal{F}, \mathcal{P}_G)$ give rise to the same distribution, then they are equal up to ambiguities specified by $\mathcal{S}$. Consequently, when attempting to invert $\mathbf{f}$ based on the data in (2), the inversion can only be achieved up to those ambiguities.

In the following, we choose $\mathcal{F}$ to be the class of all $\mathcal{C}^1$-diffeomorphisms $\mathbb{R}^d \to \mathbb{R}^d$, denoted $\mathcal{C}^1(\mathbb{R}^d)$, and suppose the distributions in $\mathcal{P}_G$ are absolutely continuous with full support in $\mathbb{R}^d$, with the density $p^k$ differentiable.

A first question is what ambiguities are unavoidable by construction in CauCA, similar to scaling and permutation in ICA [22, § 3.1]. The following Lemma characterizes this.

**Lemma 3.3.** *For any $(G, \mathbf{f}, (\mathbb{P}^k, \tau_k)_{k \in [\![0,K]\!]})$ in $(G, \mathcal{F}, \mathcal{P}_G)$, and for any $\mathbf{h} \in \mathcal{S}_{scaling}$ with*

$$\mathcal{S}_{scaling} := \left\{ \mathbf{h} : \mathbb{R}^d \to \mathbb{R}^d \mid \mathbf{h}(\mathbf{z}) = (h_1(z_1), \dots, h_d(z_d)), \ h_i \text{ is a diffeomorphism in } \mathbb{R} \right\}, \quad (3)$$

*there exists a $(G, \mathbf{f} \circ \mathbf{h}, (\mathbb{Q}^k, \tau_k)_{k \in [\![0,K]\!]})$ in $(G, \mathcal{F}, \mathcal{P}_G)$ s.t. $\mathbf{f}_* \mathbb{P}^k = (\mathbf{f} \circ \mathbf{h})_* \mathbb{Q}^k$ for all $k \in [\![0, K]\!]$.*

Lemma 3.3 states that, as in nonlinear ICA, the ambiguity up to element-wise nonlinear scaling is also unresolvable in CauCA. However, unlike in nonlinear ICA, there is no permutation ambiguity in CauCA: this is a consequence of the assumption of known intervention targets. The next question is under which conditions we can achieve identifiability up to (3), and when the ambiguity set is larger.

## 4 Theory

In this section, we investigate the identifiability of CauCA. We first study the general case (§ 4.1), and then consider the special case of ICA in which the graph is empty (§ 4.2).

### 4.1 Identifiability of CauCA

**Single-node interventions.** We start by characterizing the identifiability of CauCA based on *single-node* interventions. For datasets $\mathcal{D}_k$ defined as in (2), every $k > 0$ corresponds to interventions on a single variable: i.e., $\forall k > 0, |\tau_k| = 1$. This is the setting depicted in Fig. 1, where each interventional dataset is generated by intervening on a single latent variable. The following assumption will play a key role in our proofs.

**Assumption 4.1** (Interventional discrepancy). *Given $k \in [K]$, let $p_{\tau_k}$ denote the causal mechanism of $z_{\tau_k}$. We say that a stochastic intervention $\tilde{p}_{\tau_k}$ satisfies interventional discrepancy if*

$$\frac{\partial(\ln p_{\tau_k})}{\partial z_{\tau_k}} \left( z_{\tau_k} \mid \mathbf{z}_{pa(\tau_k)} \right) \neq \frac{\partial(\ln \tilde{p}_{\tau_k})}{\partial z_{\tau_k}} \left( z_{\tau_k} \mid \mathbf{z}_{pa^k(\tau_k)} \right) \quad \text{almost everywhere (a.e.).} \quad (4)$$

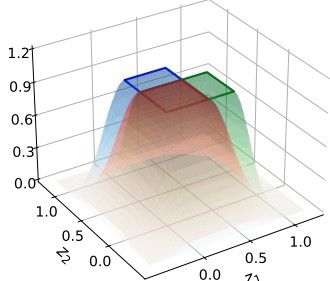 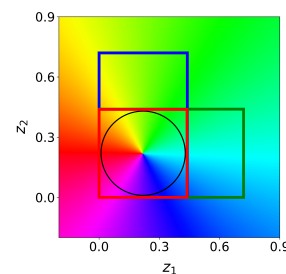 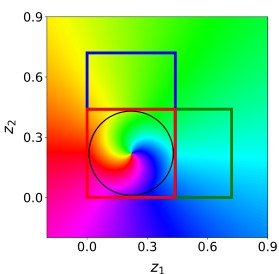

Figure 2: **Violation of the Interventional Discrepancy Assumption.** The shown distributions constitute a counterexample to identifiability that violates Asm. 4.1 and thus allows for spurious solutions, see App. C for technical details. *(Left)* Visualisation of the joint distributions of two independent latent components $z_1$ and $z_2$ after no intervention (red), and interventions on $z_1$ (green) and $z_2$ (blue). As can be seen, each distribution reaches the same plateau on some rectangular interval of the domain, coinciding within the red square. *(Center/Right)* Within the red square where all distributions agree, it is possible to apply a measure preserving automorphism which leaves all distributions unchanged, but non-trivially mixes the latents. The right plot shows a distance-dependent rotation around the centre of the black circle, whereas the middle plot show a reference identity transformation.

Asm. 4.1 can be applied to imperfect and perfect interventions alike (in the latter case the conditioning on the RHS disappears). Intuitively, Asm. 4.1 requires that the stochastic intervention is sufficiently different from the causal mechanism, formally expressed as the requirement that the partial derivative over $z_i$ of the ratio between $p_i$ and $\tilde{p}_i$ is nonzero a.e. One case in which Asm. 4.1 is violated is when $\partial p_i / \partial z_i$ and $\partial \tilde{p}_i / \partial z_i$ are both zero on the same open subset of their support. In Fig. 2 *(Left)*, we provide an example of such a violation (see App. C for its construction), and apply a measure-preserving automorphism within the area where the two distributions agree (see Fig. 2 *(Right)*).

We can now state our main result for CauCA with single-node interventions.

**Theorem 4.2.** *For CauCA in* $(G, \mathcal{F}, \mathcal{P}_G)$,

(i) *Suppose for each node in* $[d]$, *there is one (perfect or imperfect) stochastic intervention that satisfies Asm. 4.1. Then CauCA in* $(G, \mathcal{F}, \mathcal{P}_G)$ *is identifiable up to*

$$\mathcal{S}_{\overline{G}} = \left\{ \mathbf{h} : \mathbb{R}^d \to \mathbb{R}^d | \mathbf{h}(\mathbf{z}) = \left( h_i(\mathbf{z}_{\overline{anc}(i)}) \right)_{i \in [d]}, \mathbf{h} \text{ is } \mathcal{C}^1\text{-diffeomorphism} \right\}. \tag{5}$$

(ii) *Suppose for each node* $i$ *in* $[d]$, *there is one perfect stochastic intervention that satisfies Asm. 4.1. Then CauCA in* $(G, \mathcal{F}, \mathcal{P}_G)$ *is identifiable up to* $\mathcal{S}_{scaling}$.

Thm. 4.2 *(i)* states that for single-node stochastic interventions, perfect or imperfect, we can achieve identifiability up to an indeterminacy set where each reconstructed variable can at most be a mixture of ground truth variables corresponding to nodes in the closure of the ancestors of $i$. While this ambiguity set is larger than the one in eq. (3), it is still a non-trivial reduction in ambiguity with respect to the spurious solutions which could be generated without Asm. 4.1. A related result in [49, Thm. 1] shows that for *linear mixing, linear latent SCM and unknown graph*, $d$ interventions are sufficient and necessary for recovering $\overline{G}$ (the transitive closure of the ground truth graph $G$) and the latent variables up to elementwise reparametrizations. Thm. 4.2 *(i)* instead proves that $d$ interventions are sufficient for identifiability up to mixing of variables corresponding to the coordinates in $\overline{G}$ for *arbitrary* $\mathcal{C}^1$*-diffeomorphisms* $\mathcal{F}$, *non-parametric* $\mathcal{P}_G$ *and known graph*.

Thm. 4.2 *(ii)* shows that if we further constrain the set of interventions to *perfect* single-node, stochastic interventions only, then we can achieve a much stronger identifiability—i.e., identifiability up to scaling, which as discussed in § 3 is the best one we can hope to achieve in our problem setting without further assumptions. In short, the unintervened distribution together with one single-node, stochastic perfect intervention per node is sufficient to give us the strongest achievable identifiability in our considered setting. In App. D, we also discuss identifiability when only imperfect stochastic interventions are available. In short, with a higher number of imperfect interventions, the ambiguity in Thm. 4.2 *(i)* can be further constrained to the closure of parents, instead of the closure of ancestors.

Thm. 4.2 *(ii)* shows that $d$ datasets generated by single-node interventions on the latent causal variables are sufficient for identifiability up to $\mathcal{S}_{\text{scaling}}$. We additionally prove below that $d$ interventional datasets are *necessary*—i.e., for CauCA, and for any nonlinear causal representation learning problem, $d - 1$ single-node interventions are not sufficient for identifiability up to $\mathcal{S}_{\text{scaling}}$.

**Proposition 4.3.** *Given a DAG $G$, with $d - 1$ perfect stochastic single node interventions on distinct targets, if the remaining unintervened node has any parent in $G$, $(G, \mathcal{F}, P_G)$ is not identifiable up to*

$$\mathcal{S}_{\text{reparam}} := \left\{ \mathbf{g} : \mathbb{R}^d \to \mathbb{R}^d \mid \mathbf{g} = \mathbf{P} \circ \mathbf{h}, \ \mathbf{P} \text{ is a permutation matrix, } \mathbf{h} \in \mathcal{S}_{\text{scaling}} \right\}. \tag{6}$$

A similar result in [49] shows that one intervention per node is necessary when the underlying graph is unknown. Prop. 4.3 shows that this is still the case, *even when the graph is known*.

**Fat-hand interventions.** A generalization of single-node interventions are *fat-hand interventions*—i.e., interventions where $|\tau_k| > 1$. In this section, we study this more general setting and focus on a weaker form of identification than for single-node intervention.

**Assumption 4.4** (Block-interventional discrepancy)**.** *We denote $\mathbb{Q}^0_{\tau_k}$ as the causal mechanism of $\mathbf{Z}_{\tau_k}$ in the unintervened regime. For each $k \in [K]$, we denote $\mathbb{Q}^s_{\tau_k}$ as the intervention mechanism in the $s$-th interventional regime that has $\tau_k \subseteq [d]$ as targets of intervention, i.e., $\mathbb{Q}^s_{\tau_k}$ is a (conditional) joint distribution over $\mathbf{Z}_{\tau_k}$. Then the* Block-interventional discrepancy *for $\tau_k$ is defined as follows:*

- *if there is no arrow into $\tau_k$ (i.e., $\tau_k$ has no parents in $[d] \setminus \tau_k$), suppose that there are $n_k$ interventions with target $\tau_k$ such that the following $n_k \times n_k$ matrix*

$$\mathbf{M}_{\tau_k} := \begin{pmatrix} \frac{\partial}{\partial z_1}(\ln q^1_{\tau_k} - \ln q^0_{\tau_k})(\mathbf{z}_{\tau_k}) & \cdots & \frac{\partial}{\partial z_{n_k}}(\ln q^1_{\tau_k} - \ln q^0_{\tau_k})(\mathbf{z}_{\tau_k}) \\ \vdots & & \vdots \\ \frac{\partial}{\partial z_1}(\ln q^{n_k}_{\tau_k} - \ln q^0_{\tau_k})(\mathbf{z}_{\tau_k}) & \cdots & \frac{\partial}{\partial z_{n_k}}(\ln q^{n_k}_{\tau_k} - \ln q^0_{\tau_k})(\mathbf{z}_{\tau_k}) \end{pmatrix} \tag{7}$$

*is invertible for $\mathbf{z}_{\tau_k} \in \mathbb{R}^{n_k}$ almost everywhere, where $q^s_{\tau_k,j}$ denotes the $j$-th marginal of $q^s_{\tau_k}$, $z_{\tau_k,j}$ denotes the $j$-th dimension of $\mathbf{z}_{\tau_k}$, and $s = 0$ denotes the unintervened (observational) regime;*

- *otherwise, suppose that there are $n_k + 1$ interventions with target $\tau_k$ such that the matrix (7) is invertible for $\mathbf{z}_{\tau_k} \in \mathbb{R}^{n_k}$ almost everywhere, where $q^s_{\tau_k,j}$ and $z_{\tau_k,j}$ are defined as before, but $s = 0, \ldots, n_k$ now indexes the $n_k + 1$ interventions—i.e., without an unintervened regime.*

Asm. 4.4 is tightly connected to Asm. 4.1: if $G$ has no arrow, if $\forall k : n_k = 1$, then Asm. 4.4 is reduced to Asm. 4.1. However, for any $G$ that has arrows, the number of interventional regimes required by Asm. 4.4 is strictly larger than Asm. 4.1.

**Theorem 4.5.** *Given any DAG $G$. Suppose that our datasets encompass interventions over all variables in the latent graph, i.e., $\bigcup_{k \in [K]} \tau_k = [d]$. For all $k \in [K]$, suppose the targets of interventions are strict subsets of all variables, i.e., $|\tau_k| = n_k$, $n_k \in [d-1]$. Suppose the interventions over $\tau_k$ are perfect, i.e. the intervention mechanisms $\mathbb{Q}^s_{\tau_k}$ are joint distributions over $\mathbf{Z}_{\tau_k}$ without conditioning on other variables. Suppose Asm. 4.4 is satisfied for $\tau_k$.*

*Then CauCA in $(G, \mathcal{F}, \mathcal{P}_G)$ is* block-identifiable *(following [54]): namely, if for all $k \in [K]$, $\mathbf{f}_* \mathbb{P}^k = \mathbf{f}'_* \mathbb{Q}^k$, then for $\boldsymbol{\varphi} := \mathbf{f}'^{-1} \circ \mathbf{f}$, for all $k \in [K]$,*

$$[\boldsymbol{\varphi}(\mathbf{z})]_{\tau_k} = \boldsymbol{\varphi}_{\tau_k}(\mathbf{z}_{\tau_k}). \tag{8}$$

We illustrate the above identifiability results through an example in Tab. 1.

## 4.2 Special Case: ICA with stochastic interventions on the latent components

An important special case of CauCA occurs when the graph $G$ is empty, corresponding to independent latent components. This defines a nonlinear ICA generative model where, in addition to the mixtures, we observe a variable $\tau_k$ which indicates *which latent distributions* change in the interventional regime $k$, while *every other distribution is unchanged*.[4] This nonlinear ICA generative model is closely related to similar models with observed *auxiliary variables* [21, 25]: it is natural to interpret

---

[4]The relationship between CauCA and nonlinear ICA is discussed in more detail in App. F.

Table 1: **Overview of identifiability results.** For the DAG $Z_1 \rightarrow Z_2 \rightarrow Z_3$ from Fig. 1, we summarise the guarantees provided by our theoretical analysis in § 4.1 for representations learnt by maximizing the likelihoods $\mathbb{P}_\theta^k(X)$ for different sets of interventional regimes.

| Requirement of interventions | Learned representation $\hat{\mathbf{z}} = \hat{\mathbf{f}}^{-1}(\mathbf{x})$ | Reference |
|---|---|---|
| 1 intervention per node | $[h_1(z_1), h_2(z_1, z_2), h_3(z_1, z_2, z_3)]$ | Thm. 4.2 *(i)* |
| 1 perfect intervention per node | $[h_1(z_1), h_2(z_2), h_3(z_3)]$ | Thm. 4.2 *(ii)* |
| 1 intervention per node for $z_1$ and $z_2$, plus $\lvert\overline{\mathrm{pa}}(3)\rvert(\lvert\overline{\mathrm{pa}}(3)\rvert+1) = 2\times3$ imperfect interventions on $z_3$ with "variability" assumption | $[h_1(z_1), h_2(z_2), h_3(z_2, z_3)]$ | Prop. D.1 |
| 1 perfect intervention on $z_1$ and $2+1=3$ perfect fat-hand interventions on $(z_2, z_3)$ | $[h_1(z_1), h_2(z_2, z_3), h_3(z_2, z_3)]$ | Thm. 4.5 |

$\tau_k$ itself as an auxiliary variable. As we will see, our interventional interpretation allows us to derive novel results and re-interpret existing ones. Below, we characterize identifiability for this setting.

**Single-node interventions.** We first focus on *single-node* stochastic interventions, where the following result proves that we can achieve the same level of identifiability as in Thm. 4.2 *(ii)*, with one less intervention than in the case where the graph is non-trivial.

**Proposition 4.6.** *Suppose that $G$ is the empty graph, and that there are $d-1$ variables intervened on, with one single target per dataset, such that Asm. 4.1 is satisfied. Then CauCA (in this case, ICA) in $(G, \mathcal{F}, \mathcal{P}_G)$ is identifiable up to $\mathcal{S}_{scaling}$ defined as in eq. (3).*

The result above shows that identifiability can be achieved through single-node interventions on the latent variables using strictly fewer datasets (i.e., auxiliary variables) than previous results in the auxiliary variables setting ($d$ in our case, $2d+1$ in [21, Thm. 1]). One potentially confusing aspect of Prop. 4.6 is that the ambiguity set does not contain permutations—which is usually an unavoidable ambiguity in ICA. This is due to our considered setting with known targets, where a total ordering of the variables is assumed to be known. The result above can also be extended to the case of *unknown intervention targets*, where we only know that, in each dataset, a distinct variable is intervened on, but we do not know *which one* (see App. E). For that case, we prove (Prop. E.6) that ICA in $(G, \mathcal{F}, \mathcal{P}_G)$ is in fact identifiable up to scaling and *permutation*. Note that Prop. 4.6 is not a special case of Thm. 4.5 in which $n_k = 1 \; \forall k$, since it only requires $d-1$ interventions instead of $d$.

We can additionally show that for nonlinear ICA, $d-1$ interventions are *necessary* for identifiability.

**Proposition 4.7.** *Given an empty graph $G$, with $d-2$ single-node interventions on distinct targets, with one single target per dataset, such that Asm. 4.1 is satisfied. Then CauCA (in this case, ICA) in $(G, \mathcal{F}, \mathcal{P}_G)$ is not identifiable up to $\mathcal{S}_{reparam}$.*

**Fat-hand interventions.** For the special case with independent components, the following corollary characterises identifiability under fat-hand interventions.

**Corollary 4.8.** *[Corollary of Thm. 4.5] Suppose $G$ is the empty graph. Suppose that our datasets encompass interventions over all variables in the latent graph, i.e., $\bigcup_{k\in[K]} \tau_k = [d]$. Suppose for every $k$, the targets of interventions are a strict subset of all variables, i.e., $\lvert \tau_k \rvert = n_k, \; n_k \in [d-1]$.*

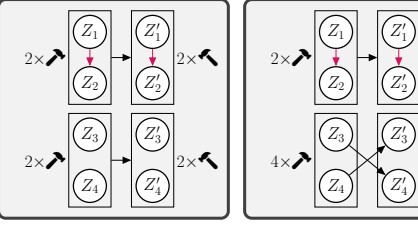

Figure 3: We use the "🔨" symbol together with a "times" symbol to represent how many interventions are required by the two assumptions. *(Left)* (Thm. 4.5) For Asm. 4.4, we need $n_k$ interventions to get block-identification of $\mathbf{z}_{\tau_k}$. *(Right)* (Prop. 4.9) For the *block-variability* assumption, we need $2n_k$ to get to elementwise identification up to scaling and permutation.

*Suppose Asm. 4.4 is verified, which has a simpler form in this case: there are $n_k$ interventions with target $\tau_k$ such that $\mathbf{v}_k(\mathbf{z}_{\tau_k}, 1) - \mathbf{v}_k(\mathbf{z}_{\tau_k}, 0), \cdots, \mathbf{v}_k(\mathbf{z}_{\tau_k}, n_k) - \mathbf{v}_k(\mathbf{z}_{\tau_k}, 0)$ are linearly independent, where*

$$\mathbf{v}_k(\mathbf{z}_{\tau_k}, s) := \left( (\ln q^s_{\tau_k, 1})'(z_{\tau_k, 1}), \cdots, (\ln q^s_{\tau_k, n_k})'(z_{\tau_k, n_k}) \right), \tag{9}$$

*where $q^s_{\tau_k}$ is the intervention of the $s$-th interventional regime that has the target $\tau_k$, and $q^s_{\tau_k, j}$ is the $j$-th marginal of it. $z_{\tau_k, j}$ is the $j$-th dimension of $\mathbf{z}_{\tau_k}$. $s = 0$ denotes the unintervened regime.*

*Then CauCA in $(G, \mathcal{F}, \mathcal{P}_G)$ is block-identifiable, in the same sense as Thm. 4.5.*

Our interventional perspective also allows us to re-interpret and extend a key result in the theory of nonlinear ICA with auxiliary variables, [21, Thm.1]. In particular, the following Proposition holds.

**Proposition 4.9.** *Under the assumptions of Thm. 4.5, suppose furthermore that all density functions in $\mathcal{P}_G$ and all mixing functions in $\mathcal{F}$ are $\mathcal{C}^2$, and suppose there exist $k \in [K]$ and there are $2n_k$ interventions with targets $\tau_k$ such that for any $\mathbf{z}_{\tau_k} \in \mathbb{R}^{n_k}$, $\mathbf{w}_k(\mathbf{z}_{\tau_k}, 1) - \mathbf{w}_k(\mathbf{z}_{\tau_k}, 0), \ldots, \mathbf{w}_k(\mathbf{z}_{\tau_k}, 2n_k) - \mathbf{w}_k(\mathbf{z}_{\tau_k}, 0)$ are linearly independent, where*

$$\mathbf{w}_k(\mathbf{z}_{\tau_k}, s) := \left( \left( \frac{q^{s'}_{\tau_k, 1}}{q^s_{\tau_k, 1}} \right)'(z_{\tau_k, 1}), \ldots, \left( \frac{q^{s'}_{\tau_k, n_k}}{q^s_{\tau_k, n_k}} \right)'(z_{\tau_k, n_k}), \frac{q^{s'}_{\tau_k, 1}}{q^s_{\tau_k, 1}}(z_{\tau_k, 1}), \ldots, \frac{q^{s'}_{\tau_k, n_k}}{q^s_{\tau_k, n_k}}(z_{\tau_k, n_k}) \right),$$

*where $q^s_{\tau_k}$ is the intervention of the $s$-th interventional regime that has the target $\tau_k$, and $q^s_{\tau_k, j}$ is the $j$-th marginal of it. $z_{\tau_k, j}$ is the $j$-th dimension of $\mathbf{z}_{\tau_k}$. $s = 0$ denotes the unintervened regime. Then*

$$\varphi_{\tau_k} \in \mathcal{S}_{reparam} := \left\{ \mathbf{g} : \mathbb{R}^{n_k} \to \mathbb{R}^{n_k} | \mathbf{g} = \mathbf{P} \circ \mathbf{h} \text{ where } \mathbf{P} \text{ is a permutation matrix and } \mathbf{h} \in \mathcal{S}_{scaling} \right\}.$$

*Remark* 4.10. The assumption of linear independence $\mathbf{w}_k(\mathbf{z}_{\tau_k}, s), s \in [2n_k]$ precisely corresponds to the assumption of *variability* in [21, Thm. 1]; however, we only assume it within a $n_k$-dimensional block (not over all $d$ variables). We refer to it as *block-variability*.

Note that the block-variability assumption *implies* block-interventional discrepancy (Asm. 4.4): i.e., it is a strictly stronger assumption, which, correspondingly, leads to a stronger identification. In fact, block-interventional discrepancy only allows *block-wise identifiability* within the $n_k$-dimensional intervened blocks based on $n_k$ interventions. In contrast, the variability assumption can be interpreted as a sufficient assumption to achieve identification *up to permutation and scaling* within a $n_k$-dimensional block, based on $2n_k$ fat-hand interventions (in both cases one unintervened dataset is required), see Fig. 3 for a visualization. We summarise our results for nonlinear ICA in Tab. 2, App. F.

In [21], the variability assumption is assumed to hold over *all* variables, which in our setting can be interpreted as a requirement over $2d$ fat-hand interventions over all latent variables simultaneously (plus one unintervened distribution). In this sense, Prop. 4.9 and *block-variability* extend the result of [21, Thm. 1], which only considers the case where *all* variables are intervened, by exploiting variability to achieve a strong identification only *within* a subset of the variables.

## 5 Experiments

Our experiments aim to estimate a CauCA model based on a known graph and a collection of interventional datasets with known targets. We focus on the scenarios with single-node, perfect interventions described in § 4. For additional technical details, see App. H.

**Synthetic data-generating process.** We first sample DAGs $G$ with an edge density of $0.5$. To model the causal dependence among the latent variables, we use the family of CBNs induced by linear Gaussian structural causal model (SCM) consistent with $G$.[5] For the ground-truth mixing function, we use $M$-layer multilayer perceptrons $\mathbf{f} = \sigma \circ \mathbf{A}_M \circ \ldots \circ \sigma \circ \mathbf{A}_1$, where $\mathbf{A}_m \in \mathbb{R}^{d \times d}$ for $m \in [\![1, M]\!]$ denote invertible linear maps (sampled from a multivariate uniform distribution), and $\sigma$ is an element-wise invertible nonlinear function. We then sample observed mixtures from these latent CBNs, as described by eq. (2).

**Likelihood-based estimation procedure.** Our objective is to learn an encoder $\mathbf{g}_{\boldsymbol{\theta}} : \mathbb{R}^d \to \mathbb{R}^d$ that approximates the inverse function $\mathbf{f}^{-1}$ up to tolerable ambiguities, together with latent densities $(p^k)_{k \in [\![0, d]\!]}$ reproducing the ground truth up to corresponding ambiguities (cf. Lemma 3.3). We

---

[5]Additional experiments with nonlinear, non-Gaussian CBNs can be found in App. I.

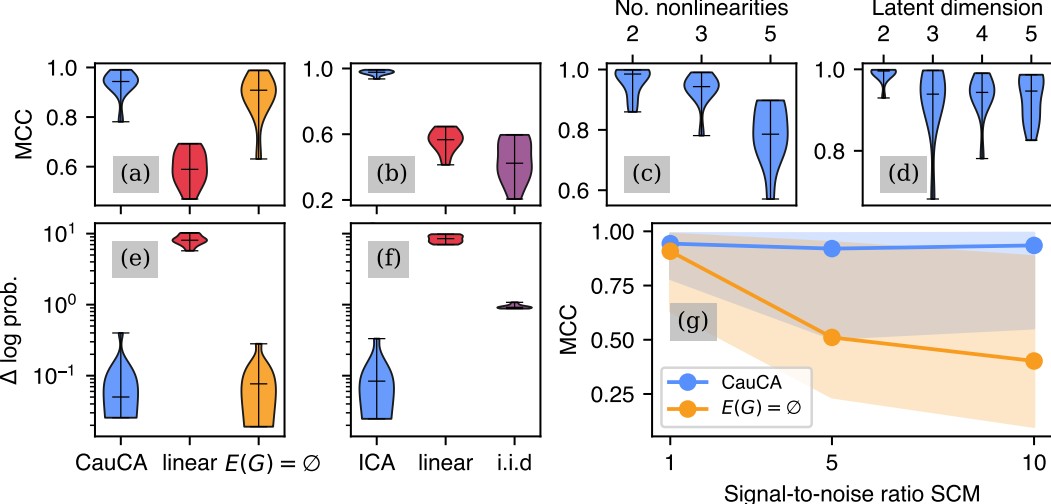

Figure 4: **Experimental results.** Figures (a) and (e) present the mean correlation coefficients (MCC) between true and learned latents and log-probability differences between the model and ground truth ($\Delta$ log prob.) for CauCA experiments. Misspecified models assuming a trivial graph ($E(G)=\varnothing$) and a linear encoder function class are compared. All violin plots show the distribution of outcomes for 10 pairs of CBNs and mixing functions. Figures (c) and (d) display CauCA results with varying numbers of nonlinearities in the mixing function and latent dimension. For the ICA setting, MCC values and log probability differences are illustrated in (b) and (f). Baselines include a misspecified model (linear mixing) and a naive (single-environment) unidentifiable normalizing flow with an independent Gaussian base distribution (labelled *i.i.d.*). The naive baseline is trained on pooled data without using information about interventions and their targets. Figure (g) shows the median MCC for CauCA and the misspecified baseline ($E(G)=\varnothing$) as the strength of the linear parameters relative to the exogenous noise in the structural causal model generating the CBN increases. The shaded areas show the range between minimum and maximum values.

estimate the encoder parameters by maximizing the likelihood, which can be derived through a change of variables from eq. (1): for an observation in dataset $k > 0$ taking a value $\mathbf{x}$, it is given by

$$\log p_{\boldsymbol{\theta}}^{k}(\mathbf{x}) = \log |\det \mathbf{J}\mathbf{g}_{\boldsymbol{\theta}}(\mathbf{x})| + \log \widetilde{p}_{\tau_k}((\mathbf{g}_{\boldsymbol{\theta}})_{\tau_k}(\mathbf{x})) + \sum_{i \neq \tau_k} \log p_i \left((\mathbf{g}_{\boldsymbol{\theta}})_i(\mathbf{x}) \mid (\mathbf{g}_{\boldsymbol{\theta}})_{\mathrm{pa}(i)}(\mathbf{x})\right), \quad (10)$$

where $\mathbf{J}\mathbf{g}_{\boldsymbol{\theta}}(\mathbf{x})$ denotes the Jacobian of $\mathbf{g}_{\boldsymbol{\theta}}$ evaluated at $\mathbf{x}$. The learning objective can be expressed as $\theta^* = \arg\max_{\boldsymbol{\theta}} \sum_{k=0}^{K} \left(\frac{1}{N_k} \sum_{n=1}^{N_k} \log p_{\boldsymbol{\theta}}^k(\mathbf{x}^{(n,k)})\right)$, with $N_k$ representing the size of dataset $k$.

**Model architecture.** We employ normalizing flows [40] to parameterize the encoder. Instead of the typically deployed base distribution with independent components, we use the collection of densities (one per interventional regime) induced by the CBN over the latents. Following the CauCA setting, the parameters of the causal mechanisms are learned while the causal graph is assumed known. For details on the model and training parameters, see App. H.

**Settings.** We investigate two learning problems: *(i)* CauCA, corresponding to § 4.1, and *(ii)* ICA, where the sampled graphs in the true latent CBN contain no arrows, as discussed in § 4.2.

**Results.** *(i)* For a data-generating process with non-empty graph, experimental outcomes are depicted in Fig. 4 (a, e). We compare a well-specified CauCA model (*blue*) to misspecified baselines, including a model with correctly specified latent model but employing a linear encoder (*red*), and a model with a nonlinear encoder but assuming a causal graph with no arrows (*orange*). See caption of Fig. 4 for details on the metrics. The results demonstrate that the CauCA model accurately identifies the latent variables, benefiting from both the nonlinear encoder and the explicit modelling of causal dependence. We additionally test the effect of increasing a parameter influencing the magnitude of the sampled linear parameters in the SCM (we refer to this as *signal-to-noise ratio*, see App. H.1 for details)— which increases the statistical *dependence* among the true latent components. The gap between the CauCA model and the baseline assuming a trivial graph widens (Fig. 4 (g)), indicating that *correctly*

*modelling the causal relationships becomes increasingly important the more (statistically) dependent the true latent variables are.* Finally, we verify that the model performs well for different number of layers $M$ in the ground-truth nonlinear mixing (c) (performance degrades slightly for higher $M$), and across various latent dimensionalities for the latent variable (d).

*(ii)* For data-generating processes where the graph contains no arrows (ICA), results are presented in Fig. 4 (b, f). Well-specified, nonlinear models *(blue)* are compared to misspecified linear baselines *(red)* and a naive normalizing flow baseline trained on pooled data *(purple)*. The findings confirm that *interventional information provides useful learning signal even in the context of nonlinear ICA*.

## 6    Related Work

**Causal representation learning.** In the present work, we focus on identifiability of *latent CBNs* with a *known graph*, based on *interventional data*, and investigate the *nonlinear and nonparametric* case. In CRL (*unknown graph*), many studies focus on identifiability of latent *SCMs* instead, which requires strong assumptions such as weak supervision (i.e., *counterfactual data*) [1, 3, 36, 54]. Alternatively, the setting where *temporal information* is available, i.e., dynamic Bayesian networks, has been studied extensively [29, 30, 33, 34, 62]. In a non-temporal setting, other works assume interventional data and *linear mixing functions* [49, 52]; or that the *latent distributions are linear Gaussian* [35]. Ahuja et al. [2] identify latent representations by *deterministic hard* interventions, together with *parametric* assumptions on the mixing, and an *independent support* assumption [56]. Concurrent work studies the cases with non-parametric mixing and linear Gaussian latent causal mode [5] or non-parametric latent causal model under faithfulness, *genericity* and Asm. 4.1 [55].

**Prior knowledge on the latent SCM.** Other prior works also leverage prior knowledge on the causal structure for representation learning. Yang et al. [61] introduce the CausalVAE model, which aims to disentangle the endogenous and exogenous variables of an SCM, and prove identifiability up to affine transformations based on known intervention targets. Shen et al. [46] also consider the setting in which the graph is (partially) known, but their approach requires additional supervision in the form of annotations of the ground truth latent. Leeb et al. [31] embed an SCM into the latent space of an autoencoder, provided with a topological ordering allowing it to learn latent DAGs.

**Statistically dependent components.** Models with causal dependences among the latent variables are a special case of models where the latent variables are statistically dependent [22]. Various extensions of the ICA setting allow for dependent variables: e.g., independent subspace analysis [16]; topographic ICA [20] (see also [24]); independently modulated component analysis [26]. Morioka and Hyvärinen [39] introduce a multi-modal model where *within-modality dependence* is described by a Bayesian network, with *joint independence across the modalities*, and a mixing function for same-index variables across these networks. Unlike our work, it encodes no explicit notions of interventions.

## 7    Discussion

**Limitations. (i) Known intervention targets:** We proved that with *fully unknown targets*, there are fundamental and strong limits to identifiability (see Corollary E.8). We also studied some relaxations of this assumption (App. E), and generalized our results to *known targets up to graph automorphisms* and *matched intervention targets* (see Prop. E.6 and Prop. E.6). Other relaxations are left for future work; e.g., the case with a non-trivial graph and matched intervention targets is studied in [55], under *faithfulness* and *genericity* assumptions. **(ii) Estimation:** More scalable estimation procedures than our likelihood-based approach (§ 5) may be developed, e.g., based on variational inference.

**CauCA as a causal generalization of ICA.** As pointed out in § 4.2, the special case of CauCA with a trivial graph corresponds to a novel ICA model. Beyond the fact that CauCA allows statistical dependence described by general DAGs among the components, we argue that it can be viewed as a *causal* generalization of ICA. Firstly, we exploit the assumption of *localized and sparse* changes in the latent mechanisms [42, 45], in contrast to previous ICA works which exploit *non-stationarity* at the level of the entire joint distribution of the latent components [17, 21, 38], leading to strong identifiability results (e.g., in Thm. 4.2 *(ii)*). Secondly, we exploit the modularity of causal mechanisms: i.e., it is possible to intervene on some of the mechanisms while leaving the others *invariant* [41, 43]. To the best of our knowledge, our work is the first ICA extension where latent dependence can actually be interpreted in a causal sense.

## Acknowledgements

The authors thank Vincent Stimper, Weiyang Liu, Siyuan Guo, Junhyung Park, Jinglin Wang, Corentin Correia, Cian Eastwood, Adrián Javaloy and the anonymous reviewers for helpful comments and discussions.

## Funding Transparency Statement

This work was supported by the Tübingen AI Center. L.G. was supported by the VideoPredict project, FKZ: 01IS21088.

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

# APPENDIX

## Overview

- App. A recapitulates the notation used in this paper.
- App. B contains the proofs of all theoretical statements presented in the paper.
- App. C contains a nontrivial counterexample of Thm. 4.2 *(ii)* when Asm. 4.1 is violated.
- App. D contains an additional result of identifiability based on Thm. 4.2 *(i)*, when more imperfect stochastic interventions are available.
- App. E contains a general discussion on CauCA with unknown intervention targets, as well as a generalization of some of the identifiability results.
- App. F contains a technical details on the relationship between CauCA and nonlinear ICA.
- App. G contains some theoretical results which were useful in the design of the experiments.
- App. H contains the details of the experiments.

# A  Notations

| Symbol | Description |
|---|---|
| $G$ | A directed acyclic graph with nodes $V(G) = [d]$ and arrows $E(G)$ |
| $(i, j)$ | An ordered tuple representing an arrow in $E(G)$, with $i, j \in V(G)$ |
| $[\![i, j]\!]$ | The integers $i, \dots, j$ |
| $[d]$ | The natural numbers $1, \dots, d$ |
| $\mathrm{pa}(i)$ | Parents of $i$, defined as $\{j \in V(G) \mid (j, i) \in E(G)\}$ |
| $\mathrm{pa}^k(j)$ | Parents of $j$ in the post-intervention graph in the intervention regime $k$ |
| $\overline{\mathrm{pa}}(i)$ | Closure of the parents of $i$, defined as $\mathrm{pa}(i) \cup \{i\}$ |
| $\mathrm{anc}(i)$ | Ancestors of $i$, nodes $j$ in $G$ such that there is a directed path from $j$ to $i$ |
| $\overline{\mathrm{anc}}(i)$ | Closure of the ancestors of $i$, defined as $\mathrm{anc}(i) \cup \{i\}$ |
| $\overline{G}$ | Transitive closure of $G$ defined by $\mathrm{pa}^{\overline{G}}(i) := \mathrm{anc}^G(i)$ |
| $X, Y, Z$ | Unidimensional random variables |
| $\mathbf{X}, \mathbf{Y}, \mathbf{Z}$ | Multidimensional random variables |
| $x, y, z$ | Scalars in $\mathbb{R}$ |
| $\mathbf{x}, \mathbf{y}, \mathbf{z}$ | Vectors in $\mathbb{R}^d$ |
| $\mathbf{z}_{[i]}$ | The $(1, \dots, i)$ dimensions of $\mathbf{z}$ |
| $\varphi_i$ | The function that outputs the $i$-th dimension of the mapping $\varphi$ |
| $\boldsymbol{\varphi}_{[i]}$ | The mapping that outputs the $(1, \dots, i)$ dimensions of the mapping $\boldsymbol{\varphi}$ |
| $\tau_k$ | Intervention targets in interventional regime $k$ |
| $\mathbb{P}, \mathbb{Q}$ | Probability distributions |
| $p, q$ | Density functions of $\mathbb{P}, \mathbb{Q}$ |
| $\mathbb{P}_i\left(Z_i \mid \mathbf{Z}_{\mathrm{pa}(i)}\right)$ | Causal mechanism of variable $Z_i$ |
| $\widetilde{\mathbb{P}}_i^k\left(Z_i \mid \mathbf{Z}_{\mathrm{pa}^k(i)}\right)$ | Intervened mechanism of variable $Z_i$ in interventional regime $k$ |
| $\mathbb{P}^k(\mathbf{Z})$ | $k \neq 0$: interventional distribution in interventional regime $k$ (Defn. 2.2) |
| | $k = 0$: unintervened distribution |
| $\mathcal{P}_G$ | Class of latent joint probabilities that are Markov relative to $G$ |
| | In this paper, it is assumed to have differentiable density and to be |
| | absolutely continuous with full support in $\mathbb{R}^d$ |
| $\mathcal{F}$ | Function class of mixing function/decoders |
| | In this paper, it is assumed to be all $\mathcal{C}^1$-diffeomorphisms |
| $\mathcal{S}$ | Indeterminacy set, defined in Defn. 3.2 |
| $\mathbf{f}$ | Mixing function or decoder, a diffeomorphism $\mathbb{R}^d \to \mathbb{R}^d$ |
| $\mathbf{f}_*\mathbb{P}$ | The pushforward measure of $\mathbb{P}$ by $\mathbf{f}$ |
| $\mathsf{G}$ | A directed acyclic graph without indices |
| $G \models \mathsf{G}$ | $G$ is an indexed graph of $\mathsf{G}$, i.e. $V(G) = [d]$ and there exists an |
| | isomorphism $G \to \mathsf{G}$ |
| $\mathrm{Aut}_G$ | Group of automorphisms of graph $G$ |
| $\mathfrak{S}_d$ | Group of permutations of $d$ elements |

# B  Proofs

## B.1  Lemmata

**Lemma B.1** (Lemma 2 of [3]). *Let $A = C = \mathbb{R}$ and $B = \mathbb{R}^d$. Let $f : A \times B \to C$ be differentiable. Define differentiable measures $\mathbb{P}_A$ on $A$ and $\mathbb{P}_C$ on $C$. Let $\forall b \in B$, $f(\cdot, b) : A \to C$ be measure-preserving, i.e. $\mathbb{P}_C = f(\cdot, b)_*\mathbb{P}_A$. Then $f$ is constant in $b$ over $B$.*

**Lemma B.2.** *For any distributions $\mathbb{P}$, $\mathbb{Q}$ of full support on $\mathbb{R}$, with c.d.f $F$, $G$, there are only two diffeomorphisms $T : \mathbb{R} \to \mathbb{R}$ such that $T_*\mathbb{P} = \mathbb{Q}$: they are $G^{-1} \circ F$ and $\overline{G}^{-1} \circ F$, where $\overline{G}(x) := 1 - G(x)$.*

*Proof.* $T$ is a diffeomorphism, then $T'(x) \neq 0 \quad \forall x \in \mathbb{R}$. Then the sign of $T'(x)$ is either positive or negative everywhere. $T$ is either strictly increasing or strictly decreasing on $\mathbb{R}$. Since $\mathbb{P}$ and $\mathbb{Q}$ are full support in $\mathbb{R}$, $F$ and $G$ are strictly increasing in $\mathbb{R}$.

- If $T$ is increasing: $T_*\mathbb{P} = \mathbb{Q}$ implies that $G(x) = \mathbb{Q}(X \le x) = \mathbb{P}(T(X) \le x)$. Since $T$ is strictly increasing, $G(x) = \mathbb{P}(T(X) \le x) = \mathbb{P}(X \le T^{-1}(x)) = F \circ T^{-1}(x)$. Thus $x = G^{-1} \circ F \circ T^{-1}(x)$. $T = G^{-1} \circ F$.

- If $T$ is decreasing: $G(x) = \mathbb{Q}(X \le x) = \mathbb{P}(T(X) \le x) = \mathbb{P}(X \ge T^{-1}(x)) = 1 - F(T^{-1}(x))$. Thus $T = \overline{G}^{-1} \circ F$.

$\square$

**Lemma B.3.** *Suppose $\mathbb{P}, \mathbb{Q}$ Markov relative to $G$, absolutely continuous with full support in $\mathbb{R}^d$. Fix any functions $\varphi_1, \ldots, \varphi_d$ diffeomorphisms strictly monotonic in $\mathbb{R}$. Let $\boldsymbol{\varphi} := (\varphi_i)_{i \in [d]}$. The following statements are equivalent:*

*(1) $\mathbb{Q} = \boldsymbol{\varphi}_*\mathbb{P}$*

*(2) $\forall i \in [d]$, $\forall z_i \in \mathbb{R}$, $\forall \mathbf{z}_{pa(i)} \in \mathbb{R}^{|pa(i)|}$, $p_i(z_i \mid \mathbf{z}_{pa(i)}) = q_i(\varphi_i(z_i) \mid \boldsymbol{\varphi}_{pa(i)}(\mathbf{z}_{pa(i)})) |\varphi_i'(z_i)|$, or equivalently, $\mathbb{Q}_i(\cdot|\boldsymbol{\varphi}_{pa(i)}(\mathbf{z}_{pa(i)})) = (\varphi_i)_*\mathbb{P}_i(\cdot \mid \mathbf{z}_{pa(i)})$.*

*Proof.* $\mathbb{Q}_i(\cdot|\boldsymbol{\varphi}_{\mathrm{pa}(i)}(\mathbf{z}_{\mathrm{pa}(i)}))$ denotes the conditional probability of $Z_i$: $\mathbb{Q}_i(Z_i|\boldsymbol{\varphi}_{\mathrm{pa}(i)}(\mathbf{z}_{\mathrm{pa}(i)}))$.

$(2) \Rightarrow (1)$: Multiply the equations in (2) for $n$ indices,

$$\prod_{i=1}^{d} p_i(z_i \mid \mathbf{z}_{\mathrm{pa}(i)}) = \prod_{i=1}^{d} q_i(\varphi_i(z_i) \mid \boldsymbol{\varphi}_{\mathrm{pa}(i)}(\mathbf{z}_{\mathrm{pa}(i)})) |\varphi_i'(z_i)|.$$

Since $\prod_{i=1}^{d} |\varphi_i'(z_i)| = |\det D\boldsymbol{\varphi}(\mathbf{z})|$, we obtain the equation in (1).

$(1) \Rightarrow (2)$: without loss of generality, choose a total order on $V(G)$ that preserves the partial order of $G$: $i > j$ if $i \in \mathrm{pa}(j)$. Since $\boldsymbol{\varphi}$ is a diffeomorphism, by the change of variables formula,

$$p(\mathbf{z}) = q(\boldsymbol{\varphi}(\mathbf{z}))|\det D\boldsymbol{\varphi}(\mathbf{z})|. \tag{11}$$

Since $\mathbb{P}, \mathbb{Q}$ are Markov relative to $G$, write $p$, $q$ as the factorization according to $G$:

$$\prod_{i=1}^{d} p_i(z_i \mid \mathbf{z}_{\mathrm{pa}(i)}) = \prod_{i=1}^{d} q_i(\varphi_i(z_i) \mid \boldsymbol{\varphi}_{\mathrm{pa}(i)}(\mathbf{z}_{\mathrm{pa}(i)})) |\varphi_i'(z_i)|. \tag{12}$$

We will show by induction on the reverse order of $[d]$ that for all $i \in [d]$,

$$p_i(z_i \mid \mathbf{z}_{\mathrm{pa}(i)}) = q_i(\varphi_i(z_i) \mid \boldsymbol{\varphi}_{\mathrm{pa}(i)}(\mathbf{z}_{\mathrm{pa}(i)})) |\varphi_i'(z_i)|.$$

In eq. (12), marginalize over $z_d$,

$$\prod_{i=1}^{n-1} p_i(z_i \mid \mathbf{z}_{\mathrm{pa}(i)}) = \int_{\mathbb{R}} \prod_{i=1}^{d} q_i(\varphi_i(z_i) \mid \boldsymbol{\varphi}_{\mathrm{pa}(i)}(\mathbf{z}_{\mathrm{pa}(i)})) |\varphi_i'(z_i)| \, dz_d. \tag{13}$$

$$\tag{14}$$

Fix $z_{[d-1]}$, change of variable $u = \varphi_d(z_d)$, $du = \varphi_d'(z_d)$,

$$\prod_{i=1}^{d-1} p_i(z_i \mid \mathbf{z}_{\mathrm{pa}(i)}) = \prod_{i=1}^{d-1} q_i(\varphi_i(z_i) \mid \boldsymbol{\varphi}_{\mathrm{pa}(i)}(\mathbf{z}_{\mathrm{pa}(i)})) |\varphi_i'(z_i)|. \tag{15}$$

Cancel the two sides of equation (12) by (15),

$$p_i(z_d \mid \mathbf{z}_{\mathrm{pa}(d)}) = q_i(\varphi_d(z_d) \mid \boldsymbol{\varphi}_{\mathrm{pa}(d)}(\mathbf{z}_{\mathrm{pa}(d)})) |\varphi_d'(z_d)|.$$

Suppose the property is true for $i + 1, \cdots, d$. Then for $i$, $i$ is a leaf node in the first $i$ nodes. We use the same proof as before, marginalize over $z_i$ (same as (13) with $d$ replaced by $i$) on the joint distribution of $i$ first variables (same as (12) with $d$ replaced by $i$), which is then divided by the obtained $i-1$ marginal equation (same as (15) with $d-1$ replaced by $i-1$). $\square$

**Lemma 3.3.** *For any* $(G, \mathbf{f}, (\mathbb{P}^k, \tau_k)_{k \in [\![0,K]\!]})$ *in* $(G, \mathcal{F}, \mathcal{P}_G)$, *and for any* $\mathbf{h} \in \mathcal{S}_{scaling}$ *with*

$$\mathcal{S}_{scaling} := \left\{ \mathbf{h} : \mathbb{R}^d \to \mathbb{R}^d \mid \mathbf{h}(\mathbf{z}) = (h_1(z_1), \ldots, h_d(z_d)), \, h_i \text{ is a diffeomorphism in } \mathbb{R} \right\}, \qquad (3)$$

*there exists a* $(G, \mathbf{f} \circ \mathbf{h}, (\mathbb{Q}^k, \tau_k)_{k \in [\![0,K]\!]})$ *in* $(G, \mathcal{F}, \mathcal{P}_G)$ *s.t.* $\mathbf{f}_* \mathbb{P}^k = (\mathbf{f} \circ \mathbf{h})_* \mathbb{Q}^k$ *for all* $k \in [\![0, K]\!]$.

*Proof.* For any $(G, \mathbf{f}, (\mathbb{P}^k, \tau_k)_{k \in [\![0,K]\!]})$ in $(G, \mathcal{F}, \mathcal{P}_G)$, and for any $\mathbf{h} \in \mathcal{S}_{\text{scaling}}$, define $\mathbf{g} := \mathbf{h}^{-1}$, then $\mathbf{g} \in \mathcal{S}_{\text{scaling}}$. Define $\mathbb{Q}^0 := \mathbf{g}_* \mathbb{P}^0$. By Lemma B.3, for all $i \in [d]$, $\mathbb{Q}_i \left( \cdot | \mathbf{h}_{\text{pa}(i)}(\mathbf{z}_{\text{pa}(i)}) \right) = (g_i)_* \mathbb{P}_i \left( \cdot \mid \mathbf{z}_{\text{pa}(i)} \right)$.

For $k \in [K]$, define

$$\widetilde{\mathbb{Q}}_j^k \left( \cdot | \mathbf{g}_{\text{pa}^k(j)}(\mathbf{z}_{\text{pa}^k(j)}) \right) := (g_j)_* \widetilde{\mathbb{P}}_j^k \left( \cdot \mid \mathbf{z}_{\text{pa}^k(j)} \right). \qquad (16)$$

Define $\mathbb{Q}^k := \prod_{j \in \tau_k} \widetilde{\mathbb{Q}}_j^k \prod_{j \notin \tau_k} \mathbb{Q}_j$, by Lemma B.3 and (16), $\mathbb{Q}^k = \mathbf{g}_* \mathbb{P}^k \, \forall k \in [K]$. By definition of $\mathbb{Q}^0$, $\mathbb{Q}^k = \mathbf{g}_* \mathbb{P}^k \, \forall k \in [\![0, K]\!]$. i.e., $\mathbf{f}_* \mathbb{P}^k = (\mathbf{f} \circ \mathbf{h})_* \mathbb{Q}^k$.

$\square$

## B.2 Proof of Thm. 4.2

**Theorem 4.2.** *For CauCA in* $(G, \mathcal{F}, \mathcal{P}_G)$,

*(i) Suppose for each node in* $[d]$, *there is one (perfect or imperfect) stochastic intervention that satisfies Asm. 4.1. Then CauCA in* $(G, \mathcal{F}, \mathcal{P}_G)$ *is identifiable up to*

$$\mathcal{S}_{\overline{G}} = \left\{ \mathbf{h} : \mathbb{R}^d \to \mathbb{R}^d | \mathbf{h}(\mathbf{z}) = \left( h_i(\mathbf{z}_{\overline{anc}(i)}) \right)_{i \in [d]}, \mathbf{h} \text{ is } \mathcal{C}^1\text{-diffeomorphism} \right\}. \qquad (5)$$

*(ii) Suppose for each node* $i$ *in* $[d]$, *there is one perfect stochastic intervention that satisfies Asm. 4.1. Then CauCA in* $(G, \mathcal{F}, \mathcal{P}_G)$ *is identifiable up to* $\mathcal{S}_{scaling}$.

*Proof.* **Proof of (i):** Consider two latent CBNs achieving the same likelihood across all interventional regimes: $\left( G, \mathbf{f}, (\mathbb{P}^i, \tau_i)_{i \in [\![0,d]\!]} \right)$ and $\left( G, \mathbf{f}', (\mathbb{Q}^i, \tau_i)_{i \in [\![0,d]\!]} \right)$. Since the intervention targets $(\tau_i)_{i \in [\![0,d]\!]}$ are the same on both latent CBN, by rearranging the indices of $G$ and correspondingly the indices in $\mathbb{P}^i$, $\mathbb{Q}^i$ and $(\tau_i)_{i \in [\![0,d]\!]}$, we can suppose without loss of generality that the index of $G$ preserves the partial order induced by $E(G)$: $i < j$ if $(i, j) \in E(G)$. Since $(\tau_i)_{i \in [d]}$ covers all $d$ nodes in $G$, by rearranging $(\tau_i)_{i \in [d]}$ we can suppose without loss of generality that $\tau_i = i \, \forall i \in [d]$.

In the $i$-th interventional regime,

$$\mathbb{P}^i(\mathbf{Z}) = \widetilde{\mathbb{P}}_i \left( Z_i \mid \mathbf{Z}_{\text{pa}^i(i)} \right) \prod_{j \in [d] \setminus i} \mathbb{P} \left( Z_j \mid \mathbf{Z}_{\text{pa}(j)} \right),$$

$$\mathbb{Q}^i(\mathbf{Z}) = \widetilde{\mathbb{Q}}_i \left( Z_i \mid \mathbf{Z}_{\text{pa}^i(i)} \right) \prod_{j \in [d] \setminus i} \mathbb{Q}_j \left( Z_j \mid \mathbf{Z}_{\text{pa}(j)} \right),$$

where $\text{pa}(j) = \text{pa}^i(j) \, \forall j \neq i$, since intervening on $i$ does not change the arrows towards $j$.

Define $\boldsymbol{\varphi} := \mathbf{f}'^{-1} \circ \mathbf{f}$. Denote its $i$-th dimension output function as $\varphi_i : \mathbb{R}^d \to \mathbb{R}$. We will prove by induction that $\forall i \in [d], \forall j \notin \overline{\text{anc}}(i), \forall \mathbf{z} \in \mathbb{R}^d, \frac{\partial \varphi_i}{\partial z_j}(\mathbf{z}) = 0$.

For any $i \in [\![0, d]\!]$, $\mathbf{f}_* \mathbb{P}^i = \mathbf{f}'_* \mathbb{Q}^i$. Since $\boldsymbol{\varphi}$ is a diffeomorphism, by the change of variable formula,

$$p^i(\mathbf{z}) = q^i(\boldsymbol{\varphi}(\mathbf{z})) |\det D\boldsymbol{\varphi}(\mathbf{z})|. \qquad (17)$$

For $i = 0$, factorize $p^i$ and $q^i$ according to $G$, then take the logarithm on both sides:

$$\sum_{j=1}^d \ln p_j \left( z_j \mid \mathbf{z}_{\text{pa}(j)} \right) = \sum_{j=1}^d \ln q_j \left( \varphi_j(\mathbf{z}) \mid \boldsymbol{\varphi}_{\text{pa}(j)}(\mathbf{z}) \right) + \ln |\det D\boldsymbol{\varphi}(\mathbf{z})|. \qquad (18)$$

For $i = 1$, $\widetilde{\mathbb{Q}}_1$ has no conditionals, thus $q^i$ is factorized as

$$q^1(\mathbf{z}) = \widetilde{q}_1(z_1) \prod_{j=2}^{d} q_j\left(z_j \mid \mathbf{z}_{\mathrm{pa}(j)}\right).$$

So the equation (17) for $i = 1$ after taking logarithm is

$$\ln \widetilde{p}_1(z_1) + \sum_{j=2}^{d} \ln p_j\left(z_j \mid \mathbf{z}_{\mathrm{pa}(j)}\right) = \ln \widetilde{q}_1(\varphi_1(\mathbf{z})) + \sum_{j=2}^{d} q_j\left(\varphi_j(\mathbf{z}) \mid \boldsymbol{\varphi}_{\mathrm{pa}(j)}(\mathbf{z})\right) \tag{19}$$
$$+ \ln |\det D\boldsymbol{\varphi}(\mathbf{z})|.$$

Subtract (19) by (18),

$$\ln \tilde{p}_1(z_1) - \ln p_1(z_1) = \ln \tilde{q}_1(\varphi_1(\mathbf{z})) - \ln q_1(\varphi_1(\mathbf{z})). \tag{20}$$

For any $i \neq 1$, take the $i$-th partial derivative of both sides:

$$0 = \left[ \frac{\tilde{q}_1'(\varphi_1(\mathbf{z}))}{\tilde{q}_1(\varphi_1(\mathbf{z}))} - \frac{q_1'(\varphi_1(\mathbf{z}))}{q_1(\varphi_1(\mathbf{z}))} \right] \frac{\partial \varphi_1}{\partial z_i}(\mathbf{z}).$$

By Asm. 4.1, the term in the parenthesis is non-zero a.e. in $\mathbb{R}^d$. Thus $\frac{\partial \varphi_1}{\partial z_i}(\mathbf{z}) = 0$ a.e. in $\mathbb{R}^d$. Since $\boldsymbol{\varphi} = \mathbf{f}'^{-1} \circ \mathbf{f}$ where $\mathbf{f}, \mathbf{f}'$ are $\mathcal{C}^1$-diffeomorphisms, so is $\boldsymbol{\varphi}$. $\frac{\partial \varphi_1}{\partial z_i}$ is continuous and thus equals zero everywhere.

Now suppose $\forall k \in [i-1], \forall j \notin \overline{\mathrm{anc}}(k), \forall \mathbf{z} \in \mathbb{R}^d, \frac{\partial \varphi_k}{\partial z_j}(\mathbf{z}) = 0$. Then for interventional regime $i$,

$$\ln \tilde{p}_i\left(z_i \mid \mathbf{z}_{\mathrm{pa}^i(i)}\right) + \sum_{j \neq i} \ln p_j\left(z_j \mid \mathbf{z}_{\mathrm{pa}(j)}\right) = \ln \tilde{q}_i\left(\varphi_i(\mathbf{z}) \mid \boldsymbol{\varphi}_{\mathrm{pa}^i(i)}(\mathbf{z})\right)$$
$$+ \sum_{j \neq i} q_j\left(\varphi_j(\mathbf{z}) \mid \boldsymbol{\varphi}_{\mathrm{pa}(j)}(\mathbf{z})\right) + \ln |\det D\boldsymbol{\varphi}(\mathbf{z})|,$$

Subtracted by (18),

$$\ln \tilde{p}_i\left(z_i \mid \mathbf{z}_{\mathrm{pa}^i(i)}\right) - \ln p_i\left(z_i \mid \mathbf{z}_{\mathrm{pa}(i)}\right) = \ln \tilde{q}_i\left(\varphi_i(\mathbf{z}) \mid \boldsymbol{\varphi}_{\mathrm{pa}^i(i)}(\mathbf{z})\right) - \ln q_i\left(\varphi_i(\mathbf{z}) \mid \boldsymbol{\varphi}_{\mathrm{pa}(i)}(\mathbf{z})\right). \tag{21}$$

For any $\forall j \notin \overline{\mathrm{anc}}(i), j \notin \mathrm{pa}(i) \supset \mathrm{pa}^i(i)$ by assumption. Take partial derivative over $z_j$:

$$0 = \frac{\sum_{k \in \overline{\mathrm{pa}}^i(i)} \frac{\partial \tilde{q}_i}{\partial x_k}\left(\varphi_i(\mathbf{z}) \mid \boldsymbol{\varphi}_{\mathrm{pa}^i(i)}(\mathbf{z})\right) \frac{\partial \varphi_k}{\partial z_j}(\mathbf{z})}{\tilde{q}_i\left(\varphi_i(\mathbf{z}) \mid \boldsymbol{\varphi}_{\mathrm{pa}^i(i)}(\mathbf{z})\right)} - \frac{\sum_{k \in \overline{\mathrm{pa}}(i)} \frac{\partial q_i}{\partial x_k}\left(\varphi_i(\mathbf{z}) \mid \boldsymbol{\varphi}_{\mathrm{pa}(i)}(\mathbf{z})\right) \frac{\partial \varphi_k}{\partial z_j}(\mathbf{z})}{q_i\left(\varphi_i(\mathbf{z}) \mid \boldsymbol{\varphi}_{\mathrm{pa}(i)}(\mathbf{z})\right)},$$

where $x_k$ denotes the $k$-th dimension of the domain of $\tilde{q}_i$ and $q_i$.

For all $k \in \mathrm{pa}(i)$, since $j \notin \overline{\mathrm{anc}}(i)$, $j$ is not in $\overline{\mathrm{anc}}(k)$ either. By the assumption of induction, $\frac{\partial \varphi_k}{\partial z_j}(\mathbf{z}) = 0$. Delete the partial derivatives that are zero:

$$0 = \left[ \frac{\frac{\partial \tilde{q}_i}{\partial x_i}\left(\varphi_i(\mathbf{z}) \mid \boldsymbol{\varphi}_{\mathrm{pa}^i(i)}(\mathbf{z})\right)}{\tilde{q}_i\left(\varphi_i(\mathbf{z}) \mid \boldsymbol{\varphi}_{\mathrm{pa}^i(i)}(\mathbf{z})\right)} - \frac{\frac{\partial q_i}{\partial x_i}\left(\varphi_i(\mathbf{z}) \mid \boldsymbol{\varphi}_{\mathrm{pa}(i)}(\mathbf{z})\right)}{q_i\left(\varphi_i(\mathbf{z}) \mid \boldsymbol{\varphi}_{\mathrm{pa}(i)}(\mathbf{z})\right)} \right] \frac{\partial \varphi_i}{\partial z_j}(\mathbf{z}).$$

By Asm. 4.1, the term in parenthesis is nonzero a.e., thus $\frac{\partial \varphi_i}{\partial z_j}(\mathbf{z}) = 0$ a.e. Since $\frac{\partial \varphi_i}{\partial z_j}$ is continuous, it equals zero everywhere.

The induction is finished when $i = d$. We have proven that $\forall i \in [d], \forall j \notin \overline{\mathrm{anc}}(i), \forall \mathbf{z} \in \mathbb{R}^d$, $\frac{\partial \varphi_i}{\partial z_j}(\mathbf{z}) = 0$. Namely, $\varphi_i$ only depends on $\mathbf{z}_{\overline{\mathrm{anc}}(i)}$.

**Proof of *(ii)*:**

By the result proved in *(i)*, $\varphi := \mathbf{f}'^{-1} \circ \mathbf{f} \in \mathcal{S}_{\overline{G}}$. Thus $D\varphi(\mathbf{z})$ is lower triangular for all $\mathbf{z} \in \mathbb{R}^d$. Thus $|\det D\varphi(\mathbf{z})| = \prod_{i=1}^{d} \left| \frac{\partial \varphi_i}{\partial z_i} \left( z_{[i]} \right) \right|$, and for all $i$, $\varphi_i$ only depends on $z_1, \cdots, z_i$. We will prove that $\varphi_i$ only depends on $z_i$, i.e., it is constant on other variables.

To prove the conclusion in this item, we need the following lemma:

**Lemma B.4.** *Given any $\varphi : \mathbb{R}^n \to \mathbb{R}^n$ diffeomorphism such that for all $i \in [n-1]$, $\frac{\partial \varphi_i}{\partial z_n}$ is zero everywhere, and given any two distributions $\mathbb{P}$, $\mathbb{Q}$ that are absolutely continuous and have full support in $\mathbb{R}^n$ such that $\varphi_* \mathbb{P} = \mathbb{Q}$, then the distributions of the first $n-1$ coordinates are preserved, i.e.,*

$$(\varphi_{[n-1]})_* \mathbb{P}_{[n-1]}(\mathbf{z}_{[n-1]}) = \mathbb{Q}_{[n-1]}(\mathbf{z}_{[n-1]}).$$

*Proof.* Fix $\mathbf{z}_{[n-1]} \in \mathbb{R}^{n-1}$. For all $i \in [n-1]$, $\frac{\partial \varphi_i}{\partial z_n}$ is zero everywhere, and $\varphi$ is a diffeomorphism, so $\frac{\partial \varphi_n}{\partial z_n}$ is nonzero everywhere, otherwise there will exist $\mathbf{z}$ such that $\frac{\partial \varphi_n}{\partial z_n}(\mathbf{z})$ is singular. Therefore $\frac{\partial \varphi_n}{\partial z_n}(\mathbf{z}_{[n-1]}, \cdot)$ is continuous and nonzero, and thus $\varphi_n(\mathbf{z}_{[n-1]}, \cdot)$ is a diffeomorphism $\mathbb{R} \to \mathbb{R}$. So we can apply the change of variable $u = \varphi_n(\mathbf{z}_{[n-1]}, z_n)$, $du = \left| \frac{\partial \varphi_n}{\partial z_n}(\mathbf{z}_{[n-1]}, z_n) \right| dz_n$:

$$\int_{\mathbb{R}} q\left( \varphi_{[n-1]}\left( \mathbf{z}_{[n-1]} \right), \varphi_n\left( z_n \right) \right) \left| \frac{\partial \varphi_n}{\partial z_n}\left( \mathbf{z}_{[n-1]}, z_n \right) \right| dz_n = \int_{\mathbb{R}} q\left( \varphi_{[n-1]}\left( \mathbf{z}_{[n-1]} \right), u \right) du \tag{22}$$
$$= q_{[n-1]}\left( \varphi_{[n-1]}\left( \mathbf{z}_{[n-1]} \right) \right).$$

In the equation $p(\mathbf{z}) = q(\varphi(\mathbf{z})) |\det \varphi(\mathbf{z})|$, marginalize over $z_n$:

$$\int_{\mathbb{R}} p\left( \mathbf{z}_{[n-1]}, z_n \right) dz_n = \int_{\mathbb{R}} q\left( \varphi_{[n-1]}\left( \mathbf{z}_{[n-1]} \right), \varphi_n\left( z_n \right) \right) \prod_{i=1}^{n} \left| \frac{\partial \varphi_i}{\partial z_i}\left( z_{[i]} \right) \right| dz_n. \tag{23}$$

Using (22), we obtain

$$p_{[n-1]}\left( \mathbf{z}_{[n-1]} \right) = q_{[n-1]}\left( \varphi_{[n-1]}\left( \mathbf{z}_{[n-1]} \right) \right) \prod_{i=1}^{n-1} \left| \frac{\partial \varphi_i}{\partial z_i}\left( z_{[i]} \right) \right|$$
$$= q_{[n-1]}\left( \varphi_{[n-1]}\left( \mathbf{z}_{[n-1]} \right) \right) |\det D\varphi_{[n-1]}\left( \mathbf{z}_{[n-1]} \right)|,$$

which is the density equation for push-forward measures that we want to prove. $\qquad \square$

Back to the proof of *(ii)*. For all $i \in [d]$, $(\varphi_1, \ldots, \varphi_{i-1})$ is a diffeomorphism because $|\det D\varphi_{[i-1]}(\mathbf{z}_{[i-1]})| = \prod_{j=1}^{i-1} \left| \frac{\partial \varphi_j}{\partial z_j}(\mathbf{z}) \right| \neq 0$.

We will prove by induction on the reverse order of $[d]$ that the i-th row off-diagonal entries of $D\varphi(\mathbf{z})$ are zero for all $\mathbf{z} \in \mathbb{R}^d$.

In the interventional regime $d$, by the assumption on the indices of $V(G) = [d]$ in the proof *(i)*, the node $d$ is not a parent of any node in $[d-1]$. Thus the perfect stochastic intervention on $z_d$ leads to the density $p^d$ and $q^d$ factorized as follows:

$$p_{[d-1]}\left( \mathbf{z}_{[d-1]} \right) \tilde{p}_d\left( z_d \right) = q_{[d-1]}\left( \varphi_{[d-1]}\left( \mathbf{z}_{[d-1]} \right) \right) \tilde{q}_d\left( \varphi_d(\mathbf{z}) \right) |\det D\varphi_{[d-1]}\left( \mathbf{z}_{[d-1]} \right)| \left| \frac{\partial \varphi_d}{\partial z_d}(\mathbf{z}) \right|.$$

Since $D\varphi$ is lower triangular everywhere, cancel the terms of coordinate $[d-1]$ on both sides by Lemma B.4,

$$\tilde{p}_d\left( z_d \right) = \tilde{q}_d\left( \varphi_d\left( \mathbf{z}_{[d-1]}, z_d \right) \right) \left| \frac{\partial \varphi_d}{\partial z_d}\left( \mathbf{z}_{[d-1]}, z_d \right) \right|, \tag{24}$$

which is equivalent to

$$\forall \mathbf{z}_{[d-1]} \in \mathbb{R}^{d-1}, \quad \widetilde{\mathbb{Q}}_d = \varphi_d \left( \mathbf{z}_{[d-1]}, \cdot \right)_* \widetilde{\mathbb{P}}_d. \tag{25}$$

By Lemma B.1, $\varphi_d$ is constant in the first $d-1$ variables.

Suppose the off-diagonal entries are zero for the $i, i+1, \ldots, d$ rows of $D\boldsymbol{\varphi}(\mathbf{z})$.

By the assumption on the indices of $V(G)$ in the proof *(i)*, the node $i$ is not a parent of any node in $[i-1]$. Thus the perfect stochastic intervention on $z_i$ leads to the density $p^i$ and $q^i$ factorized as follows:

$$\int_{\mathbb{R}^{d-i}} p\left(\mathbf{z}\right) dz_{i+1} \cdots dz_d = \int_{\mathbb{R}^{d-i}} q(\boldsymbol{\varphi}(\mathbf{z})) \prod_{j=1}^{d} \left| \frac{\partial \varphi_j}{\partial z_j}\left(\mathbf{z}_{[j]}\right) \right| dz_{i+1} \ldots dz_d$$

$$= \prod_{j=1}^{i} \left| \frac{\partial \varphi_j}{\partial z_j}\left(\mathbf{z}_{[j]}\right) \right| \int_{\mathbb{R}^{d-i}} q\left(\boldsymbol{\varphi}_{[i]}\left(\mathbf{z}_{[i]}\right), \varphi_{i+1}\left(z_{i+1}\right), \cdots, \varphi_d\left(z_d\right)\right) \prod_{k=i+1}^{d} \left| \frac{\partial \varphi_k}{\partial z_k}\left(z_k\right) \right| dz_{i+1} \cdots dz_d. \tag{26}$$

By a change of variables $\begin{cases} u_{i+1} = \varphi_{i+1}\left(z_{i+1}\right) \\ \qquad \vdots \\ u_d = \varphi_d\left(z_d\right) \end{cases}$, we get

$$p_{[i]}(\mathbf{z}_{[i]}) = \prod_{j=1}^{i} \left| \frac{\partial \varphi_j}{\partial z_j}\left(\mathbf{z}_{[j]}\right) \right| \int_{\mathbb{R}^{d-i}} q\left(\boldsymbol{\varphi}_{[i]}\left(\mathbf{z}_{[i]}\right), u_{i+1}, \cdots, u_d\right) du_{i+1} \cdots du_d$$

$$= q_{[i]}\left(\boldsymbol{\varphi}_{[i]}\left(\mathbf{z}_{[i]}\right)\right) \left| \det D\boldsymbol{\varphi}_{[i]}\left(z_{[i]}\right) \right|$$

$$= q_{[i-1]}\left(\boldsymbol{\varphi}_{[i-1]}\left(\mathbf{z}_{[i-1]}\right)\right) \tilde{q}_i\left(\varphi_i(\mathbf{z})\right) \left| \det D\boldsymbol{\varphi}_{[i-1]}\left(\mathbf{z}_{[i-1]}\right) \right| \left| \frac{\partial \varphi_i}{\partial z_i}(\mathbf{z}) \right|.$$

By Lemma B.4, $p_{[i-1]}\left(\mathbf{z}_{[i-1]}\right) = q_{[i-1]}\left(\boldsymbol{\varphi}_{[i-1]}\left(\mathbf{z}_{[i-1]}\right)\right) \left| \det D\boldsymbol{\varphi}_{[i-1]}\left(\mathbf{z}_{[i-1]}\right) \right|$. By Lemma B.1, $\boldsymbol{\varphi}_{[i]}$ is constant in the first $i-1$ variables.

In addition, $D\boldsymbol{\varphi}(\mathbf{z})$ is lower triangular for all $z$, so we have proven that $\boldsymbol{\varphi} \in \mathcal{S}_{\text{scaling}}$.

$\square$

### B.3 Proof of Prop. 4.3

**Proposition 4.3.** *Given a DAG $G$, with $d-1$ perfect stochastic single node interventions on distinct targets, if the remaining unintervened node has any parent in $G$, $(G, \mathcal{F}, P_G)$ is not identifiable up to*

$$\mathcal{S}_{reparam} := \left\{ \mathbf{g} : \mathbb{R}^d \to \mathbb{R}^d \mid \mathbf{g} = \mathbf{P} \circ \mathbf{h}, \mathbf{P} \text{ is a permutation matrix}, \mathbf{h} \in \mathcal{S}_{scaling} \right\}. \tag{6}$$

*Proof.* Without loss of generality by rearranging $\left(\mathbb{P}^i, \tau_i\right)_{i \in [\![0, d-1]\!]}$, suppose that the unintervened variable is the node $d$. Fix any $\left(G, \mathbf{f}, \left(\mathbb{P}^i, \tau_i\right)_{i \in [\![0, d-1]\!]}\right)$, s.t. $d-1$ is a parent of $d$, and s.t. the causal mechanism of $Z_d$ only has one conditional variable $Z_{d-1}$, and $Z_d \sim \mathcal{N}\left(Z_{d-1}, 1\right)$, namely,

$$p_d\left(z_d \mid z_{d-1}\right) = \frac{1}{\sqrt{2\pi}} \exp\left(-\frac{\left(z_d - z_{d-1}\right)^2}{2}\right).$$

We now construct $\left(G, \mathbf{f}', \left(\mathbb{Q}^i, \tau_i\right)_{i \in [\![0, d-1]\!]}\right)$ such that $\mathbf{f}_* \mathbb{P}^i = \mathbf{f}'_* \mathbb{Q}^i$ and $\mathbf{f}'^{-1} \circ \mathbf{f} \notin \mathcal{S}_{\text{reparam}}$.

Set $\mathbb{Q}_i\left(Z_i \mid \mathbf{Z}_{\text{pa}(i)}\right) :\stackrel{(d)}{=} \mathbb{P}_i\left(Z_i \mid \mathbf{Z}_{\text{pa}(i)}\right), \widetilde{\mathbb{Q}}_i\left(Z_i\right) :\stackrel{(d)}{=} \widetilde{\mathbb{P}}_i\left(Z_i \mid \mathbf{Z}_{\text{pa}(i)}\right) \quad \forall i \in [d-1].$

Set $\mathbb{Q}_d\left(Z_d \mid Z_{d-1}\right) \overset{(d)}{:=} \mathcal{N}\left(-Z_{d-1}, 1\right)$, $\varphi(\mathbf{z}) := \left(z_1, \cdots, -z_{d-1}, z_d - 2z_{d-1}\right)$, thus $|\operatorname{Det} D\varphi(\mathbf{z})| = 1 \quad \forall \mathbf{z} \in \mathbb{R}^d$.

$$q_d\left(\varphi_d(\mathbf{z}) \mid \varphi_{d-1}(\mathbf{z})\right) = \frac{1}{\sqrt{2\pi}} \exp\left(-\frac{\left(\varphi_d(\mathbf{z}) - \varphi_{d-1}(\mathbf{z})\right)^2}{2}\right)$$

$$= \frac{1}{\sqrt{2\pi}} \exp\left(-\frac{\left(z_d - 2z_{d-1} + z_{d-1}\right)^2}{2}\right)$$

$$= p_d\left(z_d \mid z_{d-1}\right).$$

From the above equation, we infer that for the unintervened regime,

$$\prod_{j=1}^{d} p_j\left(z_j \mid \mathbf{z}_{\mathrm{pa}(j)}\right) = \left[\prod_{j=1}^{d} q_j\left(\varphi_j(\mathbf{z}) \mid \boldsymbol{\varphi}_{\mathrm{pa}(j)}(\mathbf{z})\right)\right] |\det D\boldsymbol{\varphi}(\mathbf{z})|,$$

and for the $d - 1$ interventional regimes,

$$\forall i \in [d-1], \tilde{p}_i\left(z_i\right) \prod_{j \neq i} p_j\left(z_j \mid \mathbf{z}_{\mathrm{pa}(j)}\right) = \tilde{q}_i\left(\varphi_i(z)\right) \left[\prod_{j \neq i} q_j\left(\varphi_j(z) \mid \boldsymbol{\varphi}_{\mathrm{pa}(j)}(z)\right)\right] |\det D\boldsymbol{\varphi}(z)|,$$

i.e.,

$$\boldsymbol{\varphi}_* \mathbb{P}^i = \mathbb{Q}^i \quad \forall i \in [\![0, d-1]\!].$$

$$\mathbf{f}_* \mathbb{P}^i = \left(\mathbf{f} \circ \boldsymbol{\varphi}^{-1}\right)_* \mathbb{Q}^i$$

However, $\left(\mathbf{f} \circ \boldsymbol{\varphi}^{-1}\right)^{-1} \circ \mathbf{f} = \boldsymbol{\varphi} \notin S_{\mathrm{reparam}}$. $\qquad\square$

## B.4 Proof of Thm. 4.5

**Theorem 4.5.** *Given any DAG $G$. Suppose that our datasets encompass interventions over all variables in the latent graph, i.e., $\bigcup_{k \in [K]} \tau_k = [d]$. For all $k \in [K]$, suppose the targets of interventions are strict subsets of all variables, i.e., $|\tau_k| = n_k$, $n_k \in [d-1]$. Suppose the interventions over $\tau_k$ are perfect, i.e. the intervention mechanisms $\mathbb{Q}^s_{\tau_k}$ are joint distributions over $\mathbf{Z}_{\tau_k}$ without conditioning on other variables. Suppose Asm. 4.4 is satisfied for $\tau_k$.*

*Then CauCA in $(G, \mathcal{F}, \mathcal{P}_G)$ is block-identifiable (following [54]): namely, if for all $k \in [K]$, $\mathbf{f}_* \mathbb{P}^k = \mathbf{f}'_* \mathbb{Q}^k$, then for $\boldsymbol{\varphi} := \mathbf{f}'^{-1} \circ \mathbf{f}$, for all $k \in [K]$,*

$$[\boldsymbol{\varphi}(\mathbf{z})]_{\tau_k} = \boldsymbol{\varphi}_{\tau_k}\left(\mathbf{z}_{\tau_k}\right). \tag{8}$$

*Proof.* Fix $k \in [K]$. Then the equation of the $k$-th interventional regime is $p^k(\mathbf{z}) = q^k(\boldsymbol{\varphi}(\mathbf{z}))|\det\varphi(\mathbf{z})|$.

- If $\tau_k$ has no parents from $[d] \setminus \tau_k$, then in the unintervened regime, $p_k^0$ and $q_k^0$ have no conditional. Since interventions over $\tau_k$ are perfect, $p_k^s$ and $q_k^s$ have no conditional for all $s \in [\![1, n_k]\!]$.

- If $\tau_k$ has parents from $[d] \setminus \tau_k$, since in this case there are $n_k + 1$ perfect interventions, enumerated by $s \in [\![0, n_k]\!]$, $p_k^s$ and $q_k^s$ have no conditional for all $s \in [\![0, n_k]\!]$.

In both cases, $p_k^s$ and $q_k^s$ have no conditional for all $s \in [\![0, n_k]\!]$.

We write the equality of pushforward densities just as Prop. 4.6, and subtract the $k$-th interventional regime by the unintervened regime:

$$\ln p^s\left(\mathbf{z}_{\tau_k}\right) - \ln p^0\left(\mathbf{z}_{\tau_k}\right) = \ln q^s\left(\boldsymbol{\varphi}_{\tau_k}(\mathbf{z})\right) - \ln q^0\left(\boldsymbol{\varphi}_{\tau_k}(\mathbf{z})\right).$$

For all $i \in [d] \setminus \tau_k$ (nonempty by assumption), take the partial derivative of $z_i$:

$$0 = \sum_{j=1}^{n_k} \left[ \frac{\partial}{\partial x_j} (\ln q_{\tau_k}^1)(\boldsymbol{\varphi}_{\tau_k}(\mathbf{z}_{\tau_k})) - \frac{\partial}{\partial x_j} (\ln q_{\tau_k}^0)(\boldsymbol{\varphi}_{\tau_k}(\mathbf{z}_{\tau_k})) \right] \frac{\partial \varphi_{\tau_k,j}}{\partial z_i}(\mathbf{z})$$

By assumption, for all $s \in [n_k]$, there is one interventional regime in which the above equation holds. Those $n_k$ equations form a linear system $\mathbf{0} = \mathbf{M}_{\tau_k}(\mathbf{z}_{\tau_k}) \frac{\partial \boldsymbol{\varphi}_{\tau_k}}{\partial z_i}(\mathbf{z}_{\tau_k})$, where $\mathbf{M}_{\tau_k}(\mathbf{z}_{\tau_k})$ is defined in the statement of theorem. Since $\mathbf{M}_{\tau_k}(\mathbf{z})$ is invertible a. e. by assumption, the vector $\frac{\partial \boldsymbol{\varphi}_{\tau_k}}{\partial z_i}(\mathbf{z}) = \mathbf{0}$ $\forall \mathbf{z} \in \mathbb{R}^d$ a. e., which is furthermore strictly everywhere since $\boldsymbol{\varphi} := \mathbf{f}'^{-1} \circ \mathbf{f}$ is $\mathcal{C}^1$. Such a result is valid for all $i \in [d] \backslash \tau_k$. Since $\bigcup_{k \in I} \tau_k = [d]$, all the non-diagonal entries of $D\boldsymbol{\varphi}(\mathbf{z})$ that are not in the blocks of $\tau_k \times \tau_k$ are 0. We conclude that $\boldsymbol{\varphi}_{\tau_i}$ only depends on $\mathbf{z}_{\tau_i}$, i.e., $[\boldsymbol{\varphi}(\mathbf{z})]_{\tau_k} = \boldsymbol{\varphi}_{\tau_k}(\mathbf{z}_{\tau_k})$.

For all $k \in [K]$, $[d] \setminus \tau_k \neq \varnothing$. Suppose there exists $\mathbf{z} \in \mathbb{R}^d$ such that $\det(D\boldsymbol{\varphi}_{\tau_k}(\mathbf{z})) = 0$. Since $[\boldsymbol{\varphi}(\mathbf{z})]_{\tau_i} = \boldsymbol{\varphi}_{\tau_i}(\mathbf{z}_{\tau_i}) \, \forall i \in [n]$, the vector $\frac{\partial \boldsymbol{\varphi}_{\tau_k}}{\partial z_i}(\mathbf{z}) = \mathbf{0}$ for all $i \notin \tau_k$. Thus the rows $\tau_k$ of $D\boldsymbol{\varphi}(\mathbf{z})$ are linearly dependent, which implies $\det(D\boldsymbol{\varphi}(\mathbf{z})) = 0$, which contradicts with $\boldsymbol{\varphi}$ invertible. Thus $\boldsymbol{\varphi}_{\tau_k}$ is a diffeomorphism. $\qquad\square$

## B.5 Proof of Prop. 4.6

**Proposition 4.6.** *Suppose that $G$ is the empty graph, and that there are $d-1$ variables intervened on, with one single target per dataset, such that Asm. 4.1 is satisfied. Then CauCA (in this case, ICA) in $(G, \mathcal{F}, \mathcal{P}_G)$ is identifiable up to $\mathcal{S}_{scaling}$ defined as in eq. (3).*

*Proof.* Without loss of generality by rearranging $(\mathbb{P}^i, \tau_i)_{i \in [\![0,d-1]\!]}$, suppose that the unintervened variable is the node $d$. We apply the induction in the proof of Thm. 4.2 *(i)*. Since there are $d-1$ interventions, the induction stops at $d-1$, and we can infer that for $\boldsymbol{\varphi} := \mathbf{f}'^{-1} \circ \mathbf{f}$, for all $i \in [d-1]$, $\frac{\partial \varphi_i}{\partial z_j}(\mathbf{z}) = 0$ a.e. $\forall j \neq i$.

Similar to Thm. 4.2 *(i)*, since $\frac{\partial \varphi_i}{\partial z_j}$ is continuous, it equals zero everywhere. Thus $D\boldsymbol{\varphi}(z)$ is lower triangular for all $z \in \mathbb{R}^d$.

By Lemma B.4, $(\boldsymbol{\varphi}_{[d-1]})_* \mathbb{P}_{[d-1]}(\mathbf{Z}_{[d-1]}) = \mathbb{Q}_{[d-1]}(\mathbf{Z}_{[d-1]})$. Namely,

$$\prod_{j=1}^{d-1} p_j(z_j) = \prod_{j=1}^{d-1} q_j(\varphi_j(\mathbf{z})) \left| \det D\boldsymbol{\varphi}_{[d-1]}(\mathbf{z}_{[d-1]}) \right|. \tag{27}$$

Since for all $i \in [d-1]$, $\forall j \neq i$, $\frac{\partial \varphi_i}{\partial z_j}(\mathbf{z}) = 0$, $D\boldsymbol{\varphi}(\mathbf{z})$ is lower triangular for all $z \in \mathbb{R}^d$. Thus $|\det D\boldsymbol{\varphi}(\mathbf{z})| = \prod_{j=1}^d |\partial_j \varphi_j(\mathbf{z})|$. Moreover, in the unintervened dataset,

$$\prod_{j=1}^{d} p_j(z_j) = \prod_{j=1}^{d} q_j(\varphi_j(\mathbf{z})) |\partial_j \varphi_j(\mathbf{z})|. \tag{28}$$

Divide (28) by (27),

$$p_d(z_d) = q_d(\varphi_d(\mathbf{z})) \left| \frac{\partial \varphi_d}{\partial z_d}(\mathbf{z}_{[d-1]}, z_d) \right|,$$

which is equivalent to

$$\forall \mathbf{z}_{[d-1]} \in \mathbb{R}^{d-1}, \quad \widetilde{\mathbb{Q}_d} = \varphi_d(\mathbf{z}_{[d-1]}, \cdot)_* \widetilde{\mathbb{P}}_d. \tag{29}$$

By Lemma B.1, $\varphi_d$ is constant in the first $d-1$ variables. We have proven that $\boldsymbol{\varphi} \in \mathcal{S}_{scaling}$.

$\qquad\square$

## B.6 Proof of Prop. 4.7

**Proposition 4.7.** *Given an empty graph $G$, with $d-2$ single-node interventions on distinct targets, with one single target per dataset, such that Asm. 4.1 is satisfied. Then CauCA (in this case, ICA) in $(G, \mathcal{F}, \mathcal{P}_G)$ is not identifiable up to $\mathcal{S}_{reparam}$.*

*Proof.* Without loss of generality by rearranging $\left(\mathbb{P}^i, \tau_i\right)_{i \in [\![0,d-2]\!]}$, suppose that the two unintervened variables are the nodes $d-1, d$. Fix any $\mathbf{f}$ and $\left(\mathbb{P}_i, \widetilde{\mathbb{P}}_i\right)_{i \in [\![0,d-2]\!]}$ such that for all $i \in [d-2]$, $\mathbb{P}_i, \widetilde{\mathbb{P}}_i$ have any distribution that is absolutely continuous and full support in $\mathbb{R}$ with a differentiable density, and such that Asm. 4.1 is satisfied. We will prove that whether we suppose independent Gaussian distributions are in the class of latent distributions or not, CauCA in $(G, \mathcal{F}, \mathcal{P}_G)$ is not identifiable up to $S_{\text{reparam}}$.

**Case 1: Independent Gaussian distributions are in $\mathcal{P}_G$.**

By the famous result in linear ICA, if $\mathbb{P}_{d-1}, \mathbb{P}_d$ form an isotropic Gaussian vector $\mathcal{N}(\mathbf{0}, \boldsymbol{\Sigma})$, i.e., $\boldsymbol{\Sigma}$ is diagonal with the same variances on each dimension, then any rotation form a spurious solution. Namely, let $\boldsymbol{\varphi}(\mathbf{z}) = (z_1, \ldots, z_{d-2}, z_{d-1}\cos(\theta) - z_d \sin(\theta), z_d \cos(\theta) + z_{d-1}\sin(\theta))$, $\theta \neq k\pi$, then

$$\forall i \in [\![0, d-2]\!] \quad \boldsymbol{\varphi}_* \mathbb{P}^i = \mathbb{P}^i$$
$$\mathbf{f}_* \mathbb{P}^i = \left(\mathbf{f} \circ \boldsymbol{\varphi}^{-1}\right)_* \mathbb{P}^i$$

However, $\left(\mathbf{f} \circ \boldsymbol{\varphi}^{-1}\right)^{-1} \circ \mathbf{f} = \boldsymbol{\varphi} \notin S_{\text{reparam}}$.

**Case 2: Independent Gaussian distributions are not in $\mathcal{P}_G$.**

Suppose that for all $i \in \{d-1, d\}$, $\mathbb{P}_i$ has the same density $p_a$:

$$p_a(z) = \begin{cases} \exp(-az^2) & z < 0 \\ 1 & 0 \leq z \leq 1 - \sqrt{\frac{\pi}{a}} \\ \exp\left(-a\left(z - \left(1 - \sqrt{\frac{\pi}{a}}\right)\right)^2\right) & z > 1 - \sqrt{\frac{\pi}{a}} \end{cases}$$

where $\sqrt{\frac{\pi}{a}} < 1$. One can verify that $p_a$ is a smooth p.d.f.

We construct a measure-preserving automorphism inspired by [19].

$$\boldsymbol{\varphi}(\mathbf{Z}) = \begin{cases} \mathbf{Z} & ||\mathbf{Z}_{[\![d-1,d]\!]}|| \geq R \\ \begin{pmatrix} \mathbf{Z}_{[d-2]} \\ \cos(\alpha(||\mathbf{Z}_{[\![d-1,d]\!]} - \mathbf{C}|| - R))Z_{d-1} \\ \quad - \sin(\alpha(||\mathbf{Z}_{[\![d-1,d]\!]} - \mathbf{C}|| - R))Z_d \\ \cos(\alpha(||\mathbf{Z}_{[\![d-1,d]\!]} - \mathbf{C}|| - R))Z_d \\ \quad + \sin(\alpha(||\mathbf{Z}_{[\![d-1,d]\!]} - \mathbf{C}|| - R))Z_{d-1} \end{pmatrix} & ||\mathbf{Z}_{[\![d-1,d]\!]}|| < R \end{cases}$$

where $\alpha \neq 0$, $\mathbf{C} = \left(\frac{1}{2}\left(1 - \sqrt{\frac{\pi}{a}}\right), \frac{1}{2}\left(1 - \sqrt{\frac{\pi}{a}}\right)\right)$, $R \in \left(0, \frac{1}{2}\left(1 - \sqrt{\frac{\pi}{a}}\right)\right]$.

Now let us prove that $\boldsymbol{\varphi}$ preserves $\mathbb{P}_{[\![d-1,d]\!]}$. By shifting the center of $p_a$ to the origin, we only need to prove that the shifted $\boldsymbol{\varphi}_{[\![d-1,d]\!]}$ preserves the uniform distribution over $[-R, R]^2$. One can verify that $\boldsymbol{\varphi}_{[\![d-1,d]\!]}$ is a diffeomorphism over the 2-dimensional open disk $D^2(\mathbf{0}, R) \setminus \{\mathbf{0}\} \to D^2(\mathbf{0}, R)\setminus\{\mathbf{0}\}$ and $|\det(D\boldsymbol{\varphi}_{[\![d-1,d]\!]}(\mathbf{z}))| = 1$. Thus $p_a(\mathbf{z}) = p_a(\boldsymbol{\varphi}_{[\![d-1,d]\!]}(\mathbf{z}))|\det(D\boldsymbol{\varphi}_{[\![d-1,d]\!]}(\mathbf{z}))|$ $\forall \mathbf{z} \in D^2(\mathbf{0}, R)\setminus\{\mathbf{0}\}$. Since $\boldsymbol{\varphi} = Id$ outside of the disk, this change of variables formula holds almost everywhere in $\mathbb{R}^2$, thus $\mathbb{P}_{[\![d-1,d]\!]} = (\boldsymbol{\varphi}_{[\![d-1,d]\!]})_* \mathbb{P}_{[\![d-1,d]\!]}$, namely, $\boldsymbol{\varphi}_{[\![d-1,d]\!]}$ preserves $\mathbb{P}_{[\![d-1,d]\!]}$ on $\mathbb{R}^2$.

Moreover, since $\boldsymbol{\varphi}_{[d-2]}$ is identity, $\widetilde{\mathbb{P}}_i = (\boldsymbol{\varphi}_i)_* \widetilde{\mathbb{P}}_i$ and $\mathbb{P}_i = (\boldsymbol{\varphi}_i)_* \mathbb{P}_i$ for all $i \in [d-2]$. Thus

$$\forall i \in [\![0, d-2]\!] \quad \boldsymbol{\varphi}_* \mathbb{P}^i = \mathbb{P}^i,$$
$$\mathbf{f}_* \mathbb{P}^i = \left(\mathbf{f} \circ \boldsymbol{\varphi}^{-1}\right)_* \mathbb{P}^i.$$

However, $\left(\mathbf{f} \circ \boldsymbol{\varphi}^{-1}\right)^{-1} \circ \mathbf{f} = \boldsymbol{\varphi} \notin S_{\text{reparam}}$.

$\square$

## B.7 Further constraint on the indeterminacy set

**Corollary B.5.** *Based on the assumption of (2) of Thm. 4.2, if in every dataset we are given the set of possible intervention mechanisms: $\mathcal{M} = (\mathcal{M}_i)_{i \in [d]}$, then CauCA in $(G, \mathcal{F}, \mathcal{P}_G)$ is identifiable up to $\mathcal{S}_\mathcal{M} := \{\varphi : \mathbb{R}^n \to \mathbb{R}^n | \varphi \in \mathcal{S}_{scaling}, \forall i \in [d], \varphi_i \in \mathcal{S}_{\mathcal{M}_i}\}$ where $\mathcal{S}_{\mathcal{M}_i} := \{\bar{F}_{\mathbb{M}_i'}^{-1} \circ F_{\mathbb{M}_i} | \mathbb{M}_i, \mathbb{M}_i' \in \mathcal{M}_i\}$*

*In particular, if $\mathcal{M}_i$ is singleton for all $i$, then CauCA in $(G, \mathcal{F}, \mathcal{P}_G)$ is identifiable up to $\mathcal{S}_{reflexion} := \{\mathbf{h} : \mathbb{R}^d \to \mathbb{R}^d | \forall i \in [d], h_i = Id \text{ or } -Id\}$.*

*Proof.* Based on the conclusion of (2) of Thm. 4.2, for any $i \in [d]$, choose any $\mathbb{M}_i, \mathbb{M}_i'$ in $\mathcal{M}_i$,

$$(\varphi_i)_* \mathbb{M}_i = \mathbb{M}_i'.$$

By Lemma B.2, the only possible $\varphi_i$ are $F_{\mathbb{M}_i'}^{-1} \circ F_{\mathbb{M}_i}$ and $\bar{F}_{\mathbb{M}_i'}^{-1} \circ F_{\mathbb{M}_i}$. Thus $\varphi_i \in \mathcal{S}_{\mathcal{M}_i}$. In particular if $\mathcal{M}_i$ is a singleton $\{\mathbb{M}_i\}$, then $F_{\mathbb{M}_i}^{-1} \circ F_{\mathbb{M}_i} =$Id, and $\bar{F}_{\mathbb{M}_i}^{-1} \circ F_{\mathbb{M}_i} = -$Id. $\qquad\square$

## B.8 Proof of Corollary 4.8

**Corollary 4.8.** *[Corollary of Thm. 4.5] Suppose $G$ is the empty graph. Suppose that our datasets encompass interventions over all variables in the latent graph, i.e., $\bigcup_{k \in [K]} \tau_k = [d]$. Suppose for every $k$, the targets of interventions are a strict subset of all variables, i.e., $|\tau_k| = n_k$, $n_k \in [d-1]$.*

*Suppose Asm. 4.4 is verified, which has a simpler form in this case: there are $n_k$ interventions with target $\tau_k$ such that $\mathbf{v}_k(\mathbf{z}_{\tau_k}, 1) - \mathbf{v}_k(\mathbf{z}_{\tau_k}, 0), \cdots, \mathbf{v}_k(\mathbf{z}_{\tau_k}, n_k) - \mathbf{v}_k(\mathbf{z}_{\tau_k}, 0)$ are linearly independent, where*

$$\mathbf{v}_k(\mathbf{z}_{\tau_k}, s) := \left( (\ln q_{\tau_k,1}^s)'(z_{\tau_k,1}), \cdots, (\ln q_{\tau_k,n_k}^s)'(z_{\tau_k,n_k}) \right), \tag{9}$$

*where $q_{\tau_k}^s$ is the intervention of the $s$-th interventional regime that has the target $\tau_k$, and $q_{\tau_k,j}^s$ is the $j$-th marginal of it. $z_{\tau_k,j}$ is the $j$-th dimension of $\mathbf{z}_{\tau_k}$. $s = 0$ denotes the unintervened regime.*

*Then CauCA in $(G, \mathcal{F}, \mathcal{P}_G)$ is* block-identifiable, *in the same sense as Thm. 4.5.*

*Proof.* This corollary is a special case of Thm. 4.5 when $G$ has no arrow. Since $G$ has no arrow, all the blocks of interventions are in the first case of block-interventional discrepancy in Thm. 4.5: for all $k \in [K]$, $|\tau_k| = n_k$. To prove all the blocks satisfy the block-interventional discrepancy, it suffices to prove that the linearly independent vectors in the statement form the matrix $\mathbf{M}_{\tau_k}$. To see this, it suffices to notice that for all $s \in [\![0, n_k]\!]$, $\ln q_{\tau_k}^s(\mathbf{z}_{\tau_k}) = \sum_{j \in \tau_k} \ln q_{\tau_k,j}^s(z_{\tau_k,j})$, and therefore $\frac{\partial}{\partial z_j}(\ln q_{\tau_k}^s)(\mathbf{z}_{\tau_k}) = (\ln q_{\tau_k,j}^s)'(z_{\tau_k,j})$. $\qquad\square$

## B.9 Proof of Prop. 4.9

**Proposition 4.9.** *Under the assumptions of Thm. 4.5, suppose furthermore that all density functions in $\mathcal{P}_G$ and all mixing functions in $\mathcal{F}$ are $\mathcal{C}^2$, and suppose there exist $k \in [K]$ and there are $2n_k$ interventions with targets $\tau_k$ such that for any $\mathbf{z}_{\tau_k} \in \mathbb{R}^{n_k}$, $\mathbf{w}_k(\mathbf{z}_{\tau_k}, 1) - \mathbf{w}_k(\mathbf{z}_{\tau_k}, 0), \ldots, \mathbf{w}_k(\mathbf{z}_{\tau_k}, 2n_k) - \mathbf{w}_k(\mathbf{z}_{\tau_k}, 0)$ are linearly independent, where*

$$\mathbf{w}_k(\mathbf{z}_{\tau_k}, s) := \left( \left( \frac{q_{\tau_k,1}^{s\prime}}{q_{\tau_k,1}^s} \right)'(z_{\tau_k,1}), \ldots, \left( \frac{q_{\tau_k,n_k}^{s\prime}}{q_{\tau_k,n_k}^s} \right)'(z_{\tau_k,n_k}), \frac{q_{\tau_k,1}^{s\prime}}{q_{\tau_k,1}^s}(z_{\tau_k,1}), \ldots, \frac{q_{\tau_k,n_k}^{s\prime}}{q_{\tau_k,n_k}^s}(z_{\tau_k,n_k}) \right),$$

*where $q_{\tau_k}^s$ is the intervention of the $s$-th interventional regime that has the target $\tau_k$, and $q_{\tau_k,j}^s$ is the $j$-th marginal of it. $z_{\tau_k,j}$ is the $j$-th dimension of $\mathbf{z}_{\tau_k}$. $s = 0$ denotes the unintervened regime. Then*

$$\varphi_{\tau_k} \in \mathcal{S}_{reparam} := \left\{ \mathbf{g} : \mathbb{R}^{n_k} \to \mathbb{R}^{n_k} | \mathbf{g} = \mathbf{P} \circ \mathbf{h} \text{ where } \mathbf{P} \text{ is a permutation matrix and } \mathbf{h} \in \mathcal{S}_{scaling} \right\}.$$

*The proof is based on Theorem 1 of [21].*

*Proof.* Since the intervention targets are known, without loss of generality, suppose the interventions are on the first $n_k$ variables. By the result of Thm. 4.5 we have $p_k^s(\mathbf{z}_{\tau_k}) =$

$q_k^s\left(\varphi_{\tau_k}\left(\mathbf{z}_{\tau_k}\right)\right)\left|\det D\varphi_{\tau_k}\left(\mathbf{z}_{\tau_k}\right)\right|$ where $p_k^s$ denotes the joint distribution in the $s$-th interventional regime such that the intervention target is $\tau_k$. Factorize $p_k^s$ and $q_k^s$ in the change of variables formula, and take the logarithm:

$$\sum_{l=1}^{n_k}\ln p_{k,l}^s\left(z_l\right)=\sum_{l=1}^{n_k}\ln q_{k,l}^s\left(\varphi_l\left(\mathbf{z}_{\tau_k}\right)\right)+\ln\left|\det D\varphi_{\tau_k}\left(\mathbf{z}_{\tau_k}\right)\right|, \tag{30}$$

where $p_l^s$ is the $l$-th marginal of the intervention of the $s$-th interventional regime that has the target $\tau_k$.

For the unintervened regime, denote the density of the $l$-th marginal of $\mathbb{P}$ as $p_l$. By the result of Thm. 4.5 we have

$$\sum_{l=1}^{n_k}\ln p_l^0\left(z_l\right)=\sum_{l=1}^{n_k}\ln q_l^0\left(\varphi_l\left(\mathbf{z}_{\tau_k}\right)\right)+\ln\left|\det D\varphi_{\tau_k}\left(\mathbf{z}_{\tau_k}\right)\right|. \tag{31}$$

Subtract the equation (30) by (31):

$$\sum_{l=1}^{n_k}\left[\ln p_{k,l}^s\left(z_l\right)-\ln p_l^0\left(z_l\right)\right]=\sum_{l=1}^{n_k}\left[\ln q_{k,l}^s\left(\varphi_l\left(\mathbf{z}_{\tau_k}\right)\right)-\ln q_l^0\left(\varphi_l\left(\mathbf{z}_{\tau_k}\right)\right)\right]. \tag{32}$$

For any $j\in\tau_k=[n_k]$, take the partial derivative over $z_j$

$$\frac{p_{k,j}^{s\prime}\left(z_j\right)}{p_{k,j}^s\left(z_j\right)}-\frac{p_j^{0\prime}\left(z_j\right)}{p_j^0\left(z_j\right)}=\sum_{l=1}^{n_k}\left[\frac{q_{k,l}^{s\prime}\left(\varphi_l\left(\mathbf{z}_{\tau_k}\right)\right)}{q_{k,l}^s\left(\varphi_l\left(\mathbf{z}_{\tau_k}\right)\right)}-\frac{q_{k,l}^{0\prime}\left(\varphi_l\left(\mathbf{z}_{\tau_k}\right)\right)}{q_{k,l}^0\left(\varphi_l\left(\mathbf{z}_{\tau_k}\right)\right)}\right]\frac{\partial\varphi_l}{\partial z_j}\left(\mathbf{z}_{\tau_k}\right). \tag{33}$$

For any $1\leqslant k<j$, take the partial derivative over $z_k$,

$$0=\sum_{l=1}^{n_k}\left[\left(\frac{q_{k,l}^{s\prime}}{q_{k,l}^s}\right)^{\prime}\left(\varphi_l\left(\mathbf{z}_{\tau_k}\right)\right)-\left(\frac{q_{k,l}^{0\prime}}{q_{k,l}^0}\right)^{\prime}\left(\varphi_l\left(\mathbf{z}_{\tau_k}\right)\right)\right]\frac{\partial\varphi_l\left(\mathbf{z}_{\tau_k}\right)}{\partial z_k}\frac{\partial\varphi_l\left(\mathbf{z}_{\tau_k}\right)}{\partial z_j}$$
$$+\left[\frac{q_{k,l}^{s\prime}\left(\varphi_l\left(\mathbf{z}_{\tau_k}\right)\right)}{q_{k,l}^s\left(\varphi_l\left(\mathbf{z}_{\tau_k}\right)\right)}-\frac{q_{k,l}^{0\prime}\left(\varphi_l\left(\mathbf{z}_{\tau_k}\right)\right)}{q_{k,l}^0\left(\varphi_l\left(\mathbf{z}_{\tau_k}\right)\right)}\right]\frac{\partial^2\varphi_l\left(\mathbf{z}_{\tau_k}\right)}{\partial z_k\partial z_j}. \tag{34}$$

For $1\leqslant k<j\leqslant n_k$ there are $\frac{n_k(n_k-1)}{2}$ equations.

Define $\mathbf{a}_l\left(\mathbf{z}_{\tau_k}\right)=\left(\frac{\partial\varphi_l}{\partial z_k}\left(\mathbf{z}_{\tau_k}\right)\frac{\partial\varphi_l}{\partial z_j}\left(\mathbf{z}_{\tau_k}\right)\right)_{1\leqslant k\leqslant j\leqslant n_k}$, $\mathbf{b}_l\left(\mathbf{z}_{\tau_k}\right)=\left(\frac{\partial^2\varphi_l\left(\mathbf{z}_{\tau_k}\right)}{\partial z_k\partial z_j}\right)_{1\leqslant k<j\leqslant n_k}$.

Then the $\frac{n_k(n_k-1)}{2}$ equations can be written as a linear system

$$0=\sum_{l=1}^{n_k}\mathbf{a}_l\left(\mathbf{z}_{\tau_k}\right)\left[\left(\frac{q_{k,l}^{s\prime}}{q_{k,l}^s}\right)^{\prime}\left(\varphi_l\left(\mathbf{z}_{\tau_k}\right)\right)-\left(\frac{q_{k,l}^{0\prime}}{q_{k,l}^0}\right)^{\prime}\left(\varphi_l\left(\mathbf{z}_{\tau_k}\right)\right)\right]$$
$$+\mathbf{b}_l\left(\mathbf{z}_{\tau_k}\right)\left[\frac{q_{k,l}^{s\prime}\left(\varphi_l\left(\mathbf{z}_{\tau_k}\right)\right)}{q_{k,l}^s\left(\varphi_l\left(\mathbf{z}_{\tau_k}\right)\right)}-\frac{q_{k,l}^{0\prime}\left(\varphi_l\left(\mathbf{z}_{\tau_k}\right)\right)}{q_{k,l}^0\left(\varphi_l\left(\mathbf{z}_{\tau_k}\right)\right)}\right].$$

Define $\mathbf{M}_k\left(\mathbf{z}_{\tau_k}\right)=\left(\mathbf{a}_1\left(\mathbf{z}_{\tau_k}\right),\cdots,\mathbf{a}_{n_k}\left(\mathbf{z}_{\tau_k}\right),\mathbf{b}_1\left(\mathbf{z}_{\tau_k}\right),\cdots,\mathbf{b}_{n_k}\left(\mathbf{z}_{\tau_k}\right)\right)$.

Collect the equations for $s=1,\cdots,2n_k$,

$$\mathbf{0}=\mathbf{M}_k\left(\mathbf{z}_{\tau_k}\right)\left(\mathbf{w}_k\left(\varphi_{\tau_k}\left(\mathbf{z}_{\tau_k}\right),1\right)-\mathbf{w}_k\left(\varphi_{\tau_k}\left(\mathbf{z}_{\tau_k}\right),0\right),\ldots,\right.$$
$$\left.\mathbf{w}_k\left(\varphi_{\tau_k}\left(\mathbf{z}_{\tau_k}\right),2n_k\right)-\mathbf{w}_k\left(\varphi_{\tau_k}\left(\mathbf{z}_{\tau_k}\right),0\right)\right).$$

By assumption, the matrix containing $\mathbf{w}$ is invertible. Thus $M_k\left(\mathbf{z}_{\tau_k}\right)=\mathbf{0}$, which implies $\mathbf{a}_\ell\left(\mathbf{z}_{\tau_k}\right)$ are zero for all $\mathbf{z}_{\tau_k}$. By the same reasoning as [21], each row in $D\varphi_{\tau_k}\left(\mathbf{z}_{\tau_k}\right)$ has only one non-zero term, and this does not change for different $z$, since otherwise by continuity there exists $\mathbf{z}$ such that $D\varphi_{\tau_k}\left(\mathbf{z}_{\tau_k}\right)$ is singular, contradiction with the invertibility of $\varphi_{\tau_k}$ (Thm. 4.5). Thus $\forall i\in\tau_k$, $\varphi_i$ is a function of one coordinate of $\mathbf{z}_{\tau_k}$. Since $\varphi_{\tau_k}$ is invertible, $\det D\varphi_{\tau_k}\left(\mathbf{z}_{\tau_k}\right)\neq0$, so $\exists\sigma$ permutation of $\tau_k$ s.t. $\forall i\in\tau_k$, $\frac{\partial\varphi_{\sigma(i)}}{\partial z_i}\left(\mathbf{z}_{\tau_k}\right)\neq0$. Thus $\varphi_{\tau_k}\in\mathcal{S}_{\text{reparam}}$. $\qquad\square$

# C  A Counterexample for Thm. 4.2 *(ii)* when Asm. 4.1 is violated

One trivial case of violation of Asm. 4.1 is when $\mathbb{P}_i$ and $\widetilde{\mathbb{P}}_i$ are the same. In that case, the interventional regime $i$ is useless, namely, it does not constrain at all the indeterminacy set. Our counterexample is in a non-trivial case of violation, where the intervened mechanisms are not deterministically related, and do not share symmetries with the causal mechanisms.

We construct a counterexample for Thm. 4.2 *(ii)* when Asm. 4.1 is violated. The visualization of it is in Fig. 2. The counterexample is similar to the non-Gaussian case in the proof of Prop. 4.7.

Suppose that $d = 2$, $E(G) = \varnothing$ and that for all $i \in \{1, 2\}$, $\mathbb{P}_i$ has the same density $p_{a,b}$:

$$p_{a,b}(z) = \begin{cases} \exp(-az^2) & z < 0 \\ 1 & 0 \le z \le 1 - \frac{1}{2}(\sqrt{\frac{\pi}{a}} + \sqrt{\frac{\pi}{b}}) \\ \exp(-b(z - (1 - \frac{1}{2}(\sqrt{\frac{\pi}{a}} + \sqrt{\frac{\pi}{b}})))^2) & z > 1 - \frac{1}{2}(\sqrt{\frac{\pi}{a}} + \sqrt{\frac{\pi}{b}}) \end{cases} \qquad (35)$$

where $\sqrt{\frac{\pi}{a}} < 1$, $\sqrt{\frac{\pi}{b}} < 1$. One can verify that $p_a, p_b$ are smooth p.d.f.

Suppose that for all $i \in \{1, 2\}$, the intervened mechanism $\widetilde{\mathbb{P}}_i$ has the same density $p_{c,d}$, defined in the same way as (35), such that $\sqrt{\frac{\pi}{c}} < 1$, $\sqrt{\frac{\pi}{d}} < 1$, and $c, d \notin \{a, b\}$.

Set $\lambda := \min_{(x,y) \in \{(a,b),(c,d)\}} \left(1 - \frac{1}{2}\left(\sqrt{\frac{\pi}{x}} + \sqrt{\frac{\pi}{y}}\right)\right)$, then over $(0, \lambda)^2$ all the densities are constant, violating Asm. 4.1.

We construct a measure-preserving automorphism inspired by [19].

$$\varphi(\mathbf{z}) = \begin{cases} \mathbf{z} & \|\mathbf{z}\| \ge R \\ \begin{pmatrix} \cos(\alpha(\|\mathbf{z} - \mathbf{c}\| - R))z_1 - \sin(\alpha(\|\mathbf{z} - \mathbf{c}\| - R))z_2 \\ \cos(\alpha(\|\mathbf{z} - \mathbf{c}\| - R))z_2 + \sin(\alpha(\|\mathbf{z} - \mathbf{c}\| - R))z_1 \end{pmatrix} & \|\mathbf{z}\| < R \end{cases}$$

where $\mathbf{c} = (\frac{\lambda}{2}, \frac{\lambda}{2})$ denotes the center of rotation, $R \in (0, \frac{\lambda}{2}]$ denotes the radius of the disk, and $\alpha \ne 0$.

Now let us prove that $\varphi$ preserves $\mathbb{P}^i$ for all $i \in [\![0, 2]\!]$. By shifting $p_{a,b}$ by $-\mathbf{c}$, we only need to prove that the shifted $\varphi$ preserves the uniform distribution over $[-R, R]^2$. One can verify that $\varphi$ is a diffeomorphism over the 2-dimensional open disk $D^2(\mathbf{0}, R) \setminus \{\mathbf{0}\} \to D^2(\mathbf{0}, R) \setminus \{\mathbf{0}\}$ and $|\det(\varphi(\mathbf{z}))| = 1$. Thus $p_a(\mathbf{z}) = p_a(\varphi(\mathbf{z}))|\det(\varphi(\mathbf{z}))| \; \forall \mathbf{z} \in D^2(\mathbf{0}, R) \setminus \{\mathbf{0}\}$. Since $\varphi = Id$ outside of the disk, this change of variables formula holds almost everywhere in $\mathbb{R}^2$, thus $\mathbb{P}_i = \varphi_* \mathbb{P}_i$, $\widetilde{\mathbb{P}}_i = \varphi_* \widetilde{\mathbb{P}}_i$, which implies that $\varphi_* \mathbb{P}^i = \mathbb{P}^i \; \forall i \in [\![0, 2]\!]$.

Thus for all $\mathbf{f} \in \mathcal{F}$, $\mathbf{f}_* \mathbb{P}^i = (\mathbf{f} \circ \varphi^{-1})_* \mathbb{P}^i$. However, $(\mathbf{f} \circ \varphi^{-1})^{-1} \circ \mathbf{f} = \varphi$, which is not in $S_{\text{reparam}}$ or $S_{\text{scaling}}$.

*Remark* C.1. The above example can be easily generalized to any $p_{a,b}$ such that the constants on the plateau are different between $\mathbb{P}_i$ and $\widetilde{\mathbb{P}}_i$ and the domains of the plateau intersect on a nonzero measure set. For $d > 2$, the above example can be generalized by constructing the same $\mathbb{P}_i$ and $\widetilde{\mathbb{P}}^i$ for $i = 1, 2$, and for $i > 2$ we fix any $\mathbb{P}_i$ and $\widetilde{\mathbb{P}}_i$ verifying Asm. 4.1. Then one spurious solution $\varphi$ is as follows: let $\varphi_{[\![1,2]\!]}$ be the same measure-preserving automorphism as in the previous counterexample, and $\varphi_j = Id$ for $j > 2$.

*Remark* C.2. In CauCA, we suppose the distributions are Markov to a given graph $G$, but not necessarily faithful to $G$. This implies that independent components are in $\mathcal{P}_G$ no matter which graph $G$ is supposed given. Therefore, as long as Asm. 4.1 is not assumed, this counterexample applies to any CauCA model $(G, \mathcal{F}, \mathcal{P})$ with nonlinear $\mathcal{F}$ and nonparametric $\mathcal{P}$.

# D  Identifiability by structure-preserving stochastic interventions

In this section, we extend the result of Thm. 4.2 *(i)* to the case when we have access to a higher number of imperfect interventions. Here we focus on one special case of imperfect interventions, *structure-preserving interventions*, i.e., the interventions that do not change the parent set.

**Proposition D.1.** *For CauCA in $(G, \mathcal{F}, \mathcal{P}_G)$ assume the assumptions in Thm. 4.2 (i) hold. Fix any $i \in [d]$ such that $pa(i) \neq anc(i)$, $pa(i) \neq \varnothing$, and define $n_i := |\overline{pa}(i)|$. If there are $n_i(n_i + 1)$ structure-preserving interventions on node $i$ such that the variability assumption $V^i$ holds, then CauCA in $(G, \mathcal{F}, \mathcal{P}_G)$ is identifiable up to*

$$
\mathcal{S}_{\overline{G}_i} = \left\{ \mathbf{h} \in \mathcal{C}^1(\mathbb{R}^d) : \mathbf{h}(\mathbf{z}) = \big(h_j(\mathbf{z}_{\overline{anc}(j)})\big)_{j \in [d]} \mid h_j(\mathbf{z}_{\overline{anc}(j)}) = h_j(\mathbf{z}_{\overline{pa}(i)}) \, \forall j \in \overline{pa}(i) \right\} .
$$

*Namely, for all $\varphi \in \mathcal{S}_{\overline{G}_i}$, for all the nodes $j \in \overline{pa}(i)$, the reconstructed $Z_j$ can at most be a mixture of variables corresponding to the nodes in the closure of parents of $i$, instead of the closure of ancestors of $j$.*

*The variability assumption $V^i$ means*

$$
\begin{pmatrix} \mathbf{A}_i^1(\mathbf{z}) & \mathbf{B}_i^1(\mathbf{z}) \\ \vdots & \vdots \\ \mathbf{A}_i^{n_i(n_i+1)}(\mathbf{z}) & \mathbf{B}_i^{n_i(n_i+1)}(\mathbf{z}) \end{pmatrix} \in \mathbb{R}^{n_i(n_i+1) \times n_i(n_i+1)}
$$

*is invertible, where the symbols are defined as follows:*

$$
\mathbf{A}_i^t(\mathbf{z}) = \begin{pmatrix} \frac{\partial \left( g_i^{s_1,t} - h_i^{s_1} \right)}{\partial x_{r_1}} \big( \varphi_i(\mathbf{z}) \mid \boldsymbol{\varphi}_{pa(i)}(\mathbf{z}) \big) \\ \vdots \\ \frac{\partial \left( g_i^{s_k,t} - h_i^{s_k} \right)}{\partial x_{r_m}} \big( \varphi_i(\mathbf{z}) \mid \boldsymbol{\varphi}_{pa(i)}(\mathbf{z}) \big) \\ \vdots \\ \frac{\partial \left( g_i^{s_{n_i},t} - h_i^{s_{n_i}} \right)}{\partial x_{r_{n_i}}} \big( \varphi_i(\mathbf{z}) \mid \boldsymbol{\varphi}_{pa(i)}(\mathbf{z}) \big) \end{pmatrix}^{\top} \in \mathbb{R}^{1 \times n_i^2},
$$

$$
\mathbf{B}_i^t(\mathbf{z}) = \begin{pmatrix} \big( g_i^{s_1,t} - h_i^{s_1} \big) \big( \varphi_i(\mathbf{z}) \mid \boldsymbol{\varphi}_{pa(i)}(\mathbf{z}) \big) \\ \vdots \\ \big( g_i^{s_{n_i},t} - h_i^{s_{n_i}} \big) \big( \varphi_i(\mathbf{z}) \mid \boldsymbol{\varphi}_{pa(i)}(\mathbf{z}) \big) \end{pmatrix}^{\top} \in \mathbb{R}^{1 \times n_i}
$$

*where $r_k, s_k$ are the $k$-th variable in $\overline{pa}(i)$, and*

$$
g_i^{k,t}\big( z_i \mid \mathbf{z}_{pa(i)} \big) := \frac{\frac{\partial \tilde{q}_i^t}{\partial z_k}\big( z_i \mid \mathbf{z}_{pa(i)} \big)}{\tilde{q}_i^t\big( z_i \mid \mathbf{z}_{pa(i)} \big)}, \quad h_i^k\big( z_i \mid \mathbf{z}_{pa(i)} \big) := \frac{\frac{\partial q_i}{\partial z_k}\big( z_i \mid \mathbf{z}_{pa(i)} \big)}{q_i\big( z_i \mid \mathbf{z}_{pa(i)} \big)}
$$

*where $\tilde{q}_i^t$ denotes the intervened mechanism in $t$-th interventional regime that has the interventional target $i$, $q_i$ denotes the causal mechanism on $Z_i$.*

*Proof.* Based on the assumption of Thm. 4.2(i), reuse the proof of Thm. 4.2(i) from the equation (21):

$$
\ln \tilde{p}_i^t\big( z_i \mid \mathbf{z}_{\mathrm{pa}^i(i)} \big) - \ln p_i\big( z_i \mid \mathbf{z}_{\mathrm{pa}(i)} \big) = \ln \tilde{q}_i^t\big( \varphi_i(\mathbf{z}) \mid \boldsymbol{\varphi}_{\mathrm{pa}^i(i)}(\mathbf{z}) \big) - \ln q_i\big( \varphi_i(\mathbf{z}) \mid \boldsymbol{\varphi}_{\mathrm{pa}(i)}(\mathbf{z}) \big)
$$

where $\tilde{p}_i^t, \tilde{q}_i^t$ denote the intervened mechanism in $t$-th interventional regime that has the interventional target $i$. Notice that by the assumption of structure-preserving interventions, $\mathrm{pa}^i(i) = \mathrm{pa}(i)$.

Thm. 4.2(i) has already concluded that $\partial_j \varphi_i$ is constant 0 for all $j \notin \mathrm{anc}(i)$. Now we are interested in $\partial_j \varphi_i \, \forall j \in \mathrm{anc}(i)$. Take the partial derivative over $z_j$ with $j \in \mathrm{anc}(i) \setminus \mathrm{pa}(i)$ (non-empty by assumption):

$$
0 = \frac{\sum_{k \in \overline{\mathrm{pa}}(i)} \frac{\partial \tilde{q}_i^t}{\partial x_k}\big( \varphi_i(\mathbf{z}) \mid \boldsymbol{\varphi}_{\mathrm{pa}(i)}(\mathbf{z}) \big) \frac{\partial \varphi_k}{\partial z_j}(\mathbf{z})}{\tilde{q}_i^t\big( \varphi_i(\mathbf{z}) \mid \boldsymbol{\varphi}_{\mathrm{pa}(i)}(\mathbf{z}) \big)} - \frac{\sum_{k \in \overline{\mathrm{pa}}(i)} \frac{\partial q_i}{\partial x_k}\big( \varphi_i(\mathbf{z}) \mid \boldsymbol{\varphi}_{\mathrm{pa}(i)}(\mathbf{z}) \big) \frac{\partial \varphi_k}{\partial z_j}(\mathbf{z})}{q_i\big( \varphi_i(\mathbf{z}) \mid \boldsymbol{\varphi}_{\mathrm{pa}(i)}(\mathbf{z}) \big)} \tag{36}
$$

Recall that we define

$$g_i^{k,t}\left(z_i \mid \mathbf{z}_{\mathrm{pa}(i)}\right) := \frac{\frac{\partial \widetilde{q}_i^t}{\partial z_k}\left(z_i \mid \mathbf{z}_{\mathrm{pa}(i)}\right)}{\widetilde{q}_i^t\left(z_i \mid \mathbf{z}_{\mathrm{pa}(i)}\right)}, \quad h_i^k\left(z_i \mid \mathbf{z}_{\mathrm{pa}(i)}\right) := \frac{\frac{\partial q_i}{\partial z_k}\left(z_i \mid \mathbf{z}_{\mathrm{pa}(i)}\right)}{q_i\left(z_i \mid \mathbf{z}_{\mathrm{pa}(i)}\right)}$$

So (36) is rewritten as

$$0 = \sum_{k \in \overline{\mathrm{pa}}(i)} \left[ g_i^{k,t}\left(\varphi_i(\mathbf{z}) \mid \boldsymbol{\varphi}_{\mathrm{pa}(i)}(\mathbf{z})\right) - h_i^k\left(\varphi_i(\mathbf{z}) \mid \boldsymbol{\varphi}_{\mathrm{pa}(i)}(\mathbf{z})\right) \right] \partial_j \varphi_k(\mathbf{z})$$

Choose any $l \in \mathrm{pa}(i)$ (non-empty by assumption). Take the partial derivative of $z_l$ on two sides:

$$0 = \sum_{k \in \overline{\mathrm{pa}}(i)} \Bigg[ \sum_{m \in \overline{\mathrm{pa}}(i)} \partial_m \left( g_i^{k,t} - h_i^k \right) \left(\varphi_i(\mathbf{z}) \mid \boldsymbol{\varphi}_{\mathrm{pa}(i)}(\mathbf{z})\right) \partial_j \varphi_k(\mathbf{z}) \partial_l \varphi_m(\mathbf{z}) \Bigg]$$
$$+ \left( g_i^{k,t} - h_i^k \right) \left(\varphi_i(\mathbf{z}) \mid \boldsymbol{\varphi}_{\mathrm{pa}(i)}(\mathbf{z})\right) \partial_l \partial_j \varphi_k(\mathbf{z})$$

which can be rewritten as

$$0 = \mathbf{A}_i^t(\mathbf{z}) \mathbf{a}_{j,l}^i(\mathbf{z}) + \mathbf{B}_i^t(\mathbf{z}) \mathbf{b}_{j,l}^i(\mathbf{z}) \tag{37}$$

where $\mathbf{a}_{j,l}^i(\mathbf{z}) = \begin{pmatrix} \partial_j \varphi_{p_1}(\mathbf{z}) \partial_l \varphi_{p_1}(\mathbf{z}) \\ \vdots \\ \partial_j \varphi_{p_k}(\mathbf{z}) \partial_l \varphi_{p_m}(\mathbf{z}) \\ \vdots \\ \partial_j \varphi_{p_{n_i}}(\mathbf{z}) \partial_l \varphi_{p_{n_i}}(\mathbf{z}) \end{pmatrix} \in \mathbb{R}^{n_i^2}, \quad \mathbf{b}_{j,l}^i(\mathbf{z}) = \begin{pmatrix} \partial_l \partial_j \varphi_{p_1}(\mathbf{z}) \\ \vdots \\ \partial_l \partial_j \varphi_{p_{n_i}}(\mathbf{z}) \end{pmatrix} \in \mathbb{R}^{n_i},$

Collect $n_i\left(n_i + 1\right)$ equations, every one corresponding to one interventional regime, in the form of (37) for all $t \in [n_i\left(n_i + 1\right)]$:

$$\mathbf{0} = \mathbf{M}_i(\mathbf{z}) \begin{pmatrix} \mathbf{a}_{j,l}^i(\mathbf{z}) \\ \mathbf{b}_{j,l}^i(\mathbf{z}) \end{pmatrix} \tag{38}$$

where

$$\mathbf{M}_i(\mathbf{z}) := \begin{pmatrix} \mathbf{A}_i^1(\mathbf{z}) & \mathbf{B}_i^1(\mathbf{z}) \\ \vdots & \\ \mathbf{A}_i^{n_i(n_i+1)}(\mathbf{z}) & \mathbf{B}_i^{n_i(n_i+1)}(\mathbf{z}) \end{pmatrix} \in \mathbb{R}^{n_i(n_{i+1}) \times n_i(n_{i+1})} \tag{39}$$

By assumption of variability $V^i$, $\mathbf{M}_i(\mathbf{z})$ is invertible for all $\mathbf{z} \in \mathbb{R}^d$. Thus (38) has a unique solution, which is $\mathbf{a}_{j,l}^i = \mathbf{0}$, $\mathbf{b}_{j,l}^i = \mathbf{0}$.

$\mathbf{a}_{j,l}^i(\mathbf{z}) = \mathbf{0}$ implies $\forall k, m \in \overline{\mathrm{pa}}(i), \partial_j \varphi_k(\mathbf{z}) \partial_l \varphi_m(\mathbf{z}) = 0$.

Since $l \in pa(i)$, $\partial_l \varphi_l(\mathbf{z}) \neq 0$ is in $\mathbf{a}_{j,l}^i(\mathbf{z})$, so $\forall k \in \overline{\mathrm{pa}}(i)$, $\partial_j \varphi_k(\mathbf{z}) \partial_l \varphi_l(\mathbf{z}) = 0$, which implies $\partial_j \varphi_k(\mathbf{z}) = 0$. We have proven that for all $j \in \mathrm{anc}(i) \setminus \mathrm{pa}(i)$, for all $k \in \overline{\mathrm{pa}}(i)$, for all $\mathbf{z} \in \mathbb{R}^d$, $\partial_j \varphi_k(\mathbf{z}) = 0$. Namely, $\varphi_k$ only depends on $\overline{\mathrm{pa}}(i)$. Combining with the result in Thm. 4.2(i), we obtain the conclusion. □

# E Known vs. unknown intervention targets

In the main paper, for simplicity, we only provided the minimal set of notation required for describing the problem of CauCA with *known intervention targets*. However, we believe that a more general version of CauCA should also be considered for cases where the targets are unknown. In fact, in the following, we will distinguish many problem settings, ranging from *totally known targets* to *totally unknown targets*. Each setting may be more or less suited to model a collection of datasets, depending on the amount and kind of prior knowledge available.

In the following, we provide a general framework in which CauCA with unknown intervention targets can be rigorously formulated. Note that $G$ denotes a DAG such that $V(G) = [d]$, indexed by natural numbers, which correspond to the indices of targets $\tau_i$ of interventions. G denotes instead a DAG

equipped only with a *partial* order induced by the arrows, namely, $V(\mathsf{G})$ is a set not necessarily indexed by natural numbers. For example, consider $V(\mathsf{G}) = \{\text{"cloudy", "sprinkle", "raining", "wet grass"}\}$. In this case, the probability distribution $\mathbb{P}$ that is Markov relative to this graph is not defined because of the lack of indices: $\mathbb{P}_1$ might denote the marginal of "cloudy", "sprinkle", "raining" or "wet grass". However, $\mathbb{P}$ is Markov relative to an *indexed* DAG of $\mathsf{G}$, denoted $G \models \mathsf{G}$, which denotes that there exists a bijection $\sigma$ s.t. $(u, v) \in E(G)$ iff $(\sigma(u), \sigma(v)) \in E(\mathsf{G})$.

**Definition E.1.** *Given two DAGs $G \models \mathsf{G}$ and $G' \models \mathsf{G}$, an* isomorphism *from $G$ to $G'$ is a bijection $\sigma$ of $V(G) = [d]$ such that $(i, j) \in E(G)$ if and only if $(\sigma(i), \sigma(j)) \in E(G')$. An* automorphism *of $G$ is an isomorphism $G \to G$.*

**Definition E.2** (Identifiabilities of Causal component analysis, general setting)**.** *Given $\mathsf{G}$ a partially ordered DAG, $\mathcal{F}$ a class of diffeomorphisms, $\mathcal{P}_\mathsf{G}$ a set of distributions such that for every $\mathbb{P} \in \mathcal{P}_\mathsf{G}$ there exists $G \models \mathsf{G}$ such that $\mathbb{P} \in \mathcal{P}_G$, we define $(\mathsf{G}, \mathcal{F}, \mathcal{P}_\mathsf{G})$ as a class of latent CBN $(G, \mathbf{f}, (\mathbb{P}^i, \tau_i)_{i \in [\![0, K]\!]})$ such that $G \models \mathsf{G}$, $f \in \mathcal{F}$ and $\mathbb{P} \in \mathcal{P}_G$.*

*(i) We define CauCA with* known intervention targets *in $(\mathsf{G}, \mathcal{F}, \mathcal{P}_\mathsf{G})$ as a class of latent CBN models such that all latent CBN models have the same $G$ and $(\tau_i)_{i \in [\![0, K]\!]}$.*

*(ii) We define CauCA with* known intervention targets up to graph automorphisms *in $(\mathsf{G}, \mathcal{F}, \mathcal{P}_\mathsf{G})$ as a class of latent CBN models such that for any two latent CBN models $(G, \mathbf{f}, (\mathbb{P}^i, \tau_i)_{i \in [\![0, K]\!]})$ and $(G, \mathbf{f}', (\mathbb{P}^i, \tau_i')_{i \in [\![0, K]\!]})$ in the class, there exists $\sigma$ an automorphism of $G$ such that $\tau_i' = \sigma(\tau_i)$ for all $i \in [K]$.*

*(iii) We define CauCA with* matched intervention targets *in $(\mathsf{G}, \mathcal{F}, \mathcal{P}_\mathsf{G})$ as a class of latent CBN models such that for any two latent CBN models $(G, \mathbf{f}, (\mathbb{P}^i, \tau_i)_{i \in [\![0, K]\!]})$ and $(G', \mathbf{f}', (\mathbb{P}^i, \tau_i')_{i \in [\![0, K]\!]})$ in the class with $G, G' \models \mathsf{G}$, there exists a permutation $\pi \in \mathfrak{S}_d : V(G) \to V(G')$ such that $\tau_i' = \pi(\tau_i)$ for all $i \in [K]$.*

*(iv) We define CauCA with* unknown intervention targets *in $(\mathsf{G}, \mathcal{F}, \mathcal{P}_\mathsf{G})$ as a class of latent CBN models that contains all $(G, \mathbf{f}, (\mathbb{P}^i, \tau_i)_{i \in [\![0, K]\!]})$ such that $G \models \mathsf{G}$, $f \in \mathcal{F}$ and $\mathbb{P} \in \mathcal{P}_G$.*

*We say that the CauCA in $(\mathsf{G}, \mathcal{F}, \mathcal{P}_\mathsf{G})$[6] is* identifiable *up to $\mathcal{S}$ if for any $(G, \mathbf{f}, (\mathbb{P}^i, \tau_i)_{i \in [\![0, K]\!]})$ and $(G', \mathbf{f}', (\mathbb{Q}^i, \tau_i')_{i \in [\![0, K]\!]})$ in $(\mathsf{G}, \mathcal{F}, \mathcal{P}_\mathsf{G})$, the relation $\mathbf{f}_* \mathbb{P}^i = \mathbf{f}'_* \mathbb{Q}^i \; \forall i \in [\![0, K]\!]$ implies that there is $\mathbf{h} \in \mathcal{S}$ such that $\mathbf{h} = \mathbf{f}'^{-1} \circ \mathbf{f}$ on the support of $\mathbb{P}$.*

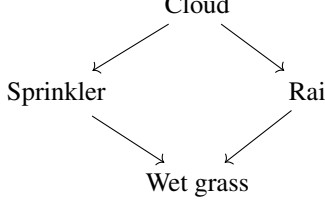

| Dataset | $\tau$ | $\tau'$(i) | $\tau'$(ii) | $\tau'$(iii) | $\tau'$(iv) |
|---------|--------|-----------|------------|-------------|-------------|
| $\mathcal{D}_1$ | C | C | C | S | S |
| $\mathcal{D}_2$ | C | C | C | S | R |
| $\mathcal{D}_3$ | S | S | R | W | W |
| $\mathcal{D}_4$ | R | R | S | C | W |
| $\mathcal{D}_5$ | W | W | W | R | W |

Figure 5: Representative cases for the CauCA settings described in Defn. E.2 *(i)-(iv)*. Each row corresponds to a dataset where one perfect intervention is performed on one of the targets: Cloud (C), Sprinkler (S), Rain (R), and Wet grass (W). Each column corresponds to an admissible choice of intervention targets within each of the settings: the discrepancies between the intervention targets $\tau'$(i)-$\tau'$(iv) and the ground truth targets $\tau$ are meant to illustrate different degrees of ignorance about $\tau$ across the settings in Defn. E.2 *(i)-(iv)*. In the *known intervention targets* setting, which we focused on in the main paper, $\tau'$(i) is the only possible choice: i.e., the targets must be perfectly aligned with the ground truth $\tau$. For the settings with *known intervention targets up to graph automorphisms* and with *matched intervention targets*, $\tau'$(ii) and $\tau'$(iii) represent admissible choices: identifiability results can be proved for both settings. We also show that in the setting of *totally unknown targets*, where $\tau'$(iv) is an admissible choice, CauCA is not identifiable (see Remark E.9).

*Remark* E.3. In this general framework, the dataset $\mathcal{D}_i := \left( \{\mathbf{x}^{(j)}\}_{j=1}^{N_i} \right), T_i$ denotes the nodes in $V(\mathsf{G})$ ("cloudy", "sprinkler" etc) instead of nodes $\tau_i \subset [d]$ in $V(G)$ (2).

---

[6]Or $(G, \mathcal{F}, \mathcal{P}_G)$ for CauCA with known intervention targets.

If all $d$ variables are included in the known targets, then there exists a unique bijection $\sigma : V(G) \to V(\mathsf{G})$. This implies that for any latent CBN $(G, \mathbf{f}, (\mathbb{P}^i, \tau_i)_{i \in [\![0,K]\!]})$ in $(G, \mathcal{F}, \mathcal{P}_G)$, $\tau_i = \sigma^{-1}(T_i)$ i.e. the targets $\tau_i$ are uniquely defined by $T_i$ in each interventional regime. In this case, without loss of generality, we can suppose that $\forall i \in V(G)$, if $i \in \mathrm{pa}(j)$, then $i < j$. This can be achieved by rearranging the nodes in the graph and, correspondingly, the coordinates of $(\mathbb{P}^k, \tau_k) \, \forall k \in [\![0,K]\!]$. When the intervention targets are totally unknown, $(\mathsf{G}, \mathcal{F}, \mathcal{P}_G)$ only gives us the information of an unordered graph $\mathsf{G}$, and in every interventional regime the candidate latent CBN models that achieve the same likelihood might intervene on totally different variables. In this case, we cannot rearrange the nodes without loss of generality.

Our main paper has shown identifiability results about Defn. E.2 *(i)*. In the following, we generalize Thm. 4.2 to CauCA with known intervention targets up to graph automorphisms. We also generalize Prop. 4.6 to matched intervention targets in ICA setting. We also prove in Corollary E.8 that CauCA with unknown intervention targets is not identifiable. An open question is whether CauCA with a nontrivial graph is identifiable with matched intervention targets.

**Assumption E.4** (Interventional discrepancy, general version). *Given* $k \in [K]$, *we say that a stochastic intervention* $\tilde{p}_{\tau_k}$ *satisfies general interventional discrepancy if for all* $i \in [d]$,

$$\frac{\partial (\ln p_i)}{\partial z_i} \left( z_i \mid \mathbf{z}_{pa(i)} \right) \neq \frac{\partial (\ln \widetilde{p}_{\tau_k})}{\partial z_{\tau_k}} \left( z_{\tau_k} \mid \mathbf{z}_{pa^k(\tau_k)} \right) \quad \text{almost everywhere (a.e.).} \tag{40}$$

**Theorem E.5.** *For CauCA with known intervention targets up to graph automorphisms in* $(G, \mathcal{F}, \mathcal{P}_G)$,

*(i) Suppose for each node in $[d]$, there is one (perfect or structure-preserving) stochastic intervention such that Asm. E.4 is satisfied. Then CauCA in $(G, \mathcal{F}, \mathcal{P}_G)$ is identifiable up to*

$$\mathcal{S}_{Aut\bar{G}} = \big\{ \mathbf{P}_\sigma \circ \mathbf{h} : \mathbb{R}^d \to \mathbb{R}^d | \sigma \in Aut_G, \mathbf{P}_\sigma \text{ permutation matrix of } \sigma,$$
$$\mathbf{h}(\mathbf{z}) = \big( h_i(\mathbf{z}_{\overline{anc}(i)}) \big)_{i \in [d]}, \mathbf{h} \text{ is } \mathcal{C}^1\text{-diffeomorphism} \big\}.$$

*(ii) Suppose for each node $i$ in $[d]$, there is one perfect stochastic intervention such that Asm. E.4 is satisfied, then CauCA in $(G, \mathcal{F}, \mathcal{P}_G)$ is identifiable up to*

$$S_{G\text{-scaling}} := \big\{ \mathbf{P}_\sigma \circ \mathbf{h} | \sigma \in Aut_G, \mathbf{P}_\sigma \text{ permutation matrix of } \sigma, \mathbf{h} : \mathbb{R}^d \to \mathbb{R}^d,$$
$$\mathbf{h}(\mathbf{z}) = (h_1(z_1), \dots, h_d(z_d)) \text{ for some } h_i \in \mathcal{C}^1(\mathbb{R}, \mathbb{R}) \text{ with } |h_i'(\mathbf{z})| > 0 \quad \forall \mathbf{z} \in \mathbb{R}^d \big\}.$$

*Proof.* **Proof of *(i)*:** The proof is based on the proof of Thm. 4.2*(i)*. Consider two latent models achieving the same likelihood across all interventional regimes: $\big(G, \mathbf{f}, (\mathbb{P}^i, \tau_i)_{i \in [\![0,d]\!]}\big)$ and $\big(G, \mathbf{f}', (\mathbb{Q}^i, \tau_i')_{i \in [\![0,d]\!]}\big)$. By the definition of latent CBN models with known targets up to graph automorphisms, there exists $\sigma$ automorphism of $G$ s.t. the targets $(\tau_i')_{i \in [d]} = (\sigma(1), \cdots, \sigma(d))$. Since $(\tau_i)_{i \in [d]}$ covers all $d$ nodes in $G$, by rearranging $(\tau_i)_{i \in [d]}$ we can suppose without loss of generality that $\tau_i = i \, \forall i \in [d]$. Namely, in the $i$-th interventional regime,

$$\mathbb{P}^i = \widetilde{\mathbb{P}}_i \big( Z_j \mid \mathbf{Z}_{\mathrm{pa}^i(j)} \big) \prod_{j \neq i} \mathbb{P}_{\big( Z_j | \mathbf{Z}_{\mathrm{pa}(j)} \big)}, \quad \mathbb{Q}^i = \widetilde{\mathbb{Q}}_{\sigma(i)} \big( Z_{\sigma(i)} \mid \mathbf{Z}_{\mathrm{pa}^{\sigma(i)}(\sigma(i))} \big) \prod_{j \neq i} \mathbb{Q}_{\big( Z_{\sigma(j)} | \mathbf{Z}_{\mathrm{pa}(\sigma(j))} \big)}.$$

Define $\boldsymbol{\varphi} := \mathbf{f}'^{-1} \circ \mathbf{f}$. Denote its $i$-th dimension output function as $\varphi_i : \mathbb{R}^d \to \mathbb{R}$. We will prove by induction that $\forall i \in [d], \forall j \notin \overline{\mathrm{anc}}(i), \forall \mathbf{z} \in \mathbb{R}^d, \frac{\partial \varphi_{\sigma(i)}}{\partial z_j}(\mathbf{z}) = 0$.

For any $i \in [\![0,d]\!]$, $\mathbf{f}_* \mathbb{P}^i = \mathbf{f}'_* \mathbb{Q}^i$. Since $\boldsymbol{\varphi}$ is a diffeomorphism, by the change of variables formula,

$$p^i(\mathbf{z}) = q^i(\boldsymbol{\varphi}(\mathbf{z})) |\det \boldsymbol{\varphi}(\mathbf{z})|. \tag{41}$$

For $i = 0$, factorize $p_i$ and $q^i$ according to $G$, then take the logarithm on both sides:

$$\sum_{j=1}^{d} \ln p_j \big( z_j \mid \mathbf{z}_{\mathrm{pa}(j)} \big) = \sum_{j=1}^{d} \ln q_j \big( \varphi_j(\mathbf{z}) \mid \boldsymbol{\varphi}_{(\mathrm{pa}(j))}(\mathbf{z}) \big) + \ln |\det D\boldsymbol{\varphi}(\mathbf{z})|. \tag{42}$$

For $i = 1$, $\widetilde{\mathbb{Q}}_1$ has no conditionals, and so does $Z_{\sigma(1)}$. Thus $q^i$ is factorized as

$$q^1(\mathbf{z}) = \widetilde{q}_{\sigma(1)}\left(z_{\sigma(1)}\right) \prod_{j \neq \sigma(1)} q_j\left(z_j \mid \mathbf{z}_{\mathrm{pa}(j)}\right).$$

So the equation (41) for $i = 1$ after taking logarithm is

$$
\ln \widetilde{p}_1\left(z_1\right) + \sum_{j \neq 1} \ln p_j\left(z_j \mid \mathbf{z}_{\mathrm{pa}(j)}\right) = \ln \widetilde{q}_{\sigma(1)}\left(\varphi_{\sigma(1)}(\mathbf{z}) \mid \boldsymbol{\varphi}_{\mathrm{pa}^{\sigma(1)}(\sigma(1))}(\mathbf{z})\right)
$$
$$
+ \sum_{j \neq \sigma(1)} q_j\left(\varphi_j(\mathbf{z}) \mid \boldsymbol{\varphi}_{\mathrm{pa}(j)}(\mathbf{z})\right) + \ln\left|\det D\boldsymbol{\varphi}(\mathbf{z})\right|. \tag{43}
$$

Subtract (43) by (42),

$$
\ln \tilde{p}_1\left(z_1\right) - \ln p_1\left(z_1\right) = \ln \tilde{q}_{\sigma(1)}\left(\varphi_{\sigma(1)}(\mathbf{z})\right) - \ln q_{\sigma(1)}\left(\varphi_{\sigma(1)}(\mathbf{z})\right). \tag{44}
$$

For any $i \neq 1$, take the $i$-th partial derivative of both sides:

$$
0 = \left[\frac{\widetilde{q}'_{\sigma(1)}\left(\varphi_{\sigma(1)}(\mathbf{z})\right)}{\widetilde{q}_{\sigma(1)}\left(\varphi_{\sigma(1)}(\mathbf{z})\right)} - \frac{q'_{\sigma(1)}\left(\varphi_{\sigma(1)}(\mathbf{z})\right)}{q_{\sigma(1)}\left(\varphi_{\sigma(1)}(\mathbf{z})\right)}\right] \frac{\partial \varphi_{\sigma(1)}}{\partial z_i}(\mathbf{z}).
$$

By Asm. E.4, the term in the parenthesis is non-zero a.e. in $\mathbb{R}^d$. Thus $\frac{\partial \varphi_{\sigma(1)}}{\partial z_i}(\mathbf{z}) = 0$ a.e. in $\mathbb{R}^d$. Since $\boldsymbol{\varphi} = \mathbf{f}'^{-1} \circ \mathbf{f}$ where $\mathbf{f}, \mathbf{f}'$ are $\mathcal{C}^1$-diffeomorphisms, so is $\boldsymbol{\varphi}$. $\frac{\partial \varphi_{\sigma(1)}}{\partial z_i}$ is continuous and thus equals zero everywhere.

Now suppose $\forall k \in [i-1], \forall j \notin \overline{\mathrm{anc}}(k), \frac{\partial \varphi_{\sigma(k)}}{\partial z_j}(\mathbf{z}) = 0, \forall \mathbf{z} \in \mathbb{R}^d$. Then for interventional regime $i$,

$$
\ln \tilde{p}_i\left(z_i \mid \mathbf{z}_{\mathrm{pa}^i(i)}\right) + \sum_{j \neq i} \ln p_j\left(z_j \mid \mathbf{z}_{\mathrm{pa}^j(j)}\right) = \ln \tilde{q}_{\sigma(i)}\left(\varphi_{\sigma(i)}(\mathbf{z}) \mid \boldsymbol{\varphi}_{\mathrm{pa}^{\sigma(i)}(\sigma(i))}(\mathbf{z})\right)
$$
$$
+ \sum_{j \neq i} q_{\sigma(j)}\left(\varphi_{\sigma(j)}(\mathbf{z}) \mid \boldsymbol{\varphi}_{\mathrm{pa}(\sigma(i))}(\mathbf{z})\right) + \ln\left|\det D\boldsymbol{\varphi}(\mathbf{z})\right|,
$$

subtracted by (42),

$$
\ln \tilde{p}_i\left(z_i \mid \mathbf{z}_{\mathrm{pa}^i(i)}\right) - \ln p_i\left(z_i \mid \mathbf{z}_{\mathrm{pa}(i)}\right) = \ln \tilde{q}_{\sigma(i)}\left(\varphi_{\sigma(i)}(\mathbf{z}) \mid \boldsymbol{\varphi}_{\mathrm{pa}^{\sigma(i)}(\sigma(i))}(\mathbf{z})\right)
$$
$$
- \ln q_{\sigma(i)}\left(\varphi_{\sigma(i)}(\mathbf{z}) \mid \boldsymbol{\varphi}_{\mathrm{pa}(\sigma(i))}(\mathbf{z})\right).
$$

For any $j \notin \overline{\mathrm{anc}}(i), j \notin pa(i) \supset \mathrm{pa}^i(i)$ by assumption. Take partial derivative over $z_j$:

$$
0 = \frac{\sum_{k \in \overline{\mathrm{pa}}^{\sigma(i)}(\sigma(i))} \frac{\partial \widetilde{q}_{\sigma(i)}}{\partial \mathbf{x}_k}\left(\varphi_{\sigma(i)}(\mathbf{z}) \mid \boldsymbol{\varphi}_{\mathrm{pa}^{\sigma(i)}(\sigma(i))}(\mathbf{z})\right) \frac{\partial \varphi_k}{\partial z_j}(\mathbf{z})}{\widetilde{q}_{\sigma(i)}\left(\varphi_{\sigma(i)}(\mathbf{z}) \mid \boldsymbol{\varphi}_{\mathrm{pa}^{\sigma(i)}(\sigma(i))}(\mathbf{z})\right)}
$$
$$
- \frac{\sum_{k \in \overline{\mathrm{pa}}(\sigma(i))} \frac{\partial q_{\sigma(i)}}{\partial \mathbf{x}_k}\left(\varphi_{\sigma(i)}(\mathbf{z}) \mid \boldsymbol{\varphi}_{\mathrm{pa}(\sigma(i))}(\mathbf{z})\right) \frac{\partial \varphi_k}{\partial z_j}(\mathbf{z})}{q_{\sigma(i)}\left(\varphi_{\sigma(i)}(\mathbf{z}) \mid \boldsymbol{\varphi}_{\mathrm{pa}(\sigma(i))}(\mathbf{z})\right)}. \tag{45}
$$

Now prove that $G_{\sigma(i)} = \sigma(G_i)$:

$G_{\sigma(i)}$ is obtained by either a perfect stochastic or a structure-preserving stochastic intervention. For a structure-preserving stochastic intervention, $G = G_{\sigma(i)} = \sigma(G_i)$. For a perfect stochastic intervention, since $\tau'_i = \sigma(\tau_i)$, in $G_{\sigma(i)}$ only the arrows towards $\sigma(\tau_i)$ are deleted, which correspond to deleting arrows towards $\tau_i$ in $G_i$.

Thus $G_{\sigma(i)} = \sigma(G_i)$. Thus $\mathrm{pa}^{\sigma(i)}(\sigma(i)) = \mathrm{pa}^{\sigma(i)}(\sigma(i)) = \sigma(\mathrm{pa}^i(i))$, the last equality by the definition of automorphism $\sigma$, $\sigma(\mathrm{pa}(i)) = \mathrm{pa}(\sigma(i))$.

The assumption of induction says $\forall k \in [i-1]$ $\forall j \notin \overline{\mathrm{anc}}(k)$ $\frac{\partial \varphi_{\sigma(k)}}{\partial z_j}(\mathbf{z}) = 0$ $\forall \mathbf{z} \in \mathbb{R}^d$. For all $k \in \mathrm{pa}(i) \supset \mathrm{pa}^i(i)$, since $j \notin \overline{\mathrm{anc}}(i)$, $j \notin \overline{\mathrm{anc}}(k)$ as well. By induction, $\frac{\partial \varphi_{\sigma(k)}}{\partial z_j}(\mathbf{z}) = 0$. So the second sum of the right-hand side of (45) can be canceled except for $k = \sigma(i)$. Also, $\mathrm{pa}(\sigma(i)) = \sigma(\mathrm{pa}(i))$, $\mathrm{pa}^{\sigma(i)}(\sigma(i)) = \sigma(\mathrm{pa}^i(i))$. By the same assumption in the current induction, for all $l \in \mathrm{pa}^{\sigma(i)}(\sigma(i)) = \sigma(\mathrm{pa}^i(i))$, $\frac{\partial \varphi_l}{\partial z_j}(\mathbf{z}) = 0$. So the first sum of the right-hand side of (45) can be canceled except for $k = \sigma(i)$.

The equation rewrites after deleting the partial derivatives that are zero:

$$0 = \left[ \frac{\frac{\partial \widetilde{q}_{\sigma(i)}}{\partial \varphi_{\sigma(i)}} \left( \varphi_{\sigma(i)}(\mathbf{z}) \mid \boldsymbol{\varphi}_{\mathrm{pa}^{\sigma(i)}(\sigma(i))}(\mathbf{z}) \right)}{\widetilde{q}_{\sigma(i)} \left( \varphi_{\sigma(i)}(\mathbf{z}) \mid \boldsymbol{\varphi}_{\mathrm{pa}^{\sigma(i)}(\sigma(i))}(\mathbf{z}) \right)} - \frac{\frac{\partial q_{\sigma(i)}}{\partial \varphi_{\sigma(i)}} \left( \varphi_{\sigma(i)}(\mathbf{z}) \mid \boldsymbol{\varphi}_{\mathrm{pa}(\sigma(i))}(\mathbf{z}) \right)}{q_{\sigma(i)} \left( \varphi_{\sigma(i)}(\mathbf{z}) \mid \boldsymbol{\varphi}_{\mathrm{pa}(\sigma(i))}(\mathbf{z}) \right)} \right] \frac{\partial \varphi_{\sigma(i)}}{\partial z_j}(\mathbf{z}).$$

By Asm. E.4, the term in parenthesis is nonzero a.e., thus $\frac{\partial \varphi_{\sigma(i)}}{\partial z_j}(\mathbf{z}) = 0$ a.e. Since $\frac{\partial \varphi_{\sigma(i)}}{\partial z_j}$ is continuous, it equals zero everywhere.

The induction is finished when $i = d$. We have proven that $\forall i \in [d], \forall j \notin \overline{\mathrm{anc}}(i), \forall \mathbf{z} \in \mathbb{R}^d$, $\frac{\partial \varphi_{\sigma(i)}}{\partial z_j}(\mathbf{z}) = 0$, i.e. $\left( \mathbf{P}_\sigma^{-1} D\boldsymbol{\varphi}(\mathbf{z}) \right)_{ij} = (D\boldsymbol{\varphi}(\mathbf{z}))_{\sigma(i)j} = \frac{\partial \varphi_{\sigma(i)}}{\partial z_j}(\mathbf{z}) = 0$

Thus $\mathbf{P}_\sigma^{-1} \boldsymbol{\varphi} \in \mathcal{S}_{\bar{G}}$. $\boldsymbol{\varphi} \in \mathcal{S}_{\mathrm{Aut}\bar{G}}$.

**Proof of (ii)**:

By the result of (i), $\boldsymbol{\psi} := \mathbf{f}'^{-1} \circ \mathbf{f} \in \mathcal{S}_{\mathrm{Aut}\bar{G}}$, thus there exists a permutation matrix $\mathbf{P}_\sigma$ s.t. $\mathbf{P}_\sigma^{-1} \boldsymbol{\psi} \in \mathcal{S}_{\bar{G}}$ Denote $\boldsymbol{\varphi} = \mathbf{P}_\sigma^{-1} \boldsymbol{\psi}$. Apply Thm. 4.2(ii), we can prove that $\boldsymbol{\varphi} \in \mathcal{S}_{\mathrm{scaling}}$. Thus $\boldsymbol{\psi} = \mathbf{P}_\sigma \boldsymbol{\varphi} \in \mathcal{S}_{G\text{-scaling}}$. $\square$

**Proposition E.6.** *Suppose that $G$ is the empty graph and that there are $d-1$ variables intervened on, with one single target per dataset, satisfying Asm. E.4. Then ICA with matched intervention targets in $(G, \mathcal{F}, \mathcal{P}_G)$ with single-node interventions (Defn. E.2) is identifiable up to*

$$\mathcal{S}_{reparam} := \left\{ \mathbf{g} : \mathbb{R}^d \to \mathbb{R}^d | \mathbf{g} = \mathbf{P} \circ \mathbf{h} \text{ where } P \text{ is a permutation matrix and } \mathbf{h} \in \mathcal{S}_{scaling} \right\}.$$

*Proof.* For an empty graph $G$, $\mathrm{Aut}_G = \mathfrak{S}_d$. Thus ICA with matched intervention targets is the same as known intervention targets up to graph automorphisms. Consider two latent models achieving the same likelihood across all interventional regimes: $\left(G, \mathbf{f}, \left(\mathbb{P}^i, \tau_i\right)_{i \in [\![0,d]\!]}\right)$ and $\left(G, \mathbf{f}', \left(\mathbb{Q}^i, \tau_i'\right)_{i \in [\![0,d]\!]}\right)$. By the definition of latent CBN models with known targets up to graph automorphisms, there exists $\sigma$ automorphism of $G$ s.t. the targets $(\tau_i')_{i \in [d]} = (\sigma(\tau_1), \cdots, \sigma(\tau_d))$. Apply Prop. 4.6 to $\left(G, \mathbf{f}, \left(\mathbb{P}^i, \tau_i\right)_{i \in [\![0,d-1]\!]}\right)$ and $\left(G, \mathbf{f}' \circ \mathbf{P}_\sigma, \left(\mathbb{Q}^{\sigma^{-1}(i)}, \tau_i\right)_{i \in [\![0,d-1]\!]}\right)$, then $\boldsymbol{\varphi} := (\mathbf{f}' \circ \mathbf{P}_\sigma)^{-1} \circ \mathbf{f} \in \mathcal{S}_{\mathrm{scaling}}$. Then for $\left(G, \mathbf{f}, \left(\mathbb{P}^i, \tau_i\right)_{i \in [\![0,d-1]\!]}\right)$ and $\left(G, \mathbf{f}', \left(\mathbb{Q}^i, \tau_i'\right)_{i \in [\![0,d-1]\!]}\right)$, $\mathbf{f}'^{-1} \circ \mathbf{f} = \mathbf{P}_\sigma \circ \boldsymbol{\varphi} \in \mathcal{S}_{\mathrm{reparam}}$. $\square$

*Remark* E.7. The setting of CauCA with matched intervention targets is similar to one scenario studied in causal representation learning. See, e.g., von Kügelgen et al. [55, Thm. 3.4]: the constraint on intervention targets expressed in their Asm. (A2') can be rephrased within our framework as the requirement that for any two latent CBN models, there exists a permutation $\pi : V(G) \to V(G')$ such that the intervention targets $\tau_{i,1}' = \tau_{i,2}' = \pi(\tau_{i,1}) = \pi(\tau_{i,2})$, where $\tau_{i,1}$ denotes the unknown target of the $(i, 1)$-indexed interventional regime. Subsequent work by Varıcı et al. [53] also studied the setting with coupled environments.[7]

---

[7]Varıcı et al. [53] additionally introduced a setting with *uncoupled environments* [53, Def. 2], relaxing the requirement of matched interventions. Note that the setting with uncoupled environments assumes strictly more information than Defn. E.2 *(iv)*, since it requires that for each latent CBN model $(G, \mathbf{f}, (\mathbb{P}^i, \tau_i)_{i \in [\![0,2d]\!]})$, for each node $j \in V(G)$, there exist $k, l \in [\![1, 2d]\!]$ such that $\tau_k = \tau_l = \{j\}$.

**Corollary E.8.** *[Corollary of Prop. 4.7] Given a DAG* $\mathsf{G}$, *CauCA with unknown intervention targets is not identifiable in* $(\mathsf{G}, \mathcal{F}, \mathcal{P}_\mathsf{G})$ *up to* $\mathcal{S}_{reparam}$.

*Proof.* Fix any $G \models \mathsf{G}$ and $\mathbf{f}$. Without loss of generality, suppose $K > d-2$. By Prop. 4.7, there exist $\left(G, \mathbf{f}', \left(\mathbb{Q}^i, \tau_i\right)_{i \in [\![0,d-2]\!]}\right)$ such that for all $i \in [\![0, d-2]\!]$, $\mathbf{f}_* \mathbb{P}^i = \mathbf{f}'_* \mathbb{Q}^i$ and $\boldsymbol{\varphi} := \mathbf{f}'^{-1} \circ \mathbf{f} \notin \mathcal{S}_{\text{reparam}}$. Notice that for all $i \in [\![0, d-2]\!]$, $\mathbb{P}^i$ and $\mathbb{Q}^i$ are independent components and therefore Markov relative to $G$. Since the targets are unknown, we can construct a spurious solution as follows: for all $i \in [\![d-1, K]\!]$, set $\tau_i = \{1\}$, choose $\mathbb{P}^i$ to be any distribution that is composed of independent components, and set $\mathbb{Q}^i := \boldsymbol{\varphi}_* \mathbb{P}^i$. Then for all $i \in [\![0, K]\!]$, $\mathbf{f}_* \mathbb{P}^i = \mathbf{f}'_* \mathbb{Q}^i$ and $\boldsymbol{\varphi} := \mathbf{f}'^{-1} \circ \mathbf{f} \notin \mathcal{S}_{\text{reparam}}$. $\qquad\square$

*Remark* E.9. Although the proof above is almost trivial, it clarifies that a minimal requirement for identifiability up to permutation and scaling, both in ICA and in CRL, is that each latent variable must be intervened on at least once.

**Relaxing the assumption of a known causal graph.** Since we only require the distributions $\mathbb{P}^k$ in Defn. 3.2 to be Markov to $G$ (resp., $\mathbb{Q}^k$ to $G'$), our theory suggests that it should be possible to identify latent variables up to the ambiguities characterised in our results even when the learned model assumes a *denser* DAG than the true one: in particular, a fully connected graph would always be a valid choice, thus partially relaxing the assumption of a known graph. A thorough characterisation of the performance of CauCA in this misspecified setting is left for future work.

# F    Relationship between CauCA and nonlinear ICA

**ICA is a special case of CauCA.** One way to show that ICA is a special case of CauCA is the following. In CauCA, we suppose the distributions are *Markov* to a given graph $G$, but not necessarily *faithful* to $G$. This implies that any CauCA model $(G, \mathcal{F}, \mathcal{P}_G)$ may allow independent components: this holds even for models with a non-empty graph (in this case, the distribution would be unfaithful to $G$). Since the distributions with independent components are a small subset within the set of distributions which are Markov to a non-trivial graph $G$, it follows that CauCA is a strictly more general, and harder, problem than ICA, and no trivial reduction of CauCA to ICA is possible.

**Comparison to the results of Hyvärinen et al. [21].** For the special case of CauCA where there are no edges in the causal graph (i.e., ICA), Hyvärinen et al. [21, Thm.1] proved an identifiability result based on (in our terminology) $2d$ interventional datasets with unknown intervention targets, plus one unintervened dataset. While our work takes inspiration from [21], our theoretical analysis presents several differences.

Our identifiability results can be compared with [21, Thm.1] in two cases:

**(i) Trivial graph:**

- **Previous work:** A core step in the proof of Thm. 1 of [21] is eq. (20,21), which is essentially the same as (33), (34) in our work. We will refer to the proof of our Prop. 4.9 in the following (since the proof of this proposition specifically is similar to the one of Thm.1 of [21]). The proof proceeds then as follows: we take twice the partial derivatives of (32)—i.e., we calculate the Hessian matrix of the two sides. We then obtain a system of $2d$ equations, and identifiability corresponds to the uniqueness of the solution of the system $\mathbf{0} = \mathbf{A}\mathbf{x}$, where $\mathbf{A}$ is in $\mathbb{R}^{2d \times 2d}$. So $\mathbf{A}$ needs to be invertible—i.e., the "variability" assumption in [21, Thm.1].
- **Our work:** Our Prop. 4.6 and Corollary 4.8 show that, if the intervention targets are known, we can provide an identifiability proof which only requires the Jacobian of the above equation to be invertible (instead of Hessian, as in the previous results). The functions in $\mathcal{F}$ can thus be $\mathcal{C}^1$, instead of $\mathcal{C}^2$, and the linear system will be of $d$ equations instead of $2d$. In short, exploiting knowledge on the intervention targets allows us to provide a different proof, even for the special case of ICA.

In Tab. 2, we summarize our theoretical results for nonlinear ICA and compare them to [21, Thm.1].

**(ii) Nontrivial graph:**

As we mentioned in the previous paragraph, CauCA is a strictly more general problem than ICA. Our proof of Thm. 4.2 is therefore even farther from all ICA results. A peculiarity of ICA is that, after

Table 2: **Overview of ICA identifiability results.** For the trivial DAG over $Z_1$, $Z_2$, $Z_3$ (i.e., no edges), we summarise our identifiability results (§ 4.2) for representations learnt by maximizing the likelihoods $\mathbb{P}_\theta^k(X)$ based on multiple interventional regimes. $\pi[\cdot]$ denotes a permutation.

| Requirement of interventions | Learned representation $\hat{\mathbf{z}} = \hat{\mathbf{f}}^{-1}(\mathbf{x})$ | Reference |
|---|---|---|
| 1 intervention on any two nodes respectively with Asm. 4.1 | $[h_1(z_1), h_2(z_2), h_3(z_3)]$ | Prop. 4.6 |
| 1 intervention on $z_1$ and 2 fat-hand interventions on $(z_2, z_3)$ with Asm. 4.4 | $[h_1(z_1), h_2(z_2, z_3), h_3(z_2, z_3)]$ | Corollary 4.8 |
| 1 intervention on $z_1$ and 4 fat-hand interventions on $(z_2, z_3)$ with "variability" assumption | $\left[h_1(z_1), \pi[h_2(z_2), h_3(z_3)]\right]$ | Prop. 4.9 |
| 1 intervention per node on any two nodes respectively with unknown order ("matched intervention target", see Defn. E.2) with Asm. E.4 | $\pi[h_1(z_1), h_2(z_2), h_3(z_3)]$ | Prop. E.6 |
| 6 fat-hand interventions on $(z_1, z_2, z_3)$ with "variability" assumption | $\pi[h_1(z_1), h_2(z_2), h_3(z_3)]$ | [21][Thm. 1] |

taking the Hessian, the left-hand side of (34) vanishes: this is exploited in the proof of [21, Thm.1]. All the proof techniques of nonlinear ICA papers we are aware of rely on this vanishing left-hand side over all coordinates of $z$. Unfortunately, with a non-trivial graph, it is impossible to get the same after taking the partial derivatives of every coordinate of $z$: in fact, for each $i \in [d]$, the left-hand side of the $i$-th equation depends on the ancestors of $z_i$. Our proofs for the case with a nontrivial graph (Thm. 4.2 and Thm. 4.5) therefore follow different strategies, based on first derivatives alone.

# G  Additional theoretical results used in the design of experiments

## G.1  Multi-objective and pooled objective

Our identifiability theory implies that the ground-truth latent CBN could be learned by maximizing likelihood across all interventional regimes, i.e.,

$$\theta^* = \bigcap_{k=0}^{d} \arg\max_\theta \frac{1}{N_k} \sum_{n=1}^{N_k} \log p_{\boldsymbol{\theta}}^k(\mathbf{x}^{(n,k)}),$$

where $\log p_{\boldsymbol{\theta}}^k(\mathbf{x}^{(n,k)})$ is defined in (10). This is a multi-objective optimization problem.

In general, multi-objective optimization is hard; however, we show that, because of our assumptions, we can equivalently optimize a pooled objective. In fact, suppose

$$f_k(\boldsymbol{\theta}) = \log p_{\boldsymbol{\theta}}^k(\mathbf{x}), \quad f(\boldsymbol{\theta}) = \sum_{k=0}^{d} f_k(\boldsymbol{\theta}).$$

Define $\boldsymbol{\Theta}_k := \arg\max_{\boldsymbol{\theta}} f_k(\boldsymbol{\theta})$. By our problem setting we know $\bigcap_{k=0}^{d} \boldsymbol{\Theta}_k \neq \varnothing$.

On the one hand, for all $\hat{\boldsymbol{\theta}} \in \bigcap_{k=0}^{d} \boldsymbol{\Theta}_i$, $\hat{\boldsymbol{\theta}} \in \arg\max_{\boldsymbol{\theta}} f(\boldsymbol{\theta})$. On the other hand, we will prove by contradiction that $\forall \hat{\boldsymbol{\theta}} \in \arg\max_{\boldsymbol{\theta}} f(\boldsymbol{\theta})$, $\hat{\boldsymbol{\theta}} \in \bigcap_{k=0}^{d} \boldsymbol{\Theta}_k$. Suppose there exists $i \in [\![0, d]\!]$ such that $\hat{\boldsymbol{\theta}} \notin \boldsymbol{\Theta}_i$. Then for all $\boldsymbol{\theta}^* \in \bigcap_{k=0}^{d} \boldsymbol{\Theta}_k$, $f_i(\hat{\boldsymbol{\theta}}) < f_i(\boldsymbol{\theta}^*)$. Thus $f(\hat{\boldsymbol{\theta}}) = \sum_{k=0}^{d} f_k(\hat{\boldsymbol{\theta}}) < \sum_{k=0}^{d} f_k(\boldsymbol{\theta}^*) = f(\boldsymbol{\theta}^*)$. This yields a contradiction with $\hat{\boldsymbol{\theta}} \in \arg\max_{\boldsymbol{\theta}} f(\boldsymbol{\theta})$.

We thus conclude that $\bigcap_{k=0}^{d} \arg\max_{\boldsymbol{\theta}} f_k(\boldsymbol{\theta}) = \arg\max_{\boldsymbol{\theta}} f(\boldsymbol{\theta})$.

## G.2  Expressivity in multi-intervention learning

To learn the latent CBN, we need to learn both the latent distributions (both the unintervened causal mechanisms and the intervened ones) and mixing functions. For the latent distributions, a natural

question is whether any of them could be fixed without loss of generality. This is for example possible in the context of nonlinear ICA, where due to the indeterminacy up to element-wise nonlinear scaling of the latent variables, we can arbitrarily fix their univariate distributions w.l.o.g. The following proposition elucidates this matter for CauCA.

**Proposition G.1.** *For a ground truth latent CBN,* $\left(G, \mathbf{f}, \left(\mathbb{P}^k, \tau_k\right)_{k \in [\![0,d]\!]}\right)$ *where* $\left(\mathbb{P}^k, \tau_k\right)_{k \in [\![1,d]\!]}$ *are obtained by perfect stochastic intervention on every variable respectively, fix at most one element for each $k$ in* $\{\mathbb{Q}_k | pa(k) = \varnothing\} \cup \{\widetilde{\mathbb{Q}}_k | k \in [d]\}$*, there exists* $\left(G, \mathbf{f}', \left(\mathbb{Q}^k, \tau_k'\right)_{k \in [\![0,d]\!]}\right)$ *s.t.* $\mathbf{f}_* \mathbb{P}^k = \mathbf{f}'_* \mathbb{Q}^k, \mathbf{f}'^{-1} \circ \mathbf{f} \in \mathcal{S}_{scaling}$.

In practice, this means that in order to learn $\mathbf{f}'$ and $(\mathbb{Q}^k)_{k \in [d]}$ with $d$ perfect stochastic interventions, even if we fix all the intervention mechanisms $(\widetilde{\mathbb{Q}}_k)_{k \in [d]}$, we can still learn a true latent model up to scaling functions. Equivalently, we could also fix instead $\{\mathbb{Q}_k | pa(k) = \varnothing\} \cup \{\widetilde{\mathbb{Q}}_k | pa(k) \neq \varnothing\}$.

*Proof.* For any $k \in [d]$, $\mathbb{P}^k = \widetilde{\mathbb{P}}_k \prod_{j \neq k} \mathbb{P}_j$. Without loss of generality by exchanging $\mathbb{Q}_k$ with $\widetilde{\mathbb{Q}}_k$, suppose we are given $\widetilde{\mathbb{Q}}_k$, by Lemma B.2, there are two possible diffeomorphisms for $T_k$ st. $(T_k)_* \widetilde{\mathbb{P}}_k = \widetilde{\mathbb{Q}}_k$. Choose one of them arbitrarily. Set $\mathbf{T} := (T_k)_{k \in [d]}$.

Define $\mathbb{Q}^0 := \mathbf{T}_* \mathbb{P}^0$. By Lemma B.3, $\forall i \in [d], \forall z_i \in \mathbb{R}, \forall \mathbf{z}_{pa(i)} \in \mathbb{R}^{\#pa(i)}$,

$$p_i \left(z_i \mid \mathbf{z}_{pa(i)}\right) = q_i \left(T_i \left(z_i\right) \mid \mathbf{T}_{pa(i)} \left(\mathbf{z}_{pa(i)}\right)\right) |T_i' \left(z_i\right)| .$$

Multiply the causal mechanisms of all $i \in [d] \setminus \{k\}$ and the intervened mechanism of $k$, we get

$$\widetilde{p}_k(z_k) \prod_{i \neq k} p_i \left(z_i \mid \mathbf{z}_{pa(i)}\right) = \widetilde{q}_k \left(T_k \left(z_k\right)\right) |T_k' (z_k)| \prod_{i \neq k} q_i \left(T_i \left(z_i\right) \mid \mathbf{T}_{pa(i)} \left(\mathbf{z}_{pa(i)}\right)\right) |T_i' \left(z_i\right)| ,$$

which is equivalent to $\widetilde{p}^k(\mathbf{z}) = \widetilde{q}^k(\mathbf{T}(\mathbf{z})) |\det \mathbf{T}(\mathbf{z})|$. This is the change of variable formula of the diffeomorphism $\mathbf{T}$ in $\mathbb{R}^d$, and implies $\mathbb{Q}^k = \mathbf{T}_* \mathbb{P}^k$.

Define $\mathbf{f}' := \mathbf{T} \circ \mathbf{f}^{-1}$, then $\mathbf{f}'^{-1} \circ \mathbf{f} = \mathbf{T} \in \mathcal{S}_{scaling}$. $\qquad \square$

**Remark** If $G$ is non trivial, given $\left(G, \mathbf{f}, \mathbb{P}^0\right)$, in general there does not exist $\mathbb{Q}^0 \in P_G, \mathbf{f}' \in \mathcal{F}$ s.t. $\mathbf{f}_* \mathbb{P}^0 = \mathbf{f}'_* \mathbb{Q}^0, \mathbf{f}'^{-1} \circ \mathbf{f} \in \mathcal{S}_{scaling}$.

To see why, consider the graph $z_1 \rightarrow z_2$. Fix $\mathbb{Q}_1$, there are only two possible diffeomorphisms for $T_1$ s.t. $\mathbb{Q}_1 = (T_1)_* \mathbb{P}_1$ by Lemma B.2.

If $\mathbb{Q}_2 \left(Z_2 \mid T_1(z_1)\right)$ is fixed, to find $T_2$ s.t.

$$(T_2)_* \mathbb{P}_2 \left(Z_2 \mid z_1\right) = \mathbb{Q}_2 \left(Z_2 \mid T_1(z_1)\right) \quad \forall z_1 \in \mathbb{R},$$

by Lemma B.2, the only possible $T_2$ are $G \left(\cdot \mid T_1 \left(z_1\right)\right)^{-1} \circ F \left(\cdot \mid z_1\right)$ and $\bar{G} \left(\cdot \mid T_1 \left(z_1\right)\right)^{-1} \circ F \left(\cdot \mid z_1\right)$. In general, these two functions depend on $z_1$. For example if $\mathbb{P}_1 \left(Z_1\right) \perp\!\!\!\perp \mathbb{P}_2 \left(Z_2\right), \mathbb{Q}_2 \left(Z_2 \mid z_1\right) \sim \mathcal{N} \left(z_1, 1\right)$ then $T_2$ depend on $z_1$. Namely, There is no $\mathbf{T} \in \mathcal{S}_{scaling}$ s.t. $\mathbb{Q}^0 = \mathbf{T}_* \mathbb{P}^0$.

### G.3 Normalizing flows for nonparametric CBN

In App. I, we learn a nonlinear mixing function and nonparametric causal mechanisms $(p_i^\phi(z_i \mid \mathbf{z}_{pa(i)}))_{i \in [d]}$. We model $p^\phi$ by a normalizing flow $\mathbf{h}^\phi : \mathbb{R}^d \rightarrow \mathbb{R}^d$ from the space of $\mathbf{Z}$ to the space of $\mathbf{U}$, while fixing $\mathbb{P}_\mathbf{u}$. The goal is to learn $\mathbf{h}^{\phi-1}$, the reduced form of an SCM which pushes forward the exogenous variables $\mathbf{U}$ to every causal mechanism $(p_i^\phi(z_i \mid \mathbf{z}_{pa(i)}))_{i \in [d]}$. ($\mathbf{h}^\phi$ is denoted as $\mathbf{h}_\phi^{\text{CBN}}$ in App. I). In the following, we prove that such $\mathbf{h}^\phi$ exists in a special case.

**Proposition G.2.** *For any CBN* $\{\mathbb{P}_i^\phi(Z_i | \mathbf{Z}_{pa(i)})\}_{i \in [d]}$ *Markov and faithful to a fully connected DAG $G$, with $\mathbb{P}^\phi$ absolutely continuous and smooth in $\mathbb{R}^d$, and for any independent component joint distribution $\mathbb{P}_\mathbf{u}$ absolutely continuous and smooth in $\mathbb{R}^d$, there exists a normalizing flow* $\mathbf{h}^\phi : \mathbb{R}^d \rightarrow \mathbb{R}^d$ *such that*

- *there exist a permutation matrix $\mathbf{P}$ and an autoregressive normalizing flow $\tilde{\mathbf{h}}$ such that $\mathbf{h}^\phi = \mathbf{P} \circ \tilde{\mathbf{h}}$ (autoregressive in the sense that $D\tilde{\mathbf{h}}(\mathbf{z})$ is lower triangular for all $\mathbf{z} \in \mathbb{R}^d$);*

- $\mathbf{h}_*^\phi \mathbb{P}^\phi = \mathbb{P}_{\mathbf{u}}$;

- *for all $i \in [d]$, $p_i^\phi(z_i|\mathbf{z}_{pa(i)}) = \left|\frac{\partial h_i^\phi}{\partial z_i}(\mathbf{z})\right| p_{\mathbf{u}_i}\left(h_i^\phi(\mathbf{z})\right)$.*

*Proof.* Without loss of generality by applying a permutation matrix $\mathbf{P}$, we can suppose that the index of $G$ preserves the partial order induced by $E(G)$: $i < j$ if $(i, j) \in E(G)$. In the following, we can thus replace $\tilde{\mathbf{h}}$ by $\mathbf{h}^\phi$. Since $\mathbb{P}^\phi$ is faithful to a fully connected graph, $\mathbf{Z}_{pa(i)} = \mathbf{Z}_{<i}$.

By [40] [Sec 2.2], there exists a Darmois transformation $\mathbf{F} : \mathbb{R}^d \to (0, 1)^d$ s.t. $\mathbf{F}_* \mathbb{P}^\phi = \mathrm{Unif}(0, 1)^d$, and a Darmois transformation $\mathbf{G} : \mathbb{R}^d \to (0, 1)^d$ s.t. $\mathbf{G}_* \mathbb{P}_{\mathbf{u}} = \mathrm{Unif}(0, 1)^d$. Define $\mathbf{h}^\phi := \mathbf{G}^{-1} \circ \mathbf{F}$, then $\mathbf{h}_*^\phi \mathbb{P}^\phi = \mathbb{P}_{\mathbf{u}}$.

Notice that the Darmois transformations $\mathbf{F}$ and $\mathbf{G}$ are autoregressive normalizing flows, i.e. $D\mathbf{F}(\mathbf{z})$ and $D\mathbf{G}(\mathbf{z})$ are lower triangular for all $\mathbf{z} \in \mathbb{R}^d$. Since the autoregressive normalizing flows are closed under composition and inverse, $D(\mathbf{h}^\phi)(\mathbf{z})$ is also lower triangular for all $\mathbf{z} \in \mathbb{R}^d$. By induction using Lemma B.4, it is direct to prove that for all $i \in [d]$, $\mathbf{h}_{[i]}^\phi$ is a diffeomorphism from the first $i$ variables of $Z$ to the first $i$ variables of $\mathbf{U}$.

By the definition of $\mathbf{h}^\phi$, $\mathbb{P}_{\mathbf{u}} = \mathbf{h}_*^\phi \mathbb{P}^\phi$. By the change of variable formula, $p^\phi(\mathbf{z}) = \left|\det D\mathbf{h}^\phi(\mathbf{z})\right| p_{\mathbf{u}}\left(\mathbf{h}^\phi(\mathbf{z})\right)$.

We now prove by induction that for all $i \in [d]$, $p_i^\phi(z_i|\mathbf{z}_{pa(i)}) = \left|\frac{\partial h_i^\phi}{\partial z_i}(\mathbf{z})\right| p_{\mathbf{u}_i}\left(h_i^\phi(\mathbf{z})\right)$. For $i = 1$, by the assumption of indices in $G$, $z_1$ has no parent. Since $\mathbf{h}_1^\phi$ is a diffeomorphism, $p_1^\phi(z_1) = \left|\frac{\partial h_1^\phi}{\partial z_1}(z_1)\right| p_{\mathbf{u}_1}\left(h_1^\phi(z_1)\right)$.

Suppose the property is true for $i$:

$$p_i^\phi(z_i|\mathbf{z}_{pa(i)}) = \left|\frac{\partial h_i^\phi}{\partial z_i}(\mathbf{z})\right| p_{\mathbf{u}_i}\left(h_i^\phi(\mathbf{z})\right) \tag{46}$$

Then for $i + 1$, since $(\mathbf{h}^\phi)_{[i+1]}$ is a diffeomorphism with lower triangular Jacobian, we have

$$\prod_{j=1}^{i+1} p_j^\phi(z_j|\mathbf{z}_{pa(j)}) = \left|\det D\mathbf{h}_{[i+1]}^\phi(\mathbf{z})\right| p_{\mathbf{u}_{[i+1]}}\left(\mathbf{h}_{[i+1]}^\phi(\mathbf{z})\right) = \prod_{j=1}^{i+1} \left|\frac{\partial h_j^\phi}{\partial z_j}(\mathbf{z})\right| p_{\mathbf{u}_j}\left(h_j^\phi(\mathbf{z})\right) \tag{47}$$

where the last equality is because $D\mathbf{h}^\phi$ is lower triangular and $\mathbb{P}_{\mathbf{u}}$ is independent components. By dividing eq. (47) by eq. (46), we obtain the result. □

In the method above, we assume that the CBN is Markov and faithful to a fully connected graph. Devising a more general method for sparse graphs is left for future work.

We note that the concurrent work by [23] may also be used to learn nonparametric latent CBNs. One main difference is that in our normalizing flow, every coordinate of $\mathbf{h}^{\phi-1}$ models separately a causal mechanism, which is not the case in [23]. In our objective function (10), we need to model and learn every causal mechanism individually.

# H  Details Experiments

## H.1  Synthetic Data Generation

**Directed Acyclic Graph (DAG).** In order to generate data, we begin by sampling a random DAG $G \sim \mathbb{Q}_G$, where $\mathbb{Q}_G$ is a distribution over DAGs. The edge density of the DAG is set to $0.5$ in topological order, meaning that the edges in the DAG are constrained to follow the variable index order: an edge $Z_i \to Z_j$ can only exist if $j > i$. To construct the DAG, we individually sample each

potential edge $Z_i \to Z_j$ with $j > i$ with a probability of $0.5$, while all other edges are assigned a probability of $0$. For experiments conducted in the CauCA setting with non-trivial graphs (i.e., not empty), we reject and redraw any sampled DAGs that do not contain any edges.

**Causal Bayesian network (CBN).** To sample data from a CBN, we start by drawing the parameters of a linear Gaussian Structural Causal Model (SCM) with additive noise:

$$Z_i := \sum_{j \in \text{pa}(i)} \alpha_{i,j} Z_j + \varepsilon_i, \tag{48}$$

where the linear parameters are drawn from a uniform distribution $\alpha_{i,j} \sim \text{Uniform}(-a, a)$ and the noise variable is Gaussian $\varepsilon_i \sim \mathcal{N}(0, 1)$. The *signal-to-noise ratio* for the SCM, denoted as $a/\text{std}(\varepsilon_i) = a$, describes the strength of the dependence of the causal variables relative to the exogenous noise. For most experiments, the signal-to-noise ratio is set to 1. In Fig. 4 (g), we explore values ranging from 1 to 10. To specify the latent CBNs, we can define the conditional distributions entailed by the SCMs defined as in eq. (48), see also App. H.2 and eq. (52).

**Generating interventional datasets.** For a given CBN, we generate $d + 1$ related datasets: one unintervened (observational) dataset and $d$ interventional datasets, where the CBN was modified by a perfect stochastic intervention on one variable (i.e., one dataset for each variable in the CBN). W.l.o.g. we assume that the $k^{\text{th}}$ variable was intervened on in dataset $k$; $k = 0$ denotes the observational dataset. The intervention is applied in the SCM by removing the influence of the parent variables and changing the exogenous noise by shifting its mean up or down. Hence, for dataset $k$ we have

$$Z_k := \tilde{\varepsilon}_k, \quad \text{with} \quad \tilde{\varepsilon}_k \sim \mathcal{N}(\mu, 1), \tag{49}$$

where the mean of the noise $\mu$ is uniformly sampled from $\{\pm 2\}$ and fixed within each dataset. Each dataset comprises a total of $200,000$ data points, resulting in $(d + 1) \times 200,000$ data points in total for each CBN.

**Mixing function.** The mixing function takes the form of a multilayer perceptron $\mathbf{f} = \sigma \circ \mathbf{A}_M \circ \ldots \circ \sigma \circ \mathbf{A}_1$, where $\mathbf{A}_m \in \mathbb{R}^{d \times d}$ for $m \in [\![1, M]\!]$ denote invertible linear maps, and $\sigma$ is an element-wise invertible nonlinear function. The elements of the linear maps are sampled independently $(\mathbf{A}_m)_{i,j} \sim \text{Uniform}(0, 1)$ for $i, j \in [\![1, d]\!]$. A sampled matrix $\mathbf{A}_m$ is rejected and re-drawn if $|\det \mathbf{A}_m| < 0.1$ to rule out linear maps that are (close to) singular. The invertible element-wise nonlinearity is a leaky-tanh activation function:

$$\sigma(x) = \tanh(x) + 0.1x, \tag{50}$$

as used in [12].

## H.2 Model architecture

**Normalizing flows.** We use normalizing flows [40] to learn an encoder $\mathbf{g}_{\boldsymbol{\theta}} : \mathbb{R}^d \to \mathbb{R}^d$. Normalizing flows model observations $\mathbf{x}$ as the result of an invertible, differentiable transformation $\mathbf{g}_{\boldsymbol{\theta}}$ on some base variables $\mathbf{z}$,

$$\mathbf{x} = \mathbf{g}_{\boldsymbol{\theta}}(\mathbf{z}). \tag{51}$$

We apply a series of $L = 12$ such transformations $\mathbf{g}_{\boldsymbol{\theta}^l}^l : \mathbb{R}^d \to \mathbb{R}^d$, which we refer to as *flow layers*, such that the resulting transformation is given by $\mathbf{g}_{\boldsymbol{\theta}} = \mathbf{g}_{\boldsymbol{\theta}^L}^L \circ \ldots \circ \mathbf{g}_{\boldsymbol{\theta}^1}^1$. We use Neural Spline Flows [10] for the invertible transformation, with a 3-layer feedforward neural network with hidden dimension 128 and a permutation in each flow layer.

**Base distribution.** We extend the typically used simple base distributions to encode information about the CBN. We have one distribution per dataset $(\hat{p}_{\boldsymbol{\theta}_{\text{CBN}}^k}^k)_{k \in [\![0,d]\!]}$ over the learned base noise variables $\mathbf{z}$. The conditional density of latent variable $i$ in dataset $k$ is given by

$$\hat{p}_{\boldsymbol{\theta}_{\text{CBN}}^k}^k(z_i \mid \mathbf{z}_{\text{pa}(i)}) = \mathcal{N}\left( \sum_{j \in \text{pa}(i)} \hat{\alpha}_{i,j} z_j, \hat{\sigma}_i \right), \tag{52}$$

when $i \neq k$, i.e., when variable $i$ is not intervened on. For $i = k$, we have

$$\hat{p}_{\boldsymbol{\theta}_{\text{CBN}}^k}^k(z_i) = \mathcal{N}(\hat{\mu}_i^k, \hat{\sigma}_i^k). \tag{53}$$

In summary, the base distribution parameters are the parameters of the linear relationships between parents and children in the CBN $(\hat{\alpha}_{i,j})_{i,j\in[\![1,d]\!]}$, the standard deviations for each variable in the observational setting $(\hat{\sigma}_i)_{i\in[\![1,d]\!]}$, and mean and standard deviation for the intervened variable in each dataset $(\hat{\mu}_i^k, \hat{\sigma}_i^k)_{i,k\in[\![1,d]\!]}$.

**Linear baseline model.** For the linear baseline models shown in Fig. 4 (a, b, e, f) we replace the nonlinear transformations $(\mathbf{g}_{\boldsymbol{\theta}^l}^l)_{l\in[\![1,L]\!]}$ by a single linear transformation. The base distribution stays the same.

**Graph-misspecified model.** In order to test the impact of providing knowledge about the causal structure, we compare the CauCA model to one that assumes the latents are independent. This is achieved by setting $\hat{\alpha}_{i,j} = 0 \; \forall i,j \in [\![1,d]\!]$ in the base distribution (52).

### H.3 Training and model selection

**Training parameters.** We use the ADAM optimizer [27] with cosine annealing learning rate scheduling, starting with a learning rate of $5 \times 10^{-3}$ and ending with $1 \times 10^{-7}$. We train the model for 50–200 epochs with a batch size of 4096. The number of epochs was tuned manually for each type of experiment to ensure reliable convergence of the validation log probability.

**Pooled objective.** The learning objective described in § 5 is using the pooled rather than the multi-objective formulation. In App. G, we prove that for our problem the two are equivalent.

**Fixing CBN parameters.** As explained in Prop. G.1, we can fix some of the CBN parameters w.l.o.g. In our case, we fix the noise parameters for intervened mechanisms of non-root variables and observational mechanisms of root variables.

**Model selection.** For each drawn latent CBN, we train three models with different initializations and select the model with the highest validation log probability at the end of training.

**Compute.** Each training run takes 2–8 hours on NVIDIA RTX-6000 gpus. For the experiments shown in the main paper, we performed 450 training runs (30 per violin plot / point in Fig. 4 (g)) which sums up to around 2250 compute hours (assuming an average run time of 5 hours).

## I Nonparametric Experiments

The main experiments in § 5 were made under the restriction that the ground-truth CBN and the learned latent CBN were assumed to be linear Gaussian. In the experiments shown here, we relax those restrictions. In the following, we describe the differences with respect to the main experiments. All other settings and parameters are the same as described in § 5 and App. H.

### I.1 Synthetic Data Generation

**Causal Bayesian network (CBN).** To sample data from a CBN, we draw the parameters of a nonlinear non-additive-noise Structural Causal Model (SCM):
$$Z_i := h^{\mathrm{loc}}(Z_{\mathrm{pa}}(i)) + h^{\mathrm{scale}}(Z_{\mathrm{pa}}(i))\,\varepsilon_i, \tag{54}$$
where the location and scale functions $h^{\mathrm{loc}}, h^{\mathrm{scale}} : \mathbb{R}^{\#\mathrm{pa}(i)} \to \mathbb{R}$ are parameterized by random 3-layer neural networks (similar to the random mixing function) and the noise variable is Gaussian $\varepsilon_i \sim \mathcal{N}(0,1)$.

### I.2 Model architecture

**Base distribution.** We extend the typically used simple base distributions to encode information about the CBN. We train a second normalizing flow to learn a mapping $\mathbf{h}_{\boldsymbol{\phi}}^{\mathrm{CBN}} : \mathbb{R}^d \to \mathbb{R}^d$ from the latent causal variables $\mathbf{z}$ to exogenous noise variables $\mathbf{u}$, which we explain in App. G.3. We encode knowledge about the causal graph in the base distribution by passing the latent variables in causal order to the invertible transformation $\mathbf{h}^{\mathrm{CBN}}$, whose Jacobian is lower triangular. This ensures the correct causal relationships among the elements of $\mathbf{z}$. During training, we fix the distribution of $\mathbf{u}$ and the intervened upon latent variables, i.e. $z_k$ for dataset $k$, to be standard normal. We use a similar architecture based on Neural Spline Flows, with one difference: we omit the permutation layer, which would violate the topological order of the variables.

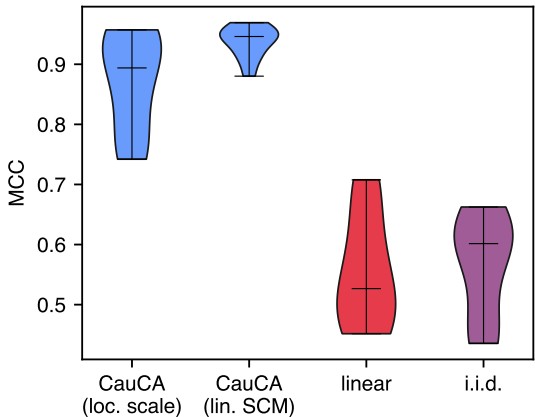

Figure 6: **Experimental results with nonparametric model.** The figure shows the mean correlation coefficients (MCC) between true and learned latents for Causal Component Analysis (CauCA) experiments. The first two violin plots show the fully nonparametric model when the ground truth latent CBN is generated by a location-scale model and for the case with a linear SCM generating the ground truth CBN. Misspecified models assuming a linear encoder function class and a naive (single-environment) unidentifiable normalizing flow with an independent Gaussian base distribution (labelled *i.i.d.*) are compared. The misspecified models are trained on a location-scale CBN. All violin plots show the distribution of outcomes for 10 pairs of CBNs and mixing functions.

## I.3 Results

In line with the experiments presented in § 5, we compare the nonparametric model (*blue*) to misspecified baselines: one with a correctly specified nonparametric latent model but employing a linear encoder (*red*), and a model with a nonlinear encoder but assuming a causal graph with no arrows (*orange*). The results shown in Fig. 6 show that the nonparametric model accurately identifies the latent variables, benefiting from both the nonlinear encoder and the explicit modelling of causal dependence. When we compare the nonparametric model trained on the location-scale data generating process (*left blue violin*) to one trained on the linear Gaussian process (*right blue violin*), we observe that in the location-scale case it seems to be more challenging for the model to uncover the true latent factors.

