# OpenReview forum: "Causal Component Analysis"
_NeurIPS.cc/2023/Conference — NeurIPS 2023 poster_

### Official Review · Reviewer_8r7j · 2023-06-28

**Soundness:** 3 good
**Presentation:** 4 excellent
**Contribution:** 3 good
**Rating:** 5
**Confidence:** 3

**Summary:**

The authors propose CauCA as a middle ground between ICA and CRL. They introduce two new assumptions, which is (i) knowledge of the causal variables and how they are related, and (ii) knowledge of the causal variables that were intervened upon to generate each dataset. With these two assumptions, CauCA identifies the latent variables up to element-wise nonlinearity. This is done with a single technical assumption regarding the pre- and post- intervention causal mechanisms, and without the more common assumptions e.g. restrictions on the prior or mixing function. The authors validate their method with a toy problem with synthetic data.

**Strengths:**

The paper is well written, and I appreciate the effort the authors made to intuitively explain their definitions, assumptions, and results, and to contextualize their work in relation to the broader landscape of identifiable representation learning. This made the paper significantly easier to understand.

I like the authors' approach of tackling a middle ground between ICA and CRL, where we use domain knowledge of what causal variables are involved, and how they are related. The main strength of this paper is that it provides a strong identifiability result (element-wise nonlinearity, no permutation ambiguity) with few technical assumptions relative to the existing literature. There is also a nice connection to the existing identifiability results in nonlinear ICA / VAE with auxiliary variables (Khemakhem et al., 2020).

**Weaknesses:**

This paper is weakened by a highly unrealistic assumption, which is knowledge of the intervention targets $\tau_k$ for each dataset $\mathcal{D}_k$. I can't see how this assumption could be satisfied on a non-synthetic dataset.

Other existing work also make strong assumptions that are arguably unrealistic, but those models can at least be estimated from real-world datasets, and whether they are identifiable or not is a separate question. The fact that the estimation of this model hinges on an unrealistic assumption is a significant limitation in terms of its practicality.

**Questions:**

Can you think of a way to use an existing dataset to demonstrate this method on, without having access to the underlying data generating process? The data does not need to be complex.

Using something like dSprites as an example, the dataset would need to be partitioned such that e.g. in $\mathcal{D}_1$, only the shape changes, and in $\mathcal{D}_2$, only the color changes, etc. You need access to the ground truth latents to split the dataset this way, which is what I mean by "requiring access to the underlying data generating process." I will increase my score if I can be convinced that this approach has some practicality.

**Limitations:**

As the rest of my review makes clear, I believe that assuming access to $\tau_k$ is a significant limitation of this work, but the authors did not discuss this point in the paper.

---

> ### Author Rebuttal · Authors · 2023-08-09
>
> Thank you for the detailed review and for valuing _“the effort the authors made to intuitively explain their definitions, assumptions, and results, and to contextualize their work in relation to the broader landscape of identifiable representation learning”_.
>
> Let us emphasize that learning causal relations from real-world observations is an important but still very challenging problem. The goal of this paper is not to solve this problem directly: instead, we try to better understand the problem of learning causal representations through the lens of identifiability. We propose a new framework that, as you point out, “tackles a middle ground between ICA and CRL” and show strong identifiability results in this settings. We believe that this is an important step towards a better understanding of the theoretical foundations of causal representation learning and that this will, eventually, help to find learning signals that can be exploited for practical algorithms. Therefore, we think that the technical contribution of our paper helps to advance the field even though its direct practical implications may be limited.
>
> > “This paper is weakened by a highly unrealistic assumption, which is knowledge of the intervention targets $\tau_k$ for each dataset  ${\mathcal{D}}_k$.”
>
> > “Other existing work also make strong assumptions that are arguably unrealistic, but those models can at least be estimated from real-world datasets, and whether they are identifiable or not is a separate question.”
>
> We agree that this is a strong assumption, and we therefore investigated its necessity and potential relaxations (see App. E). We will extend this based on the discussion here and add parts to a dedicated limitations section. In summary, our main findings are:
>
> **1. New result: Non-identifiability of CauCA with totally unknown targets (c.f. Appendix E, Def. E.2 (iv))**
>
> **Corollary.** Given a nontrivial DAG $G$, if the intervention targets are unknown, CauCA is not identifiable under any number of interventions.
>
> We include a proof sketch below.
>
> **Sketch of proof:** This is directly deduced from Prop. 4.3. Since the only information we have about interventional regimes are the cardinals, it suffices that we add more regimes where the first $d-1$ variables are intervened on, while remaining a spurious / nonidentifiable solution. Should this be unclear, we are happy to elaborate in the reviewer-author discussion period.
>
> **Remark:** A similar result can also be found in [1] (where the mixing is linear, but the graph is unknown). Our result is stronger than the one in [1] since it applies even for known causal graph.
>
> **2. Different kinds of knowledge about the intervention targets**
>
> In the main paper, for simplicity, we only provided the minimal set of notation required for describing the problem of CauCA with known intervention targets. However, we believe that a more general version of CauCA should also be considered for cases where the targets are partially unknown. In fact, in Appendix E, we have generalized our identifiability result to semi-unknown targets: see also. For non-trivial graphs, we generalized our results to intervention targets known only up to graph automorphisms (Thm. E.5). See also Figure 2 in the pdf attached to the response to all reviewers.
>
> Note that if there are few arrows in the ground truth graph, this relaxation is quite powerful: in the limit of an empty graph, i.e., for ICA, it corresponds to completely generalizing to the setting of matched intervention targets (see Prop. E.6). Nonetheless, there are arguably other intermediate settings to be considered, for example the one of matched intervention targets with a non-trivial DAG, which would be an even stronger relaxation and which we leave for future work.
>
> Another way to make CauCA more practical is by fat-hand interventions, where we only have partial knowledge about the targets. As we remarked, even the variability assumption of Thm1 of [2] (which is essentially our Prop. 4.7) implies that every variable should be intervened on at least once, so that the variability matrix on line 246 can be full rank. We have a new theorem generalizing Thm 4.6 to any DAG, proving that by shrinking a group of nodes into one, and assuming this does not introduce cycles in the resulting graph, then under technical assumptions on the fat-hand interventions, CauCA is block-identifiable. Due to the character limit, we are unable to provide technical details here, but we can do so during the author-reviewer discussion period should you be interested.
>
> > “Can you think of a way to use an existing dataset to demonstrate this method on, without having access to the underlying data generating process?”
>
> We believe that there are domains where it is possible to perform interventions, and where at the same time we may have some (partial) knowledge on the intervention targets. For example, in genomics, researchers can collect datasets about multiple phenotypes, and at the same time, perturb single or multiple genes through CRISPR experiments [3]. In these settings, the perturbations could be thought of as a kind of interventions; moreover, domain knowledge may (approximately) specify some of the latent causal links [4], including those that are perturbed. While we do not want to claim that our method may be applied to such settings in its current form, this is a domain where it may not be unreasonable to assume some knowledge about the intervention targets.
>
> [1] Seigal, A et al. (2022) Linear causal disentanglement via interventions.
>
> [2] Hyvarinen, A. et al. (2019). Nonlinear ICA using auxiliary variables and generalized contrastive learning.
>
> [3] Dixit, A. et al. (2016). Perturb-Seq: dissecting molecular circuits with scalable single-cell RNA profiling of pooled genetic screens.
>
> [4] Norman, M. et al. (2019). Exploring genetic interaction manifolds constructed from rich single-cell phenotypes.

---

> > ### Comment · Reviewer_8r7j · 2023-08-17
> >
> > Thank you for your detailed response, and for developing the additional theory on different degrees of knowledge about the intervention targets. I was mainly interested in the final paragraph of your response, and I actually did think about single cell genomics as a potential application, since the intervention targets are known. However, in this setting it's unrealistic to assume knowledge of the variables that govern the biological processes of interest. I still can't think of any non-toy settings where both assumptions are met. Since my personal bias is that we should be advancing methods that have potential for practical applicability, I am maintaining my score.

---

> > > ### Author Response · Authors · 2023-08-18
> > >
> > > Thank you for your response.
> > >
> > > This work is not necessarily focused on developing new practical methods, but rather on understanding what is in principle possible (identifiability) and impossible in CauCA or CRL.
> > >
> > > The setting in our paper (knowing intervention targets up to graph automorphisms) is strictly included in the setting where less knowledge about targets are given (i.e. unknown targets with every node intervened on at least once), thus our results are a special but unavoidable case in the proof for the latter setting in the future.
> > >
> > > While we agree, that the method and its underlying assumptions are not very practical in its current form, we still believe (and reviewers xM6s and nsXw seem to agree) that our theoretical results---despite the personal bias of the reviewer---still constitute a valuable contribution, whose insights may help construct more practical methods in future work.

---

### Official Review · Reviewer_xr1H · 2023-07-06

**Soundness:** 3 good
**Presentation:** 3 good
**Contribution:** 2 fair
**Rating:** 4
**Confidence:** 4

**Summary:**

The paper provides identifiability results for causal component analysis, where the causal structure of the latent variables is assumed to be known. The goal is to identify the causal mechanisms and mixing functions up to a certain degree of uncertainty. Compared to the previous work with unknown graphs and linearity, the proposed result is a trade-off between constraints on the structure (known causal graph) and function.

**Strengths:**

1. The paper considers an interesting problem that is important for the development of causal analysis in the context of deep learning.

2. The writing is very clear.

3. The visualization of the violation of Assumption 4.1 is particularly helpful.

**Weaknesses:**

My comments are as follows. They are not necessarily weaknesses since some of them could be further clarified during the rebuttal.

1. If the causal graph is given, it seems that the considered setting could possibly be transferred to the traditional one where there is no dependence among latent variables but only an unknown mixing process between latent and observed variables. Since interventional data is a specific type of multi-environment data (with an additional indicator for interventions), the novelty compared to the previous results (e.g., [1] and many others) does not appear to be fully clear. This has been partially discussed in Sec. 4.2 but it seems that the principal difference and connection, in both setting and proof techniques, should also be discussed deeper in other parts of the paper. The required number of environments is strictly lower but at the cost of interventional data.

2. To justify the assumption on the known causal graph, the paper compared the spaces of potential DAGs and spurious solutions for the representation, and suggested that assuming the known structure is reasonable since 'there are only finitely many possible DAGs'. IMHO, it seems that the statement is not strong enough to support the assumption, since the exponential nature of the DAG space, although finite, is still unacceptable in most scenarios.

3. With imperfect intervention, the ambiguity of the identification is a mixture of nodes that are ancestors of the target node. While this is a non-trivial reduction from the unconstrained space, it remains uncertain about the significance of this result. In the considered setting (i.e., there are no edges among observed variables), only the ancestors of each node can have causal effects on the target node. Thus, the ambiguity does not seem to be surprising with known causal graphs.

4. Assuming a known causal graph is a relatively strong assumption. Since the considered setting is nonparametric, causal discovery is a challenging task. Thus, more discussion on the assumption of a known causal graph would be helpful to support the task.



[1] Hyvarinen, A., Sasaki, H., and Turner, R. (2019). Nonlinear ICA using auxiliary variables and generalized contrastive learning.

**Questions:**

Apart from the comments above, here are some questions:

1. Are there any noises considered during the generating process or it is fully deterministic?

2. If single-node interventions can be performed in practice, in most cases, doesn't this mean that the ground-truth latent variables have already been taken as a prior, especially with the known causal graph?

**Limitations:**

It would be better if a separate discussion of limitations could be included.

---

> ### Author Rebuttal · Authors · 2023-08-09
>
> > _“it seems that the considered setting could possibly be transferred to the traditional one where there is no dependence among latent variables.”_
>
> Thank you for this suggestion. As we explain below, CauCA cannot be reduced to nonlinear ICA when the graph is non-trivial. The intuition is that the space of possible models in CauCA is much broader than in ICA: the distributions that satisfy $p(x,y)=p(x)p(y)$ are only a small subset of all distributions that are Markovian to the graph $x \to y$.  Thus, the CauCA problem is strictly harder than ICA. We are happy to elaborate further in case this point is unclear, and if you find this point interesting, we will be happy to include it in the revised version.
>
> > _“the novelty compared to the previous results (e.g., [1] and many others) does not appear to be fully clear. [...] the principal difference and connection, in both setting and proof techniques, should also be discussed deeper in other parts of the paper.”_
>
> We can provide a detailed technical argument that all the proofs in our work are different and irreducible to Thm. 1 of [1]---with the exception of Prop. 4.7, which might have caused the confusion. In our work, we introduced a different proof technique, which extends them to the setting of non-trivial latent graphs. Due to the character limit, we are unable to provide technical details here, but we can do so during the author-reviewer discussion period in case you are interested. Based on your suggestion, we will also clarify this in the paper.
>
> > _“The required number of environments is strictly lower but at the cost of interventional data.”_
>
> Previous results in nonlinear ICA rely on some form of assumption on interventional data (i.e., the variability assumption is considered as a condition on fat-hand interventions involving all variables simultaneously). From this point of view, our work elucidates how different kinds of technical assumptions on the interventions may lead to different identifiability results, requiring more or fewer environments.
>
> > _“With imperfect intervention, the ambiguity of the identification is a mixture of nodes that are ancestors of the target node. [...] In the considered setting (i.e., there are no edges among observed variables), only the ancestors of each node can have causal effects on the target node. Thus, the ambiguity does not seem to be surprising with known causal graphs.”_
>
> We believe the resulting ambiguity is indeed surprising. Your comment may stem from a misunderstanding: for example, even with a known graph, a simple violation of Assumption 4.1 yields a much broader ambiguity set, since it can lead to an arbitrary mixture between two latent models $Z$ and $Z’$: $z_1=h_1(z’_1, z’_2)$, $ z_2=h_2(z’_1, z’_2)$.
>
> To visualize the different identifiability results we proved for both imperfect and perfect interventions, we refer to Table 1 in PDF. There, it can be seen that the ambiguities in our results are always strictly smaller than arbitrary mixtures of all variables. Therefore, our results do not trivially follow from the assumption of a known graph, and we will try to make this more clear.
>
> > _“the paper [...] suggested that assuming the known structure is reasonable since 'there are only finitely many possible DAGs'. [...] the exponential nature of the DAG space, although finite, is still unacceptable in most scenarios.”_
>
> We agree with the reviewer about the large cardinality of possible DAGs, and we did not mean to imply this is a generally acceptable search space in CRL. Rather, we meant to highlight that we introduced a new approach which is unlike previous approaches to CRL, which use strong assumptions on the _uncountable_ sets $\mathcal{F}$ and $\mathcal{P}_G$.
>
> > _“Since the considered setting is nonparametric, causal discovery is a challenging task. Thus, more discussion on the assumption of a known causal graph would be helpful to support the task.”_
>
> The known causal graph is indeed a strong assumption, but in some settings, as argued by reviewer 8r7j, it might be possible to “use domain knowledge of what causal variables are involved, and how they are related”.
>
> > _“Are there any noises considered during the generating process or it is fully deterministic?”_
>
> There is stochasticity in the following sense: The generating process is close to ICA, with the difference being that there are causal dependencies among latent variables. As usual in nonlinear ICA literature, the mixing function is deterministic. While we did not consider additional observational noise in the process, we believe that our results could be extended to include noisy observations—similar to how results in nonlinear ICA with auxiliary variables [1] can be extended to observations with additive noise [2].
>
> > _“If single-node interventions can be performed in practice, in most cases, doesn't this mean that the ground-truth latent variables have already been taken as a prior, especially with the known causal graph?”_
>
> If we understand your question correctly, the answer is no: we do not have access to any latent distribution. What we observe is only the discrete graph $G$, and the dataset in observation space. The causal mechanisms and intervention distributions are all unknown.
>
> > _“It would be better if a separate discussion of limitations could be included.”_
>
> Thanks for the suggestion. Based on this comment and on a similar one by reviewer xM6s, and on the discussion with all reviewers, we will add a paragraph discussing Limitations, see the response to all reviewers for details.
>
> Finally, thanks for all the detailed questions. We take this to mean that you found the paper interesting in spite of the relatively low score, and we hope that our answers have improved your view of the paper.
>
> [1] Hyvarinen, A. et al. (2019). Nonlinear ICA using auxiliary variables and generalized contrastive learning.
>
> [2] Khemakhem, I. et al. (2020). Variational autoencoders and nonlinear ICA: A unifying framework.

---

> > ### Comment · Area_Chair_Vgu8 · 2023-08-18
> > **Response to author rebuttal**
> >
> > Dear reviewer,
> >
> > The author rebuttal appears to have presented several targeted responses to your questions.
> >
> > Are your questions appropriately addressed?
> > If they are, would you consider re-assessing your score in light of them.
> > If not, please do provide additional context and feedback to the author.
> >
> > In either case, please provide an acknowledgement of the effort the authors put in, why your questions have (or have not) been addressed and what your assessment of the work is in light of this evidence with a view to reach consensus with the other reviewers on this work.
> >
> > -AC

---

> > > ### Comment · Reviewer_xr1H · 2023-08-18
> > >
> > > Thank you for the detailed responses, and thanks AC for the reminder. After reading the responses and considering discussions from other reviews, I still have several questions about my concerns. Of course, please let me know if I have misunderstood anything.
> > >
> > > - When considering the transfer of the given setting to the original ICA problems with a known structure, I'm referring to a process like the following: Suppose we have [{z2} -> x1, {z1} -> x2, z1->z2] as the ground-truth structure in CauCA. It appears we can directly transfer this to [{z1, z2} -> x1, {z1} -> x2] since all observed variables influenced by z2 are also influenced by z1 (given z1->z2). Earlier works have addressed the identifiability of latent variables by assuming distributional changes, which includes the introduction of interventional data. Since the proposed method doesn't aim to uncover the hidden structure among latent variables (because it's provided as an assumption), the principal difference is not very clear to me.
> > >
> > > - Considering that the ultimate goal of identifiability theory is component-wise identifiability, the "known intervention targets" assumption seems somewhat akin to disentangling individual latent variables before the estimation process. If we already know the targets of single-node interventions, these latent variables appear to be pre-disentangled.
> > >
> > > - In addition to the lacking of application scenarios discussed by other reviewers, I'm not fully convinced of its theoretical significance, especially when compared to existing works on nonlinear ICA. The combined assumptions of known structures and known intervention targets seem relatively unrealistic. They provide a substantial amount of information about the hidden data-generating process, especially when indeterminacies like element-wise rescaling persist after estimation. It feels like too much information is pre-supplied in the form of assumptions. Specifically, we would like to recover the generating process both qualitatively and quantitatively. However, in the proposed setting, the information from both perspectives is either given as assumptions or not possible to recover. From a qualitative perspective, the structural information is provided through known graphs, and the individual latent variables are pre-disentangled, allowing us to ascertain the targets of single-node interventions prior to estimation. Quantitatively, the indeterminacies introduced by element-wise rescaling render the recovery of the effect theoretically impossible.

---

> > > > ### Author Response · Authors · 2023-08-19
> > > >
> > > > Thank you for engaging in a discussion of our work. We address your remaining questions in our reply below. Please do not hesitate to ask in case you have additional questions.
> > > >
> > > > > “Suppose we have [{z2} -> x1, {z1} -> x2, z1->z2] as the ground-truth structure in CauCA. It appears we can directly transfer this to [{z1, z2} -> x1, {z1} -> x2] since all observed variables influenced by z2 are also influenced by z1 (given z1->z2).”
> > > >
> > > > In a latent CBN model (Def. 3.1), the graph $G$ only describes the causal relationships among the latent variables $\mathbf{Z}$: **it contains no information on the functional relationship between the $\mathbf{Z}$ and the $\mathbf{X}$ variables**, which is described by the unknown mixing function $\mathbf{f}$. Moreover, **$\mathbf{f}: (z_1, z_2) \mapsto (x_1, x_2)$ maps all variables in $\mathbf{Z}$ to all variables in $\mathbf{X}$**. The objective we focus on is _blind source separation_: i.e., reconstructing (up to unavoidable ambiguities) the sources $z_1, z_2$ based on the observed mixtures $x_1, x_2$.
> > > >
> > > > The mixing described in your example can be considered as a trivial case in our setting, since the sources $z_1, z_2$ would already be “separated” in the observed components $x_1, x_2$. Conversely, in our work, we allow $\mathbf{f}$ to be any diffeomorphism: this matches the generative model of nonlinear ICA and many works on CRL such as [1][2][3][4].
> > > >
> > > > We hope that our explanation above also clarifies that the example you provided cannot be considered as a reduction of CauCA to the ICA setting, since, in the setting you described, the variables $z_1$ and $z_2$ are not independent, and the mixing is “trivial”. As we explained in the first paragraph of our rebuttal, we believe that such a reduction is impossible.
> > > >
> > > > > “Earlier works have addressed the identifiability of latent variables by assuming distributional changes, which includes the introduction of interventional data. Since the proposed method doesn't aim to uncover the hidden structure among latent variables (because it's provided as an assumption), the principal difference is not very clear to me.”
> > > >
> > > > To explain the contributions of CauCA, we can refer to the two closest lines of work. (1) The first one is interventional CRL, which differs from CauCA by considering an **unknown** latent graph, and where identifiability results are based on strong assumptions on either the mixing (e.g.,  linear [5]) or the latent SCMs (e.g.,  linear [2]); and (2) nonlinear ICA with auxiliary variables, where the mixing is nonlinear and the graph is **known and trivial** (i.e., empty, with no arrows).
> > > >
> > > > The difference is that in our work we consider a **known but nontrivial graph**, and a **nonlinear** mixing and latent SCM. The extra information on known (single or block) targets up to graph automorphism allows us to trade off the complexity coming from considering nontrivial graphs, as opposed to ICA.
> > > >
> > > > Therefore, CauCA cannot be reduced to any of the results in nonlinear ICA (see the answer above), and is not a special case of any existing results in interventional CRL.
> > > >
> > > > > “Considering that the ultimate goal of identifiability theory is component-wise identifiability, the "known intervention targets" assumption seems somewhat akin to disentangling individual latent variables before the estimation process. If we already know the targets of single-node interventions, these latent variables appear to be pre-disentangled.”
> > > >
> > > >
> > > > To show that the latent variables are not pre-disentangled in this setting, we refer to Fig.2 in the manuscript (we were happy to read that you found it “particularly helpful”): without assumption 4.1, even if we already know the targets of single-node interventions in the latent model $(\mathbf{f}, (\mathbb{P}^k_Z)_{k\in \{0, \ldots, d\}})$,
> > > >
> > > >  there are still uncountably many spurious solutions $(\mathbf{f}’, (\mathbb{Q}^k_Z)_{k\in \{0, \ldots, d\}})$
> > > >  such that $\mathbf{f} \mathbb{P}^k = \mathbf{f}’ \mathbb{Q}^k$ $  \forall k \in \{0, \ldots, d\}$
> > > >
> > > >  while $\mathbf{f}’^{-1} \circ \mathbf{f}$ is not an elementwise scaling.
> > > >
> > > >
> > > > This shows that disentangling the latent variables is not trivial even with known intervention targets.
> > > >
> > > >
> > > > > “Quantitatively, the indeterminacies introduced by element-wise rescaling render the recovery of the effect theoretically impossible.”
> > > >
> > > > Our Lemma 3.3 proves that the ambiguity up to element-wise scaling is unavoidable in CauCA if no constraints exist in the latent distributions. Therefore, identifiable up to element-wise monotonic scaling is the best one can hope for. Notice that in those ICA/CRL papers where the mixing function is any diffeomorphism, all the identifiability results allow ambiguities up to elementwise scalings: [1][2][3][4], etc. Even in Thm3 of [1] and Thm1 of [4], identifiability up to affine transformation of the $nk$ sufficient statistics in an exponential model is in fact still a *nonlinear* ambiguity of $\mathbf{f}$.

---

> > > > > ### Author Response · Authors · 2023-08-19
> > > > >
> > > > > One could criticize that the assumption of a known graph is unrealistic. Viewed as a special case of our method, ICA is also making an unrealistic assumption: that of a trivial graph. This analogy might lead you to think that our methods should somehow be reducible (modulo a different graph) to those known from nonlinear ICA. However, in CauCA, we suppose the distributions are **Markov** to a given graph $G$, but not necessarily **faithful** to $G$. This implies that independent components are allowed in any CauCA model $(G,\mathcal{F}, \mathcal{P}_G)$. Since the independent component distributions are a null subset in the set of distributions Markov to a nontrivial G, CauCA is a strictly harder problem than ICA.
> > > > >
> > > > > To further clarify the difference between CauCA and nonlinear ICA, we will detail how our proofs differ from the ones in nonlinear ICA. We thus far omitted this for conciseness, but we will be happy to elaborate on this in the Appendix.
> > > > >
> > > > > For the special cases of CauCA where there are no edges in the causal graph, corresponding to ICA, Thm.1 of [1] proves identifiability with (in our terminology) $2d$ interventional environments of unknown intervention targets ($d$ is the number of nodes). Our identifiability results can then be compared with Thm. 1 of [1] in two cases:
> > > > >
> > > > > i) **Trivial graph—Previous work:** A core step in the proof of Thm. 1 of [1] is in the last two equations on page 23 of CauCA (which is essentially the same as eq.(20,21) of [1]). We will refer to the proof of Prop. 4.7 of CauCA in the following (since the proof of _this proposition specifically_ is indeed similar to the one of Thm.1 of [1]): we take twice the partial derivatives of the equation below eq.(30) of CauCA, i.e., we calculate the Hessian matrix of the two sides. Then one can obtain a system of $2d$ equations, and identifiability corresponds to the uniqueness of the solution of the system $\mathbf{0}=\mathbf{A}\mathbf{x}$, where $\mathbf{A}$ is $\mathbb{R}^{2d \times 2d}$. So $\mathbf{A}$ needs to be invertible—i.e., the “variability” assumption.
> > > > >
> > > > > **Our work:** Our Prop. 4.4 and Thm. 4.6 show that if we know the intervention targets, then _only the Jacobian_ of the above equation, instead of Hessian, is required, and thus the linear system will become $d$ equations instead of $2d$.
> > > > >
> > > > > ii) **Nontrivial graph:** Our proof of Thm. 4.2 is even farther from all ICA proof techniques, including [1]. A peculiarity of ICA is that, after taking the Hessian, the LHS of the last equation on page 23 of CauCA becomes zero. Unfortunately, with a nontrivial graph, it is impossible to get $0$ after taking the partial derivatives of every coordinate of $z$, c.f. eq. (20) of our paper. All the techniques of nonlinear ICA rely on this zero LHS over every coordinate of $z$.
> > > > >
> > > > > **Therefore, all the proofs in our work are indeed different and irreducible to Thm. 1 of [1]**---with the exception of Prop. 4.7, which might have caused the confusion.
> > > > >
> > > > > [1] Hyvarinen, A. et al. (2019). Nonlinear ICA using auxiliary variables and generalized contrastive learning.
> > > > >
> > > > > [2] Buchholz, S. et al.. (2023). Learning Linear Causal Representations from Interventions under General Nonlinear Mixing.
> > > > >
> > > > > [3] Brehmer, J., et al. (2022). Weakly supervised causal representation learning.
> > > > >
> > > > > [4] Khemakhem, I. et al. (2020). Variational autoencoders and nonlinear ICA: A unifying framework.
> > > > >
> > > > > [5] Seigal, A et al. (2022) Linear causal disentanglement via interventions.

---

### Official Review · Reviewer_nsXw · 2023-07-07

**Soundness:** 3 good
**Presentation:** 3 good
**Contribution:** 3 good
**Rating:** 7
**Confidence:** 2

**Summary:**

The authors consider a class of settings in the realm of causal representation learning (CRL) and study their identifiability. More specifically, they consider Causal Component Analysis where the latent causal graph of the generative model is known and interventions are performed. As the authors show, the problem is still not trivial - the mixing function and the mechanisms are to be learned - and identifiability only up to some transformation is possible. The authors prove identifiability for several different cases: the type of interventions performed (perfect vs imperfect, single-node vs fat-hand) and the structure of the latent causal graph (empty vs non-empty). Finally, the authors corroborate their theoretical findings by performing experiments with synthetic data.

**Strengths:**

The article is clear and well-written. It is original as it addresses a slightly different setting than "classical" CRL. The results can have an impact on CRL research since they show in easier cases what is the best form of identifiability one can achieve.

**Weaknesses:**

One limitation is the fact that the settings considered are extremely artificial and contain a lot of knowledge about the problem at hand. The latent graph is known and interventions are performed on almost all latent variables. So, compared to other CRL works, the assumptions are too strong to have practical applications. However, the goal of the authors seems more to have identifiability that gives a kind of "upper bound" (i.e., the identifiability in this simpler setting is the best we can do) for actual CRL that considers weaker assumptions.

Another weaker part is the experiments. I understand that there are toy experiments to validate the identifiability results, but I am not sure that the baselines are that interesting: linear and empty graphs. Could it be compared to another method that learns the graph? Some results could be interpreted or explained in more detail (see the Question section).

**Questions:**

### Questions
- Surprisingly in the results presented in Fig.4 a) and e) the empty graph, which has no information about the latent causal graph, seems pretty competitive with CauCA. Any thoughts on why it is the case?

- Also, any intuition why having more non-linearity would lead to a lower MCC (Fig 4.c)?

- Another interesting parameter to investigate would be the edge density of the latent graph. Have you considered this experiment?


### Typo:
- line 72: "leads extensions", words are missing
- line 98: "the" is repeated
- Fig 2, caption: "As can be seen", words are missing
- Some punctuation marks are missing in the equation (e.g., Eq.5)
- line 193: "untintervened"
- Fig 4.c, "No. nonliniearities"
- Eq. 9, the second term of LHS is missing a log.
- Suggestion: I would specify the type of normalizing flow employed in the main text (I know it is in the appendix).

**Limitations:**

While the assumption are clearly defined in the different theoretical results, the limitation could be more explicitly presented (see my first point in the weakness). Societal impact is not applicable.

---

> ### Author Rebuttal · Authors · 2023-08-09
>
> Thank you for your positive evaluation of our work.
>
> Comments on theory:
>
> > _“The latent graph is known and interventions are performed on almost all latent variables. [...] the assumptions are too strong to have practical applications.”_
>
> We agree these are strong assumptions. As you said, part of our motivation is in seeking _“identifiability that gives a kind of "upper bound" (i.e., the identifiability in this simpler setting is the best we can do) for actual CRL that considers weaker assumptions”_. In the context of state-of-the-art in CRL, there is a complete lack of similar identifiability results without stronger assumptions on the mixing function, or latent SCM, or on the nature of the available data (e.g., counterfactual as opposed to interventional). For this reason, we maintain that our investigation of the setting with a known graph is of significant interest for CRL theory.
>
> Regarding the number of interventions, requiring interventions on all latent variables is not specific to our work: a similar assumption can also be found in [3]. We also prove an **additional result** (see response to 8r7j) showing that without the assumption that each variable is intervened at least once, it is impossible to achieve identifiability up to scaling.
>
> Questions on Experiments:
>
> > _“Surprisingly in the results presented in Fig.4 a) and e) the empty graph [...] seems pretty competitive with CauCA. Any thoughts on why it is the case?”_
>
> In the ground truth latent Structural Causal Model (SCM), there are two contributions to each variables’ variance: one given by the influence from its parent variables, and one given by the exogenous noise (for technical details, see Equation 42, Appendix G). These two contributions can have different strengths, prompting us to introduce a parameter to quantify the relative importance of the causal influence from parent variables and the impact of exogenous noise (“signal-to-noise ratio (SNR)”, see line 946 in Appendix G).
>
> At low SNR values, the variance of each latent variable is predominantly driven by the exogenous noise, while the causal influence from parent variables has a lesser impact. In contrast, high SNR values result in a more pronounced influence of the parent variables on the variance of each latent variable. Notably, when the SNR approaches zero, the setting becomes close to Independent Component Analysis (ICA), where all latent variables are independent.
>
> In Figure 4 (a), we present the MCC results for a ground truth latent SCM with SNR=1, where the contribution of the independent exogenous noise is relatively strong. In this case, the misspecified baseline, which assumes independence among the latent variables (indicated in yellow), appears to provide a reasonable approximation of the latent structure.
>
> Additionally, in Figure 4 (g), we explore how CauCA behaves relative to the misspecified baseline as the SNR increases. The CauCA model consistently and effectively recovers the ground truth latent variables even for high SNR values. In contrast, the misspecified baseline performs substantially worse as the SNR rises, as approximating the ground truth latents as independent becomes a poorer representation in the presence of stronger causal influences.
>
> > _“Any intuition why having more non-linearity would lead to a lower MCC (Fig 4.c)?”_
>
> Our intuition behind the observed trend of lower MCC with an increasing nonlinearity (as shown in Figure 4 (c)) stems from the complexity of the mixing function. As we introduce more nonlinearity into the model, the mixing function becomes progressively more intricate, making it inherently challenging to learn an accurate approximation of its inverse.
>
> We note that the degradation of the MCC with an increasing number of layers in the ground truth mixing function _is not specific to CauCA_: in fact, it can also be observed in nonlinear ICA works, see, e.g., [1] Figure 2, and [2] Figures 2 and 3.
>
> The results depicted in Figure 4 (c) represent the best performance we were able to achieve. A more exhaustive fine-tuning of the model architecture and training parameters might further improve performance in cases with strong nonlinearities.
>
> > _“Could [CauCA] be compared to another method that learns the graph?”_
>
> The closest method that solves the same problem as CauCA (identifiable representation learning based on multiple _interventional_ datasets) and _additionally_ learns the graph, is [3]: however, the method assumes a linear mixing. Based on this, we thought that a linear method should feature as a baseline: note that the performance of our linear baseline should in principle upper bound the method by [3], since it assumes strictly more information (knowledge of the causal graph). The most closely related nonlinear methods are in nonlinear ICA with auxiliary variables, which motivated our choice of the empty graph as an additional baseline.
>
> > _“Another interesting parameter to investigate would be the edge density of the latent graph”_
>
> Thank you for the suggestion. We performed an additional experiment to investigate this, see the results in the PDF attached in our response to all reviewers: the method’s performance degrades gracefully for higher edge densities. This is expected, since more edges means more free parameters in the conditional, and (loosely speaking) the distributions that have fewer parameters are of measure zero in the set of the distribution that have more parameters: CauCA with more edges is strictly harder than with fewer.
>
> Also, thanks for pointing out the typos, which we will correct in the revised version.
>
> [1] Hyvarinen, Aapo, and Hiroshi Morioka. (2016) "Unsupervised feature extraction by time-contrastive learning and nonlinear ICA."
>
> [2] Hyvarinen, Aapo, and Hiroshi Morioka. (2017) “Nonlinear ICA of temporally dependent stationary sources."
>
> [3] Squires, Chandler, et al. (2023) "Linear Causal Disentanglement via Interventions."

---

> > ### Comment · Reviewer_nsXw · 2023-08-11
> >
> > I appreciate the comprehensive response provided by the reviewers, which effectively addressed my concerns. I am also thankful for the inclusion of additional experiments. I believe this paper should be accepted.

---

### Official Review · Reviewer_xM6s · 2023-07-27

**Soundness:** 3 good
**Presentation:** 4 excellent
**Contribution:** 3 good
**Rating:** 7
**Confidence:** 1

**Summary:**

This paper introduces the concept of Causal Component Analysis (CauCA), which lies between Independent Component Analysis (ICA) and Causal Representation Learning (CRL). CauCA generalizes ICA by considering the causal dependence among latent components while assuming knowledge of the causal graph, focusing on learning the unmixing function and causal mechanisms. The paper establishes identifiability results for CauCA based on multiple datasets generated through interventions on the latent causal variables, leading to new identifiability findings for nonlinear ICA. It also proposes a likelihood-based approach using normalizing flows to estimate the unmixing function and causal mechanisms, demonstrating its efficacy through extensive synthetic experiments in both CauCA and ICA settings.

**Strengths:**

Strengths of this paper include a comprehensive presentation of Causal Component Analysis (CauCA) and its connections to existing methods such as Independent Component Analysis (ICA) and Causal Representation Learning (CRL). The paper thoroughly characterizes the identifiability of CauCA from multiple datasets, considering various types of interventions on latent causal variables, even in the nonlinear and nonparametric case. By leveraging the modularity of causal relationships, the paper introduces a novel perspective to tackle nonlinear ICA, leading to new identifiability results that require fewer datasets for the same level of identifiability. The paper also proposes an estimation procedure based on normalizing flows, and extensive synthetic experiments in both CauCA and ICA settings demonstrate the effectiveness of this approach in accurately recovering the latent causal components.

**Weaknesses:**

None.

**Questions:**

None.

**Limitations:**

The authors did not discuss the limitations, so one paragraph on this may need to be added.

---

> ### Author Rebuttal · Authors · 2023-08-09
>
> Thank you for your positive feedback on our work.
>
> > _“The authors did not discuss the limitations, so one paragraph on this may need to be added.”_
>
> Based on this comment and on the discussion with all reviewers, we will add a paragraph discussing limitations, including: (i) Known intervention targets: We discussed this limitation in the response to reviewer 8r7j, and proved that with fully unknown targets there are fundamental and strong limits to identifiability. We also studied some possible relaxations of this assumption, leaving others for future work. (ii) Algorithm: Our likelihood-based algorithm, while a principled approach to CauCA estimation, may not be optimal for applications to high-dimensional data. We believe that more practical algorithms may be developed, e.g., based on variational inference.

---

> > ### Comment · Reviewer_xM6s · 2023-08-13
> >
> > Thank you for your response. I have read the other reviews and rebuttals and will keep my original score.

---

### Author Rebuttal · Authors · 2023-08-09

Thank you for the valuable reviews, acknowledging that Causal Component Analysis is “important for the development of causal analysis in the context of deep learning” (xr1H), and that our work will have “an impact on [Causal Representation Learning] research” (nsXw). Moreover, our results introduce a “novel perspective to tackle nonlinear ICA” (xM6s), leading to “new identifiability results” (xM6s) with “few technical assumptions relative to the existing literature” (8r7j).

Questions raised by the reviewers mostly focused on the assumptions of (1) known causal graph (nsXw, xr1H); (2) known intervention targets (8r7j).

Regarding the assumption of a known causal graph (1): Clearly, this assumption is a choice, and whether this is a good choice is debatable. Most reviewers agree it is an interesting theoretical question to ask what this choice implies, and that it tackles "a middle ground between ICA and CRL" (8r7j). This is what motivated us to study the problem - together with the fact that the previous state-of-the-art causal representation learning (CRL) theory relies on strong assumptions of a different nature to prove identifiability—e.g., restrictions on the mixing function, or on the latent structural causal model, or on the kind of available data (e.g., counterfactual as opposed to interventional).

Regarding (2), we clarify that some knowledge about the intervention targets is necessary, in the sense that dropping all knowledge provably invalidates the identifiability result (see response to 8r7j). Moreover, in Appendix E and in the response to reviewer 8r7j, we generalise the results of our main paper, requiring strictly less information on the intervention targets. This was only added to the Supplementary material due to space constraints, but we will expand on this in the revised version of our paper using the additional page.

Based on the suggestions of (xM6s, xr1H), we will also add a “Limitations” section, also including the additional results we derived in answering the reviewers’ questions, discussing the following points: (i) Known intervention targets: We discussed this limitation in the response to reviewer 8r7j, and proved that with fully unknown targets there are fundamental and strong limits to identifiability. We also studied some possible relaxations of this assumption, leaving others for future work. (ii) Algorithm: Our likelihood-based algorithm, while a principled approach to CauCA estimation, may not be optimal for applications to high-dimensional data. We believe that more practical algorithms may be developed, e.g., based on variational inference.

Finally, to address a question by reviewer nsXw, we performed an ablation experiment, where we varied the edge density of the underlying graph: see Figure 1 in the attached PDF, showing that our method’s performance degrades gracefully as the graph becomes closer to fully connected.

---

### Decision · Program_Chairs · 2023-09-21

**Decision:**

Accept (poster)

**Comment:**

The paper provides identifiability results for causal component analysis, where the causal structure of the latent variables is assumed to be known and access to interventional data (on the latent variables) is present. The work requires few assumptions on the mixing process (the function mapping latent variables to observations). While the reviewers found the problem interesting and the results sensible, there were several questions raised about the assumptions made by the work and the relationship to existing work in the context of nonlinear ICA. There was a significant deal of back and forth discussion among the authors and the reviewers on highlighting the limitations of the work, whether CauCA can be reduced to nonlinear ICA (it cannot but a clear exposition of the same is missing in the current manuscript however a sketch for the same was provided in the rebuttal) and the degree to which the assumption of knowing the causal graph among latent variables can ever hold in practice. I agree with the reviewers that result herein is primarily of theoretical interest since finding cases where one has perfect knowledge of the causal graph on the latent processes is indeed a very strong assumption and one would be hardpressed to find practical scenarios where they hold.

Ultimately there was a lack of consensus among the reviewers on this version of the manuscript and this came down to a borderline decision.

I'm going to lean on the side of acceptance because the work itself is solid and even though it makes strong (bordering unrealistic) assumptions, there is care given to the analysis of the results, when they hold, and why they differ from prior work. I believe the result would be of interest to the ICA community and perhaps could serve as a starting point to relax the assumptions herein.

That its primarily of theoretical import is not problematic but I do think there remains room to improve the clarity of the manuscript with respect to the work's theoretical distinction over nonlinear ICA (based on the feedback provided in response to xr1H), a clear eyed view of the limitations of the work and an incorporation of the relaxations (presented to reviewer 8r7j). Please do make the relevant changes to the revised manuscript.